



**Controls on redox-sensitive trace metals in the Mauritanian oxygen minimum zone**
Insa Rapp[1*], Christian Schlosser[1], Jan-Lukas Menzel Barraqueta[1,2], Bernhard Wenzel[1], Jan Lüdke[1],
Jan Scholten[3], Beat Gasser[4], Patrick Reichert[1], Martha Gledhill[1], Marcus Dengler[1], and Eric P.
Achterberg[1]
[1]Helmholtz Centre for Ocean Research Kiel (GEOMAR), Wischhofstr. 1-3, 24148 Kiel, Germany
[2]Department of Earth Sciences, Stellenbosch University, Stellenbosch, 7600, South Africa
[3]Institute of Geosciences, Christian-Albrecht University Kiel (CAU), Otto-Hahn-Platz 1, 24118 Kiel,
Germany
[4]International Atomic Energy Agency (IAEA), Environment Laboratories, 4 Quai Antoine 1er, 98012
Monaco
*Corresponding author. irapp@geomar.de




**ABSTRACT**
The availability of the micronutrient iron (Fe) in surface waters determines primary production, $N_2$
fixation and microbial community structure in large parts of the world´s ocean, and thus plays an
important role in ocean carbon and nitrogen cycles. Eastern boundary upwelling systems and the
connected oxygen minimum zones (OMZs) are typically associated with elevated concentrations of
redox-sensitive trace metals (e.g. Fe, manganese (Mn) and cobalt (Co)), with shelf sediments typically
forming a key source. Over the last five decades, an expansion and intensification of OMZs has been
observed and this trend is likely to proceed. However, it is unclear how trace metal (TM) distributions
and transport are influenced by decreasing oxygen ($O_2$) concentrations. Here we present dissolved (d;
<0.2 µm) and leachable particulate (Lp; >0.2 µm) TM data collected at 7 stations along a 50 km
transect in the Mauritanian shelf region. We observed enhanced concentrations of Fe, Co and Mn
corresponding with low $O_2$ concentrations (<50 µmol $kg^{-1}$), which were decoupled from major
nutrients and nutrient-like and scavenged TMs (cadmium (Cd), lead (Pb), nickel (Ni) and copper
(Cu)). Additionally, data from repeated station occupations indicated a direct link between dissolved
and leachable particulate Fe, Co, Mn, and $O_2$. An observed dFe decrease from 10 to 5 nmol $L^{-1}$
coincided with an $O_2$ increase from 30 to 50 µmol $kg^{-1}$ and with a concomitant decrease in turbidity.
The changes in Fe (Co and Mn) were likely driven by variations in their release from sediment pore
water, facilitated by lower $O_2$ concentrations and longer residence time of the water mass on the shelf.
Variations in organic matter remineralization and lithogenic inputs (atmospheric deposition or
sediment resuspension) only played a minor role in redox-sensitive TM variability. Vertical dFe fluxes
from $O_2$-depleted subsurface to surface waters (0.08–13.5 µmol $m^{-2}$ $d^{-1}$) were driven by turbulent
mixing and vertical advection, and were an order of magnitude larger than atmospheric deposition
fluxes (0.63–1.43 µmol $m^{-2}$ $d^{-1}$). Benthic fluxes are therefore the dominant dFe supply to surface
waters on the continental margins of the Mauritanian upwelling region. Overall, our results indicated
that the projected future decrease in $O_2$ concentrations in OMZs may result in increases in Fe, Mn and
Co concentrations.
**1. INTRODUCTION**
The micronutrient iron (Fe) is essential for phytoplankton growth, but due to biological uptake
coupled with a low solubility and low supply rates, the availability of Fe is typically low  in open
ocean surface waters (Bruland and Lohan, 2006). As a result, Fe limits primary production in high
nitrate low chlorophyll regions (Boyd, 2007) and regulates dinitrogen ($N_2$) fixation in (sub)-tropical
waters (Moore et al., 2009). Alongside Fe, other trace metals (TMs) such as cobalt (Co), manganese
(Mn), zinc (Zn), cadmium (Cd) and copper (Cu) may (co-)limit phytoplankton growth and influence
community composition (Browning et al., 2017; Moore et al., 2013; Morel and Price, 2003; Saito et
al., 2008).



Oxygen minimum zones (OMZs) are characterized by stable subsurface oxygen ($O_2$) minima, which
are maintained by a combination of enhanced $O_2$ consumption in the thermocline and a limited supply
of $O_2$ rich water masses (e.g. Brandt et al., 2015; Karstensen et al., 2008; Wyrtki, 1962). Enhanced $O_2$
consumption is a result of elevated surface productivity caused by upwelling of nutrient-rich
subsurface waters in eastern boundary regions of the oceans through Ekman divergence, and intense
remineralization of sinking particles (e.g. Helly and Levin, 2004). Elevated organic matter supply and
water column $O_2$ depletion lead to enhanced benthic release of redox-sensitive elements by influencing
sediment diagenetic processes (Noffke et al., 2012; Severmann et al., 2010). Elevated concentrations
of sediment derived dissolved Fe, Co and Mn have been associated with lateral offshore advection in
$O_2$ depleted waters in the Arabian Sea, Pacific and Atlantic Ocean (Biller and Bruland, 2013; Hatta et
al., 2015; Hawco et al., 2016; Milne et al., 2017; Moffett et al., 2015; Noble et al., 2012).
Oxygen concentrations affect the distribution of redox-sensitive TMs by controlling oxidation rates
and influencing microbially mediated redox transformations. The reduced form of redox-sensitive
TMs, such as iron (Fe(II)), cobalt (Co(II)) and manganese (Mn(II)), have a higher solubility in
aqueous solutions than their oxidized forms (Fe(III), Co(III), Mn(III/IV)) (Liu and Millero, 2002;
Stumm and Morgan, 1995). Reduction of these metals occurs to a large extent in anoxic sediment pore
waters by microbial induced dissolution of particulate Fe(III) and Mn(III/IV) oxyhydroxides (Burdige,
1993; Chaillou et al., 2002; Froelich et al., 1979). Sediment pore waters are released to overlying
bottom waters by diffusion and bio-irrigation and during submarine groundwater discharge (Beck et
al., 2007; Elrod et al., 2004; Green et al., 2002). In contact with $O_2$ and other oxidants (e.g. nitrate
(Schlosser et al., 2018) and hydrogen peroxide (Moffett and Zika, 1987)), Fe(II) oxidizes to the poorly
soluble Fe(III) species, that are rapidly transformed into amorphous Fe oxyhydroxides or scavenged
onto particle surfaces (Moffett and Zika, 1987; Scholz et al., 2016; Wu and Luther, 1994). Mn(II) also
oxidizes to insoluble Mn(III/IV) oxides, but due to the slow abiotic oxidation kinetics, especially
under low $O_2$ conditions (e.g. von Langen et al., 1997), biotic oxidation by manganese oxidizing
bacteria is the main oxidation mechanism for Mn (Moffett, 1994; Sunda and Huntsman, 1988; Tebo
and Emerson, 1986). Co(II) removal is mainly associated with incorporation of Co into Mn oxides by
Co co-oxidation (Moffett and Ho, 1996).
Stabilizing mechanisms that prevent removal by scavenging and precipitation of Fe, Co and Mn are
organic ligand complexation (Elrod et al., 2004; Liu and Millero, 2002; Oldham et al., 2017; Parker et
al., 2007) and adsorption onto small slow sinking or neutral buoyant particles (Lam et al., 2012).
Recent studies suggest a potentially important role for dynamic exchange processes between dissolved
and particulate phases of Fe, thereby influencing cycling and transport (Achterberg et al., 2018;
Fitzsimmons et al., 2017; Labatut et al., 2014; Milne et al., 2017). This was further indicated by Fe
isotope studies suggesting an equilibrium isotopic fractionation between dissolved and particulate





phase in deep waters (Labatut et al., 2014) and the concomitant deepening of the dissolved and
particulate Fe plume that originated from a hydrothermal vent (Fitzsimmons et al., 2017).
Spatial and seasonal variations in TMs that are released from sediments, as well as ex-situ sediment
incubation experiments suggest a direct influence of bottom water and water column $O_2$ concentrations
on the distribution of Fe, Co and Mn (e.g. Biller and Bruland, 2013; Homoky et al., 2012). Differences
in benthic TM supply in field studies however are also influenced by a range of other processes as for
example sediment type and organic matter supply (Homoky et al., 2016). Ex-situ sediment incubation
experiments offer a potential means to disentangle the influence of $O_2$ concentrations relative to these
controls (Homoky et al., 2012). These experiments, however, need to be interpreted within the context
of the confined conditions that eliminate potentially important interactions in open systems, such as
seawater exchange and mixing. Furthermore, they offer no means to confidently evaluate controls on
TM distributions in the pelagic water column.
In an attempt to resolve the controls on TM release and stabilization in OMZs we measured the
concentration of a suite of TMs along a 50 km long transect on the Mauritanian shelf in the Eastern
Tropical North Atlantic (ETNA). The Mauritanian shelf is associated with a major OMZ (minimum $O_2$
concentrations below 40 µmol kg$^{-1}$; Brandt et al., 2015) and is an important Fe source to the North
Atlantic Ocean (Milne et al., 2017). Furthermore, atmospheric dust deposition from the Saharan desert
can markedly elevate surface water Fe concentrations in the ETNA (Conway and John, 2014;
Rijkenberg et al., 2012). Recent observations suggest a decline in $O_2$ content of the oceans,
particularly in the northern and southern eastern Atlantic, and an expansion of OMZs, modulated by
the variability of our climate system (Hahn et al., 2017; Schmidtko et al., 2017; Stramma et al.,
2008b). These changes may result in changes in TM supply, and a mechanistic understanding of the
factors regulating TM release and stabilization in OMZs is therefore urgently needed. The aim of this
study was to evaluate the direct influence of variability in water column $O_2$ concentrations on the
distribution of redox-sensitive TMs and to identify responsible control mechanisms. Firstly, we assess
the fluxes of dFe in the OMZ to surface waters by advection and diffusive mixing and compared those
to the atmospheric deposition flux of dFe. Secondly, we evaluate the importance of redox and non-
redox controls on Fe, Co and Mn by focusing on the influence of $O_2$ and particles on the distribution
of dissolved and leachable particulate TMs, including redox-sensitive (Fe, Co and Mn) and nutrient-
type and scavenged trace metals (aluminum (Al), lead (Pb), nickel (Ni), Cd and Cu). Thirdly, we
determine the influence of variability of the eastern boundary circulation and $O_2$ concentrations in
regulating TM concentrations.
**2. METHODS**
**2.1 Sampling**





Samples were collected on RV Meteor cruise M107 in June 2014 during nine deployments at seven
locations (two stations were occupied twice) along a cross-shelf transect at 18°20'N on the
Mauritanian shelf in the ETNA (Figure 1). The bottom depths of stations varied between 50 m on the
shelf to 1136 m furthest off shore. Seawater sampling was carried out using a trace metal clean CTD
(TM-CTD, Sea-Bird SBE25) rosette frame equipped with 24 trace metal clean samplers (12 L, Ocean
Test Equipment (OTE)). The CTD frame was attached to plastic coated nonconductive steel cable and
deployed using a carousel auto-fire module (AFM, Sea-Bird) that closed the bottles at predefined
depths. After recovery, the bottles were transferred to a clean-laboratory container and pressurized to
0.2 bar overpressure using filtered $N_2$ gas. Samples were collected unfiltered for total dissolvable (TD)
TM measurements, and filtered using a 0.2 µm cartridge filter (Acropack 500, Pall) for dissolved (d)
TMs and iodide. Trace metal samples were collected in acid clean 125 mL low density polyethylene
(LDPE) bottles (Nalgene), and iodide samples in opaque 60 mL high density polyethylene (HDPE)
bottles (Nalgene). Trace metal samples were acidified to pH 1.9 using ultra clean HCl (UpA, Romil)
and stored double-bagged for >6 months before preconcentration and analysis. Samples for iodide
measurements were stored frozen at -20°C until analysis.
Samples for the determination of radium isotopes ($^{223}$Ra; $t_{1/2}$ = 11.4 d; $^{224}$Ra $t_{1/2}$ = 3.7 d) were obtained
using in-situ filtration pumps (Challenger Oceanic) following the procedures described in Charette et
al. (2015) and Henderson et al. (2013). Briefly, each in-situ filtration pump was equipped with two
particle filters (70 µm; 1 µm) and two Mn dioxide ($MnO_2$) impregnated cartridges (CUNO Micro
Klean III acrylic) on which dissolved Ra adsorbs. The pumped water volumes varied between 1000 L
and 1700 L. For the determination of Ra in surface waters (~5 m water depth) about 200–300 L of
seawater was pumped into 500 L plastic barrels followed by filtration over $MnO_2$ coated acrylic fibers
(Mn-fibers).
**2.2 Trace metal analysis**
Determination of Co, Mn, Fe, Cd, Pb, Ni and Cu was carried out as described in Rapp et al. (2017).
Briefly, samples were preconcentrated using an automated preconcentration device (Sea*FAST*,
Elemental Scientific Inc.) equipped with a cation chelating resin (WAKO; Kagaya et al., 2009).
Samples were UV-digested prior to preconcentration to breakdown metal-organic complexes, which
would cause an underestimation of the determined TM concentrations. Samples were buffered in-line
to pH 6.4 ± 0.2 using 1.5 M ammonium acetate buffer, before loading onto the resin. The pH buffer
was prepared using an ammonium hydroxide solution (22%, OPTIMA grade, Fisher) and acetic acid
(glacial, OPTIMA grade, Fisher) in de-ionized water (MilliQ, Millipore), adjusted to pH 8.5. Retained
TMs were eluted from the resin using 1 M distilled $HNO_3$ and collected in 4 mL polypropylene
scintillation vials (Wheaton). The acid was distilled from supra-pure $HNO_3$ (SpA grade, Romil) using
a sub-boiling PFA distillation system (DST-1000, Savillex). Preconcentration was performed within a
clean laboratory (ISO 5) and all sample and reagent handling was performed within the same



laboratory in an ISO 3 laminar flow bench with a HEPA filter unit. Preconcentrated samples were
analyzed by high resolution inductively coupled plasma-mass spectrometry (HR-ICP-MS, ELEMENT
XR, ThermoFisher Scientific) using isotope dilution for Fe, Cd, Pb, Cu and Ni and standard additions
for Co and Mn. SAFe reference seawater S and D2 were analyzed with each analytical run and
concentrations produced were in good agreement with consensus values (Table 1).
Leachable particulate (Lp) concentrations were calculated as the difference between total dissolvable
and dissolved concentrations. The limit of quantification (LOQ) for the Lp concentrations was
determined as the sum of the analytical standard deviations of TD and dissolved concentrations.
Extended uncertainty calculations were performed using the Nordtest approach (Naykki et al., 2015)
accounting for random as well as systematic errors (Rapp et al., 2017). The Lp fraction represents the
particulate fraction which is readily dissolvable in the acidified samples during storage at pH 1.9 for 6
months and therefore does not contain any refractory particle components. This more labile fraction of
particulate TMs mainly includes TMs in organic/biogenic particles, adsorbed to particle surfaces and
TM oxides/oxyhydroxides (Hurst et al., 2010).
**2.3 Aluminum measurements**
Aluminum concentrations were determined in surface water samples for all stations along the transect
and at two stations (3 and 8) for the entire water column. Samples were analyzed for Al according to
Hydes and Liss (1976). Acidified samples were buffered with a 2 M ammonium acetate buffer (Romil,
UpA) to a pH between 5.1 and 5.2. Buffered samples were spiked with a 2 mg $L^{-1}$ lumogallium (TCI)
solution. The lumogallium solution was prepared in 2 M ammonium acetate buffer (Romil, UpA).
After spiking, samples were heated up for 1.5 h at 80°C in an oven (Heratherm, Thermo Scientific)
and left to cool down overnight at room temperature to allow the formation of a fluorescence Al
complex. Samples were measured using a fluorescence spectrophotometer (Cary Eclipse, Agilent).
The samples were measured with an excitation and emission wavelength of 465 and 555 nm,
respectively. All samples were analyzed in duplicate and the concentrations calculated from the peak
heights via standard addition. GEOTRACES reference seaweater (GS) was run with a mean average
Al value of $27.76 \pm 0.17$ nmol $L^{-1}$ (n=4; consensus value $28.2 \pm 0.2$ nmol $L^{-1}$).
**2.4 Iodide measurements**
Frozen samples were defrosted overnight at room temperature prior to analysis for iodide by cathodic
stripping square wave voltammetry after Luther et al. (1988). The voltammetry unit consisted of a
voltammeter stand (663 VA, Metrohm), an autosampler (863 Compact Autosampler, Metrohm) and an
automatic burette (843 Pump Station, Metrohm) for automated spike addition. The system was
controlled by Computrace software (797 VA; Metrohm).
**2.5 Oxygen, salinity, nutrient, turbidity and chlorophyll fluorescence analysis**



Oxygen, salinity, nutrients, turbidity and chlorophyll fluorescence was measured during 62 CTD
deployments (including some repeated deployments at the same location) along the 18°20`N transect
using a Sea-Bird SBE 9 CTD rosette system equipped with double sensor packages for $O_2$, salinity and
temperature and 24 niskin samplers (10 L; OTE). Turbidity and chlorophyll a fluorescence were
measured with single sensor units on the CTD. Oxygen sensor data were calibrated by Winkler
titration (Hansen, 2007; Winkler, 1988) on 348 discrete water samples that were collected from the
OTE samplers. The $O_2$ calibration was undertaken using a linear fit with respect to $O_2$ concentration,
temperature, and pressure. An uncertainty of 1.5 µmol kg$^{-1}$ was determined. On-board nutrient
measurements of nitrite ($NO_2^-$), nitrate ($NO_3^-$), phosphate ($PO_4^{3-}$) and silicic acid ($Si(OH)_4$) of the
discrete water samples were conducted using a QuAAtro autoanalyzer (Seal Analytical) according to
Grasshoff et al. (1983).
Apparent Oxygen Utilization (AOU) was calculated as the difference between saturation
concentrations of $O_2$ and measured $O_2$ concentrations. The saturation concentration of $O_2$ was
calculated after the Weiss methods (Weiss, 1970) using the R package marelac (Soataert et al., 2016),
taking into account salinity and temperature.
**2.6 Radium analysis**
On-board the ship the Mn-cartridges and Mn-fibers were washed with Ra-free tap water and
afterwards partially dried with filtered compressed air to remove excess water. The samples were
analyzed for $^{223}$Ra, $^{224}$Ra and $^{228}$Th using a Radium Delayed Coincidence Counting System (RaDeCC)
(Moore and Arnold, 1996). For the efficiency calibration of the RaDeCC, $^{227}$Ac and $^{232}$Th standard
solutions were used, and the calibration followed the procedure described in Scholten et al. (2010) and
Moore and Cai (2013). Counting errors were propagated following Garcia-Solsona et al. (2008).
Excess $^{224}$Ra ($^{224}$Ra$_{ex}$), i.e. the $^{224}$Ra activity corrected for $^{228}$Th-supported $^{224}$Ra was calculated by
subtracting the $^{228}$Th activity from the $^{224}$Ra activity. As we measured only the first Mn cartridge and
the Mn cartridges do not adsorb radium quantitatively, we report here only $^{224}$Ra$_{ex}$/$^{223}$Ra ratios.
**2.7 Turbulence measurements and vertical flux calculations**
In order to advance understanding of the role of benthic Fe supply to the high productive surface
waters of the upwelling region, vertical diffusive fluxes (eq 1: left term, right hand side) and upwelling
induced vertical advective fluxes (eq 1: right term, right hand side) were estimated. At depth in the
water on a continental margin, solutes are transferred vertically toward the surface waters by turbulent
mixing processes and by vertical advection forced by Ekman divergence (e.g. Steinfeldt et al., 2015):

$$J_Z = K_z \frac{\partial [TM]}{\partial z} + w \cdot \Delta [TM] \qquad (1)$$



Here, $K_z$ is the turbulent eddy diffusivity in m$^2$ s$^{-1}$, $\partial[TM]/\partial z$ the vertical gradient with depth (z) of the
TM concentration [TM] in µmol m$^{-4}$, $\Delta[TM]$ a TM concentration difference in µmol m$^{-3}$ and $w$
represents vertical velocity in m s$^{-1}$. The equation is solved by vertically integrating the tracer transport
budget equation between two vertical layers while ignoring lateral fluxes, changes of $w$ with depth and
assuming steady state. Vertical advective fluxes resulting from meso- and submesoscale processes
along sloping isopycnals were not considered. The TM-fluxes were evaluated for the depth interval
from the upper boundary of the shallow O$_2$-depleted waters to a depth of increased chlorophyll a
fluorescence (8–29 m depth).
Diffusive Fe fluxes were determined by combining TM concentration measurements from the TM-
CTD stations with nearby measured microstructure profiles. The microstructure measurements were
performed with an MSS90-D profiler (S/N 32, Sea & Sun Technology). The loosely-tethered profiler
was optimized to sink at a rate of 0.55 m s$^{-1}$ and equipped with three shear sensors, a fast-response
temperature sensor, and an acceleration sensor, two tilt sensors and conductivity, temperature, depth
sensors sampling with a lower response time. At TM-CTD stations with bottom depths less than 400
m, 18 to 65 microstructure profiles were available at each station. At deeper stations, the number
reduced to 5 to 12 profiles. Standard processing procedures were used to determine the rate of kinetic
energy dissipation (ε) of turbulence in the water column (see Schafstall et al. (2010) for detailed
description). Subsequently, $K_Z$ values were determined from $K_p = \Gamma \varepsilon N^{-2}$ (Osborn, 1980), where $N$ is
stratification and $\Gamma$ is the mixing efficiency for which a value of 0.2 was used. The use of this value
has recently shown to yield good agreement between turbulent eddy diffusivities determined from
microstructure measurements and from tracer release experiments performed in our study region
(Köllner et al., 2016). The 95% confidence intervals for station-averaged $K_\rho$ values were determined
from Gaussian error propagation following Schafstall et al. (2010). Finally, diffusive fluxes were
estimated by multiplying station-averaged $K_\rho$ with the vertical gradient of the respective TM solute,
implicitly assuming $K_Z = K_\rho$.
The vertical advective flux by Ekman divergence requires determination of vertical velocity in the
water column that varies with depth and distance from the coast line. Recent studies found good
agreement between vertical velocities derived from Ekman divergence and from helium isotope
equilibrium within the Mauritanian and Peruvian coastal upwelling regions (Steinfeldt et al., 2015)
when parameterizing vertical velocities as (Gill, 1982):
$w = \dfrac{\tau_y}{\rho f L_r} e^{-x/L_r}$
where $\tau_y$ represents the alongshore wind stress, $\rho$ the density of sea water, $x$ the distance from
maximum Ekman divergence taken here as the position at 50 m bottom depth on the shelf and $L_r$ the
first baroclinic Rossby radius. The parameterization results from considering the baroclinic response




of winds parallel to a coastline in a two-layer ocean (Gill, 1982). The baroclinic Rossby radius
$L_r = f^{-1} \sqrt{g \frac{\rho_2 - \rho_1}{\rho} \frac{H_1 H_2}{H_1 + H_2}}$ ($\rho_{1/2}$ and $H_{1/2}$ is density and thickness of the surface and lower layer,
respectively) was found to be 15 km from hydrography collected during the cruise, similar to the
values determined by Steinfeld et al. (2015) in the same region. Using average alongshore wind stress
from satellite data (0.025 Nm$^{-2}$, ASCAT winds; Ricciardulli and Wentz, 2016) for June 2014,
maximum vertical velocities of $3.7 \times 10^{-5}$ m s$^{-1}$ were determined for the shelf region (50 m water depth),
which decayed offshore to $1.7 \times 10^{-6}$ m s$^{-1}$ at the position of the 1000 m isobath at 18°N. As these
vertical velocities describe the magnitude of upwelling at the base of the mixed-layer, additional
corrections need to be considered for deeper depths. Here, we approximated the vertical decay of $w$ as
a linear function which diminishes at the ocean floor.
The calculation of the vertical advective flux supplying solutes from the shallow $O_2$-depleted waters to
the chlorophyll $a$ maximum requires knowledge of a concentration difference $\Delta$[TM] associated with
the upwelling flux. Ideally, the vertical scale of the concentration difference is determined by
correlation analysis of vertical velocity fluctuations and concentration variability at different depths
($w'\cdot$[TM]$'$). As these data are not available, we chose to use the mean vertical concentration
differences over a vertical distance of 10 m. Thus, the vertical advective flux $F_{az}$ at each station was
estimated from $F_{az} = w(x,z) \cdot \overline{\frac{\partial [TM]}{\partial z}} 10\ m$.
**2.8 Figures**
All figures were produced in R (version 3.4.3). Data gridding in figures 2 and 3 was performed using
the Tps function within the fields package in R (Nychka et al., 2016).
**3.   RESULTS & DISCUSSION**
**3.1 Oceanographic settings of the study area**
The cruise was conducted in June 2014 along a transect crossing a narrow shelf off the Mauritanian
coast at 18°20'N. The vertical structure of the OMZ in this region is characterized by a deep OMZ at
about 400 m depth, and a shallow OMZ at about 100 m depth (Brandt et al., 2015). Coastal upwelling
of nutrient-rich deep water occurs as a result of offshore transport of surface waters caused by a
Northeast Trade wind component parallel to the coast. While north of 20°N upwelling persists
throughout the year, upwelling south of 20°N, including the Mauritanian upwelling region, undergoes
seasonal changes in upwelling strength (Barton et al., 1998), with strongest upwelling occurring
between December and April. The seasonal variability is mainly driven by changes in wind forcing
associated with the migration of the Intertropical Convergence Zone (Schafstall et al., 2010).





The eastern boundary circulation consists of the Mauritania Current (MC, Fig. 1) flowing poleward at
the surface against the equatorward winds and of the Poleward Undercurrent (PUC) flowing in the
same direction at depths between 50 and 300 m (Barton, 1989; Klenz et al., 2018; Mittelstaedt, 1983;
Peña-Izquierdo et al., 2015). Both currents supply cold, $O_2$ and nutrient-rich waters of predominantly
South Atlantic origin (South Atlantic Central Water, SACW) to the coastal upwelling region (e.g.
Mittelstaedt, 1991; Mittelstaedt, 1983; Peña-Izquierdo et al., 2015). In response to the changing winds,
the eastern boundary circulation likewise exhibits a pronounced seasonal variability (Klenz et al.,
2018; Stramma et al., 2008a). The strongest poleward flow is observed during the relaxation period
between May and July when alongshore, upwelling-favorable winds weaken but wind stress curl is at
its maximum (Klenz et al., 2018). During the upwelling season in boreal winter, the circulation more
closely resembles the classical eastern boundary circulation regime, with a weak poleward
undercurrent flowing beneath an equatorward coastal jet. At deeper levels (300–500 m depth), flow
was found to be equatorward during both seasons. The shallow (<300 m depth) boundary circulations
turn offshore at the southern flank of the Cape Verde frontal zone (CVFZ) (e.g. Tomczak, 1981; Zenk
et al., 1991) at about 20°N, separating SACW from more saline and $O_2$-rich Central Waters formed in
the North Atlantic (NACW). The circulation in June 2014 was typical for a relaxation period
characterized by little upwelling and a strong poleward flow over the entire shelf between the surface
and 250 depth (Klenz et al., 2018).
Meridional sections of water mass properties and $O_2$ concentrations from around 18°N showed that
waters with an enhanced SACW proportion advected from the south as well as NACW coming from
the north, have higher $O_2$ concentrations than the ambient waters (Klenz et al., 2018). The mixture of
SACW and NACW waters found in the thermocline particularly during boreal winter, previously
identified as a regional water mass and termed the Cape Verde SACW (SACWcv) by Peña-Izquierdo
et al. (2015), is a signature of an older water mass with lower $O_2$ concentrations than those of SACW
or NACW due to a longer residence time and $O_2$ consumption through remineralization. Elevated
pelagic oxygen consumption levels at the Mauritanian continental margin were recently determined by
Thomsen et al. (2018). During the transition period in May through July upper Central Waters (50–
300 m depth) are dominated by SACW accounting for 80–90 % of the water masses in the boundary
current region (Klenz et al., 2018).
The SACW transported poleward within the boundary circulation is supplied by the zonal North
Equatorial Counter Current (NECC) and North Equatorial Under Current (NEUC), which flow
eastward at about 5°N (Brandt et al., 2015) before diverging into a northward and a southward flowing
branch in front of the African coast.
As a result of interactions between tidal currents, topography and critically sloping upper continental
slope topography (e.g. Eriksen, 1982), the Mauritanian upwelling region is known for elevated
nonlinear internal wave activity resulting in enhanced mixing in the water column of the upper slope



and shelf region (Schafstall et al., 2010). Vertical fluxes of nutrients driven by mixing processes are
amongst the largest reported in literature (Cyr et al., 2015).
The CTD and microstructure deployments were performed along the east-west transect in the period
June 8 to June 27 (2014) (Fig. 1). Oxygen concentrations reached a deep minimum of 40–50 µmol kg$^{-1}$
at about 400 m and a shallow minimum of 30–50 µmol kg$^{-1}$ at about 50–100 m (Fig. 2), which is in
agreement with previous studies (Brandt et al., 2015; Thomsen et al., 2018). Mixed layer depths
ranged between 10 and 22 m during the cruise. Salinity was highest at the surface (ca. 36.02) and
generally decreased with depth to a minimum of 34.71 at around 1000 m. Nitrate ($NO_3^-$)
concentrations in the surface mixed layer varied between 0.1 and 11.3 µmol L$^{-1}$ and phosphate ($PO_4^{2-}$)
between 0.15 and 0.91 µmol L$^{-1}$. $NO_3^-$ and $PO_4^{2-}$ concentrations increased with depth to a maximum of
47.6 and 3.2 µmol L$^{-1}$, respectively (Fig. 2).
Over a time period of 19 days, two trace metal stations were reoccupied along the transect at water
depths of 170 m (18.23 °N, 16.52 °W, 1$^{st}$ deployment: June 12, 2$^{nd}$ deployment: June 21) and 189–238
m (18.22°N, 16.55°N, 1$^{st}$ deployment: June 24, 2$^{nd}$ deployment: June 26). Minimum $O_2$ concentrations
of 30 µmol kg$^{-1}$ observed before June 15, which increased to 50 µmol kg$^{-1}$ after June 19 or June 24,
depending on the location. This oxygenation event that was also captured in ocean glider
measurements is discussed in detail by Thomsen et al. (2018). They attributed the change to physical
transport of SACW into the region (Thomsen et al., 2018), most likely associated with the observed
increase in current speed of the MC flowing northward parallel to the coast line and transporting
relatively $O_2$-rich water while decreasing the residence time of the SACW along the continental
margin. Additionally,  pelagic oxygen consumption was found to contribute to the variability in
oxygen concentrations close to the seafloor (Thomsen et al., 2018).
**3.2 Spatial distributions of dissolved and leachable particulate trace metals**
Dissolved Fe and LpFe concentrations ranged between 0.97–18.5 nmol L$^{-1}$ and 1.6–351 nmol L$^{-1}$,
respectively (Fig. 3a, b). Surface waters (5–29 m) had lowest dFe (0.97–4.7 nmol L$^{-1}$) and LpFe (1.6–
35.9 nmol L$^{-1}$) concentrations, whereas highest concentrations were present on the shelf close to the
seafloor (up to 18.5 nmol L$^{-1}$ dFe and 351 nmol L$^{-1}$ LpFe). Enhanced concentrations of both Fe
fractions at any given station were observed at depths with low $O_2$ concentrations (30–60 µmol $O_2$ kg$^{-}$
$^{1}$). A similar distribution pattern was observed for dCo, with concentrations between 0.069 and 0.185
nmol L$^{-1}$ (Fig. 3c). In contrast, LpCo concentrations varied between below LOQ and 0.179 nmol L$^{-1}$
and were generally highest in surface waters and close to the coast (Fig. 3d). Compared to dFe, the
concentration range of dCo was much narrower and enhanced concentrations were observed over a
broader depth range and further offshore.
Surface dFe and dCo concentrations were low, presumably due to enhanced biological uptake. No
clear increasing trend in dFe and dCo with depth was observed, indicating that processes other than, or



in addition to, remineralization influenced their distributions. Elevated concentrations were found
close to the sediments and within low $O_2$ waters. This suggested a benthic source of Fe and Co under
$O_2$-depleted conditions, and offshore transport along $O_2$-depleted water filaments, which is in
agreement with previous studies (e.g. Hatta et al., 2015; Hawco et al., 2016; Noble et al., 2012). Our
sharper onshore-offshore gradient of dFe concentrations compared to dCo in $O_2$-depleted waters
shows that oxidation and removal mechanisms/scavenging rates were faster for Fe than Co (Noble et
al., 2012). Previously reported dFe concentrations in coastal regions of the tropical North Atlantic
were lower than we observed, between 0.5–6.3 nmol $L^{-1}$ (Hatta et al., 2015; Milne et al., 2017).
However, all these samples were collected at a greater distance from the coast. In the near-coastal
Oregon and Washington shelf bottom water dFe concentrations were similar to our study under
equivalent $O_2$ concentrations (18.7–42.4 nmol $L^{-1}$ dFe, 42–61 µmol $kg^{-1}$ $O_2$; Lohan and Bruland,
2008), whereas in the euxinic waters from the Peruvian shelf region, dFe concentrations were more
than an order of magnitude higher, exceeding 200 to 300 nmol $L^{-1}$ (Schlosser et al., 2018; Scholz et al.,
2016). Similar dCo concentrations to our study were observed in the North and South Atlantic, with
highest concentrations of ~0.16 nmol $L^{-1}$ present within $O_2$-depleted waters (Noble et al., 2012; Noble
et al., 2017).
Dissolved Mn concentrations ranged between 0.46–13.8 nmol $L^{-1}$ and LpMn between below LOQ–
4.4 nmol $L^{-1}$ (Fig. 3e, f). Highest dMn and LpMn concentrations were observed in surface waters,
generally decreasing with depth. Additionally, concentrations were highest on the shelf and decreased
offshore. The dMn concentrations were generally elevated within and below the deeper $O_2$-depleted
waters with 0.70–1.34 nmol $L^{-1}$ compared to 0.46–0.91 nmol $L^{-1}$ just above. The increased dMn
concentrations within the deeper $O_2$-depleted waters (~350–500 m depth) indicate a benthic source,
similar to Fe and Co, which is in accordance with previous studies (Noble et al., 2012). However, in
the shallow $O_2$-depleted waters (~50–200 m depth), this effect is not resolvable due to high surface
concentrations, which were maintained by photo-reduction of Mn oxides to soluble Mn(II) that
prevents loss of Mn from solution (Sunda and Huntsman, 1994). Reported dMn concentrations in the
North and South Atlantic were lower than in our study, with concentrations <3.5 nmol $L^{-1}$ in surface
waters and around 0.5–1 nmol $L^{-1}$ dMn within the OMZ (Hatta et al., 2015; Noble et al., 2012). As for
dFe, these lower reported values can also be explained by sampling stations positioned at further
distance from the coast and removal of dMn via biological oxidation processes with distance from the
source (Moffett and Ho, 1996).
Dissolved Cd and Ni concentrations were lowest in surface waters with 0.022–0.032 nmol Cd $L^{-1}$ and
2.6–2.8 nmol Ni $L^{-1}$, and showed an increasing trend with depth to maximum values of 0.60 nmol $L^{-1}$
and 5.8 nmol $L^{-1}$, respectively (Fig. 3g, m). Leachable particulate Cd concentrations were between
below LOQ and 0.20 nmol $L^{-1}$, and LpNi concentrations between below LOQ and 1.7 nmol $L^{-1}$. A
large fraction of Ni (72–100%) was present in the dissolved form. The majority of LpNi samples were





below the LOQ (>70% of the data) and LpNi is therefore not included in Fig. 3. LpCd concentrations
were highest close to the coast and decreased offshore (Fig. 3h). In surface waters close to the coast
the LpCd fraction was dominant with up to 84.3% of the entire Cd pool (d + Lp). The fraction of LpCd
in surface water beyond the shelf break (including stations 2, 1 and 9) contributed still up to 54.3% of
the Cd pool, whereas below 50 m only 0–12.8% of TDCd was in the Lp phase beyond the shelf break.
In contrast to Fe, Co and Mn, no increases in Cd and Ni were observed near the seafloor and within
the $O_2$-depleted waters indicating that Cd and Ni concentrations are mainly controlled by
remineralization of sinking organic matter, which is typical for these two nutrient-like TMs (Biller and
Bruland, 2013). Similar distributions with concentrations between 0 and 1000 m water depth ranging
from ~2–5.5 and ~0–0.55 nmol $L^{-1}$ for dNi and dCd, respectively, were observed during the
GEOTRACES transect GA03_w in the tropical North Atlantic (Mawji et al., 2015; Schlitzer et al.,

12 2018).

Dissolved Cu concentrations in surface waters ranged between 0.63–0.81 nmol $L^{-1}$ (Fig. 3i).
Concentrations increased with depth to around 1.37 nmol $L^{-1}$ at 700 m depth close to the seafloor,
whereas highest observed concentrations further offshore were 0.95 nmol $L^{-1}$ at the greatest sampled
depth of 850 m. These results indicate that in addition to remineralization processes of sinking
biogenic particles, the distribution of Cu is influenced by inputs from the seafloor. This is in
accordance with previous studies, suggesting that Cu is released from continental shelf sediments
under oxic and moderately reducing conditions (Biller and Bruland, 2013; Heggie, 1982), whereas no
increase in Cu concentrations near the seafloor was observed at low bottom water $O_2$ concentrations
(O2 <10 µM; Johnson et al., 1988). A decrease in Cu concentrations in the bottom boundary layer was
also reported with a seasonal decrease in $O_2$ in summer from a minimum of 70 µM $O_2$ in May to 40
µM $O_2$ in August, suggesting a decrease in sedimentary release of Cu (Biller and Bruland, 2013). In
strongly reducing sediments and the presence of $H_2S$, Cu forms inorganic sulfides and precipitates,
which may explain reduced sedimentary Cu release under low bottom water $O_2$ concentrations (Biller
and Bruland, 2013). Therefore, the sediment source of dCu might show a different dependency on
bottom water $O_2$ concentrations than dFe, dCo and dMn explaining the distinct distribution of dCu.
Concentrations of LpCu were between below the LOQ to 0.61 nmol $L^{-1}$ with enhanced levels at station
4 close to the coast and at mid depths of the three stations furthest offshore (9, 5 and 2) (Fig. 3j).
Observed dPb concentrations were lowest in the surface waters at 9–14 pmol $L^{-1}$ and increased with
depth to 29–86 pmol $L^{-1}$ below 600 m depth (Fig. 3k). Lead is not considered a nutrient-like TM (e.g.
Boyle et al., 2014), but our observations indicate a release of Pb from sinking particles following
remineralization. The concentration range and depth distribution is similar to reported distributions
further offshore at about 21°W (Noble et al., 2015). These authors suggested that increased
concentrations of up to 70 pmol $L^{-1}$ between 600 and 800 m depth were related to the influence of
Mediterranean Outflow Waters (MOW). Additionally, increased Pb concentrations in proximity to





sediments have been attributed to the benthic release of historic Pb through reversible scavenging from
particles and the release of dPb associated with Fe/Mn oxyhydroxides during reductive dissolution of
those oxides in anoxic sediments (Rusiecka et al., 2018). The major source of Pb to the ocean is
atmospheric dust deposition from anthropogenic emissions (Bridgestock et al., 2016; Nriagu and
Pacyna, 1988; Veron et al., 1994) with a recent indication of reduced anthropogenic Pb inputs to
surface waters in the eastern tropical Atlantic under the North African dust plume (Bridgestock et al.,
2016). Low surface water concentrations on the Mauritanian shelf indicate low atmospheric inputs of
Pb to this region. LpPb was below the LOQ–27 pmol L$^{-1}$, and the distribution of LpPb was similar to
that of LpFe, with subsurface maxima within $O_2$-depleted waters (Fig. 3l) and may indicate increased
scavenging of dPb in these layers which might be associated with Fe containing particles.
In general, sediment derived TM concentrations decrease with distance from the shelf and with time
that passed since the water mass has been in contact with the sediments due to water mass mixing and
removal processes such as precipitation and scavenging (Bruland and Lohan, 2006). Radium isotopes
can be used as a tracer for benthic sources. The major source of Ra to the ocean is input from
sediments through the efflux of pore water, sediment resuspension, and submarine groundwater
discharge (Moore, 1987; Moore and Arnold, 1996; Rama and Moore, 1996). Due to the distinctive
half-lives of the different Ra isotopes (e.g. $^{224}$Ra ($t_{1/2}$ = 3.66 d) and $^{223}$Ra ($t_{1/2}$ = 11.4 d)) and their
conservative behaviour in seawater, it is possible to quantify the time that has passed since a parcel of
water was in contact with the sediments using the following equation by Moore (2000):

$$\left(\frac{A_{224}}{A_{223}}\right)_{obs} = \left(\frac{A_{224}}{A_{223}}\right)_i \frac{e^{-\lambda_{224}\tau}}{e^{-\lambda_{223}\tau}} \qquad (2)$$

solved for water mass age ($\tau$):

$$\tau = \frac{\ln\left(\frac{A_{224}}{A_{223}}\right)_{obs} - \ln\left(\frac{A_{224}}{A_{223}}\right)_i}{\lambda_{223} - \lambda_{224}} \qquad (3)$$

where $A_{224}/A_{223}$ is the activity ratio of $^{223}$Ra and $^{224}$Ra, with the subscript *obs* for the observed seawater
ratio and the subscript *i* for the initial groundwater endmember ratio, and $\lambda_{223}$ and $\lambda_{224}$ are the decay
constants in d$^{-1}$ for $^{223}$Ra and $^{224}$Ra.
Highest $^{224}$Ra$_{ex}$/$^{223}$Ra activity ratios were observed close to the seafloor (Fig. 3n). The average
$^{224}$Ra$_{ex}$/$^{223}$Ra ratio in proximity to the sediment source (< 20 m above seafloor) was 4.1 ± 0.7 and was
similar to reported ratios for shelf waters off South Carolina ($^{224}$Ra$_{ex}$/$^{223}$Ra = 4.1 ± 0.7; Moore, 2000).
The $^{224}$Ra$_{ex}$/$^{223}$Ra ratios decreased away from their benthic source due to decay ($^{224}$Ra$_{ex}$/$^{223}$Ra = 0–0.5
in surface waters). Ratios close to the seafloor were relatively constant along the transect at bottom
depths <600 m, whereas dFe, dCo and dMn concentrations varied largely in the bottom samples. This
suggests that factors, which are not influencing the Ra distribution, impacted the distributions of dFe,





dCo and dMn, with a likely influence of enhanced $O_2$ concentrations reducing sediment release or
increasing removal rates of these metals at water depths between 200 and 400 m. At around 800 m
bottom depth, $^{224}Ra_{ex}/^{223}Ra$ ratios were slightly elevated and coincided with increased dCo, dFe, dMn
and dCu concentrations despite $O_2$ concentrations >70 µmol $kg^{-1}$. This suggests that the enhanced TM
concentrations at this location were influenced by a strong sediment source which may be related to
the presence of a benthic nepheloid layer as indicated by an increase in turbidity in proximity to the
seafloor. An elevated $^{224}Ra_{ex}/^{223}Ra$ ratio of 3.5 ± 0.6 was observed at about 16.65°N and 80 m water
depth (bottom depth 782 m) and coincided with a local maximum of dFe, dMn and dCo and reduced
$O_2$ concentrations. These observations indicate that the waters with the local maximum of dFe, dMn
and dCo have been in relatively recent contact (12–20 days assuming initial pore water $^{224}Ra_{ex}/^{223}Ra$
ratios between 18–38; Moore, 2007) with sediments, likely originated from south of our transect as a
result of a strong poleward flow (Klenz et al., 2018), and that the dynamic current system in this
region can cause local and short-term variability in the transport of sediment derived TMs.
**3.3 Classification of different groups of trace metals based on principal component analysis**
Principal Component Analysis (PCA) was performed (using the RDA function within the vegan
package in R; Oksanen et al., 2017) to investigate different groups and correlations in the data set.
Dissolved TMs (Fe, Mn, Co, Ni, Pb, Cu and Cd), nutrients (silicic acid, nitrate and phosphate),
dissolved $O_2$, Apparent Oxygen Utilization (AOU), depth and iodide concentrations were utilized in
the PCA. Radium data were not included in the PCA, as the number of available data points for
$^{224}Ra_{ex}/^{223}Ra$ was much lower than for the other parameters. Surface waters shallower than 50 m were
excluded from the PCA to remove the influence of local processes in surface waters, such as localized
atmospheric deposition and photochemical processes, which in particular influence Mn and iodide
distributions. The PCA generated three principal components (PC) with eigenvalues larger than 1,
with PC1 explaining 53.6% and PC2 25.5% of the total variance in the dataset (together 79.1%).
Inclusion of PC3 in the analysis explained only 6.8% more of the variance.
The first PC group is formed by dCd, dCu, dNi and dPb (Fig. 4), which are associated with depth,
AOU, nitrate and phosphate. This indicates that the distribution of Cd, Cu, Ni, and potentially Pb, are
controlled by organic matter remineralization processes. This is in agreement with strong Pearson
correlations R >0.9 for the relationships of dCd and dNi with depth, nitrate and silicic acid
(Supplementary Material, Table S1). Weaker correlations with major nutrients were observed for dPb
(R >0.6) and dCu (R >0.4), potentially due to additional remineralization or removal mechanisms for
these elements (e.g. prior atmospheric inputs and water mass transport, Pb; sediments, Cu and Pb, and
scavenging). The second group of TMs is composed of dFe, dCo and dMn that are associated with
elevated iodide and turbidity, and low dissolved $O_2$ (Fig. 4). Iodide ($I^-$) is the reduced form of iodine
(I), which is typically present as iodate ($IO_3^-$) in oxygenated subsurface water. Both I forms are present
as soluble anions in seawater. Due to a relatively high redox potential (pE ~10), iodine is one of the





first redox-sensitive elements to undergo reduction under suboxic conditions and is therefore a useful
indicator for active reductive processes (Rue et al., 1997). Despite their role as micronutrients, Fe, Mn
and Co do not correlate with nutrients indicating that processes other than remineralization controlled
their distributions.
The anti-correlation with $O_2$ (also shown in Fig. S1) and correlation with iodide support the notion that
Fe, Co and Mn distributions were strongly influenced by water column $O_2$ concentrations, presumably
through: (i) enhanced benthic metal fluxes from anoxic sediments, and (ii) decreased oxidation rates in
the overlying water column under $O_2$-depleted conditions. This is also supported by elevated benthic
Fe(II) fluxes observed at the seafloor within the shallow OMZ, with benthic fluxes of 15–27 µmol m$^{-2}$
d$^{-1}$ (Schroller-Lomnitz et al., 2018).
Variability in the redox-sensitive metals, Fe, Mn and Co, were not fully explained by either $O_2$ or
iodide concentrations; Pearson correlations with $O_2$ were -0.55, -0.61 and -0.58, respectively
(Supplementary Material, Table S1). As shown before, other factors such as, for example, water mass
mixing and age, the amount and type of particles present, and remineralization all likely impact their
dissolved concentrations. Consequently, such a complex chain of factors and processes means that one
variable alone is unlikely to explain the behaviour of Fe, Mn, and Co.
**3.4 Influence of the different sources of Fe, Mn and Co**
The main sources of TMs in our study region are sedimentary release and atmospheric dust deposition
(e.g. Rijkenberg et al., 2012). Also release of TMs via organic matter remineralization may have an
important influence on the distribution of TMs. In the following, we discuss the relative influence of
remineralization, atmospheric dust deposition and sedimentary release on the supply of Fe, Co and Mn
to surface waters.
*3.4.1   Remineralization*
To quantify the influence of remineralization for dFe, we employed dFe to carbon (dFe/C) ratios
(carbon was calculated using AOU, with an AOU/carbon ratio of 1.6; Martin et al., 1989). Surface
data, where $O_2$ was over-saturated (due to biological $O_2$ production), were excluded. Dissolved Fe/C
ratios for the entire transect varied between 15 and 74 µmol mol$^{-1}$. These results agree with those for
shelf-influenced waters with dFe/C ratios of 13.3–40.6 µmol mol$^{-1}$ further south at 12°N (Milne et al.,
2017). Reported ratios for the North Atlantic, further away from the shelf were lower and ranged
between 4 and 12.4 µmol mol$^{-1}$ (Fitzsimmons et al., 2013; Milne et al., 2017; Rijkenberg et al., 2014).
To estimate the amount of dFe being derived by remineralization, we assume a dFe/C ratio of 4–12
µmol mol$^{-1}$ from organic matter remineralization, similar to the observed dFe/C ratios in the open
ocean close to our study area without a strong shelf influence. These offshore ratios may still be
influenced by an atmospheric source of dFe, which would result in an overestimation of dFe/C ratios



from remineralization and thereby an overestimation of the fraction of remineralized dFe. Apart from
additional inputs, the dFe/C ratios are influenced by the respective Fe/C stoichiometry in the sinking
organic matter and removal of dFe by scavenging. Furthermore, it is not clear if the offshore ratios can
be transferred to a location close to the coast, as the balance between remineralization and scavenging
processes might be different due to differences in phytoplankton productivity and particle load. Hence,
this approach only provides a broad estimate of the relative influence of remineralization on the
distribution of dFe in the study area.
We obtain a range between $5 \pm 3\%$ and $54 \pm 27\%$ for dFe being derived from remineralization
processes with lowest values observed on the shelf at 34 m depth at station 4 ($5 \pm 3\%$) and highest
values estimated beyond the shelf break at Stn 9 at 213 m depth ($54 \pm 27\%$) and Stn 2 at 450 m depth
($52 \pm 26\%$). However, no clear increase in the contribution of remineralized dFe to total dFe with
depth or distance to the coast was observed. For example at depths between 35 and 200 m, our
estimates of dFe from remineralization ranged between $10 \pm 5\%$ and $51 \pm 25\%$ with high values of up
to $41 \pm 20\%$ at 50 m depth at station 7 close to the coast, whereas relatively low values of $19 \pm 9\%$
were observed at 89 m at station 2. These results indicate that, locally, remineralization can be an
important control on dFe concentrations, but that the contribution varies largely with additional
important controls, often dominating over remineralization.
Similar analysis for dCo/C ratios revealed an increased importance of an additional source close to the
shelf. Observed dCo/C ratios ranged between 0.81 and 2.2 µmol mol$^{-1}$. The larger ratios were
observed close to the coast and decreased further offshore. Overall, the observed ratios were somewhat
higher than reported cellular ratios of phytoplankton in the North Atlantic of 0.5–1.4 µmol mol$^{-1}$
(Twining et al., 2015). However, relatively constant dCo/C ratios beyond the shelf break (dCo/C:
0.82–1.09 µmol mol$^{-1}$, stations 2, 5 and 9) that are similar to cellular ratios of phytoplankton suggest a
large influence of remineralization on dCo beyond the shelf break, whereas enhanced ratios close to
the coast suggest an additional benthic source. Due to the lack of comparable data of offshore dCo/C
ratios and the multiple processes influencing this ratio (varying phytoplankton nutrient stoichiometry
and scavenging), we did not use these values to estimate the remineralized dCo fraction.
The distribution of Mn was not predominantly determined by biological uptake and remineralization
processes in our study region. In contrast, dMn/C ratios were largely influenced by photoreduction in
the surface (Sunda and Huntsman, 1994), removal via biotic oxidation and formation of Mn oxides at
depth (Tebo et al., 2004). Therefore, we did not assess remineralization processes for Mn using dMn/C
ratios.
*3.4.2   Atmospheric deposition*
Aluminum is present as a relatively constant fraction of ~8.15 wt% in the continental crust (Rudnick
and Gao, 2006), is supplied to open ocean surface waters mainly by atmospheric deposition (Orians





and Bruland, 1986) and is considered not to be taken up by phytoplankton (apart from a small amount
being incorporated into siliceous diatom frustules; Gehlen et al., 2002). Therefore, dAl in the surface
mixed layer is used as a tracer for atmospheric deposition to the surface ocean (Measures and Brown,
1996; Measures and Vink, 2000). The atmospheric input in the study region is mainly influenced by
North African/Saharan mineral dust with only a small contribution of anthropogenic sources which
differ greatly in TM composition and solubilities from mineral dust (Baker et al., 2013; Patey et al.,
2015; Shelley et al., 2015). Close to continental shelves, in addition to atmospheric input, Al can also
be supplied by sediment resuspension (Menzel Barraqueta et al., 2018; Middag et al., 2012; Moran
and Moore, 1991).
Our dAl concentrations in surface water ranged between 30 and 49 nmol $L^{-1}$ and LpAl between 3.4
and 18.2 nmol $L^{-1}$. Dissolved Al concentrations decreased with depth (Fig. 8), indicating that Al was
released by aeolian dust deposition to surface waters and removed through scavenging at depth
(Orians and Bruland, 1985). Trace metal (Fe, Co, and Mn) to Al ratios were utilized to investigate the
influence of atmospheric dust deposition. We present molar ratios for dissolved (dTM/dAl), total
dissolvable (TDTM/TDAl) and leachable particulate (LpTM/LpAl) concentrations. In the surface
mixed layer, dFe/dAl molar ratios ranged between 0.019 and 0.114, TDFe/TDAl between 0.236 and
0.826 and LpFe/LpAl between 1.04 and 9.50.
Literature particulate Fe/Al ratios from aerosol samples collected in the remote North Atlantic
between 8.7°N and 23°N were in the range of 0.31 ± 0.06 (Buck et al., 2010; Patey et al., 2015) and
0.37 ± 0.02 in the North East Atlantic ~18°N under the Saharan dust plume (Shelley et al., 2015). In
contrast, upper crustal material ratios are lower ranging from 0.19 to 0.23 suggesting a slight Fe
enrichment of aeolian mineral dust particles (McLennan, 2001; Rudnick and Gao, 2006; Wedepohl,
1995). Lower Fe than Al solubilities from aerosol leach experiments in ultra-high purity water (UHP)
and 25% acetic acid (HAc) and seawater have been reported (Baker et al., 2006; Buck et al., 2010;
Shelley et al., 2018), but soluble Fe/Al ratios from these experiments varied dependant on the leach
medium (UHP: 0.21 ± 0.04, 25% HAc: 0.25 ± 0.04, seawater: 0.051 ± 0.009; Shelley et al., 2018).
This indicates that dFe/dAl and LpFe/LpAl ratios in seawater from atmospheric deposition are likely
to be lower than particulate ratios of digested aerosol samples in the study region.
Our dFe/dAl ratios at the upper end (dFe/dAl: 0.114) are larger than aerosol leaches in seawater
indicating a potential additional input of dFe, whereas our lower dFe/dAl ratios than reported ratios in
aerosol leaches suggest removal of dFe by biological uptake or scavenging. Our LpFe/LpAl are all
larger than reported ratios in aerosol leaches and total aerosol ratios, which shows that there is an
additional source of LpFe or transfer of sediment-derived dFe onto the particulate phase by biological
uptake or sorption to particles. Total dissolvable ratios comprise both dissolved and leachable
particulate phases, thereby being independent of the phase transfer from dissolved to particulate phase
(via biological uptake or sorption). The lower end of total dissolvable ratios (TDFe/TDAl: 0.236) were





close to the total ratios in aerosol samples, suggesting that atmospheric deposition represented an
important source of Fe and Al to the surface ocean. At the upper end, ratios were much larger
(TDFe/TDAl: 0.826) than aerosol ratios and indicate an additional benthic source of Fe.
These interpretations only apply, however, if residence times of dissolved and particulate Fe and Al
phases supplied via atmospheric deposition are similar. This is difficult to assess, as estimated
residence times for both elements are dependent on input and removal rates and vary largely between
locations. Overall, our Fe/Al ratios suggest that atmospheric deposition is an important source of Fe to
surface waters with an additional contribution of benthic inputs. However, uncertainties in solubilities
and residence times cause a high uncertainty in the interpretation of the role of atmospheric deposition.
Observed dCo/dAl ratios in the upper 50 m were 0.001–0.004, TDCo/TDAl ratios were slightly higher
at 0.003–0.005 and LpCo/LpAl ratios were 0.006–0.020. Cobalt is present in the upper continental
crust in a much smaller molar fraction than Fe (Co/Al: 0.000071–0.000097; McLennan, 2001;
Rudnick and Gao, 2006; Wedepohl, 1995). However, ratios in aerosol samples under the North
African dust plume were slightly higher (Co/Al: $0.00016 \pm 0.00002$; Shelley et al., 2015) than crustal
ratios  and solubility of Co from these aerosol samples was much higher than Al solubility resulting in
soluble Co/Al of $0.0021 \pm 0.0009$ in UHP (Shelley et al., 2018). The soluble ratios also varied largely
depending on the leach medium and might therefore also vary from the actual aerosol solubility in
seawater at our study site. Our ratios of all fractions were larger than total aerosol ratios and mostly
higher than soluble ratios from aerosol leaches. This indicates that an additional benthic source of Co
likely contributed to the Co present in surface waters.
Dissolved Mn/dAl ratios in the upper 50 m ranged between 0.082 and 0.347, and TDMn/TDAl
between 0.083 and 0.256. The ratios are much larger than upper crustal ratios (Mn/Al: 0.0032–0.0037;
McLennan, 2001; Rudnick and Gao, 2006; Wedepohl, 1995) but similar to the soluble ratios of Mn/Al
from aerosols in UHP (Mn/Al: $0.24 \pm 0.09$; Shelley et al., 2018) indicating that a large amount of Mn
may be derived from atmospheric deposition. However, these ratios are heavily overprinted by the
long residence time of Mn in surface waters due to photoreduction. Therefore, it is not possible to
reliably estimate the contribution of atmospheric Mn deposition based on the Al data.
Atmospheric dFe fluxes were calculated using the dAl inventory in the surface mixed layer, a
residence time of dAl of $0.65 \pm 0.45$ years as reported for the Canary Current System (Dammshäuser
et al., 2011), and a ratio of 0.31 for dust derived dissolved Fe/Al (Buck et al., 2010). This approach
assumes that dAl is only supplied to the surface ocean via atmospheric deposition. Vertical fluxes of
Al from sediment resuspension are unlikely to largely contribute to concentrations of dAl in surface
waters here as dAl concentrations were decreasing with depth. Mean atmospheric dFe fluxes of the
individual stations were 0.63–1.43 µmol m$^{-2}$ d$^{-1}$ (Fig. 5, Supplementary Table S2), values similar to
reported fluxes close to our study region of 2.12 µmol m$^{-2}$ d$^{-1}$ further north between 22.5–25°N and





26.5–27.5°W (Rijkenberg et al., 2012) and 0.120 nmol m$^{-2}$ d$^{-1}$ around 20°N close to the African coast
(Ussher et al., 2013). The uncertainty in the residence time of dAl, however, creates a large
uncertainty in calculated fluxes resulting in a lowest flux of 0.37 µmol m$^{-2}$ d$^{-1}$ when using the largest
estimated residence time of 1.1 years and a highest flux of 4.65 µmol m$^{-2}$ d$^{-1}$ when using the shortest
estimated residence time of 0.2 years.
*3.4.3    Vertical trace element fluxes to surface waters*
The vertical fluxes (diffusive and advective) of dFe from the top of the shallow O$_2$-depleted waters
(between 23 and 89 m depending on station) into surface waters were determined to assess the
potential Fe contribution to phytoplankton growth. A detailed summary of calculated fluxes, the
contribution of diffusive and advective term and uncertainties for dFe for all stations is given in
Supplementary Information Table S2. Closest to the shelf (bottom depth: 50 m) mean dFe fluxes were
13.5 µmol m$^{-2}$ d$^{-1}$. Further offshore, vertical dFe fluxes decreased to 0.16 µmol m$^{-2}$ d$^{-1}$ (station 2,
bottom depth: 1136 m, 77 km offshore) (Fig. 5). However at station 5 higher dFe fluxes were observed
(dFe: 1.3 µmol m$^{-2}$ d$^{-1}$) than at stations 9 (closer to the shelf) and 2 (further offshore). At station 5,
eddy diffusivity was determined from only 5 microstructure profiles and was unusual high at this
station. Therefore, the enhanced vertical fluxes are likely caused by a rare elevated mixing event and
do not represent a long-term average. Between repeat stations 3A and 3B, mean fluxes decreased from
2.3 (Stn 3A) to 1.35 µmol m$^{-2}$ d$^{-1}$ (Stn 3B), which was partly caused by a difference in the vertical
concentration gradient of dFe and partly by a change in diffusivity.
On the shelf (station 4, bottom depth: 45 m), dFe fluxes were dominated by vertical advective rather
than diffusive fluxes due to the strong upwelling velocity on the shelf (Table S2). At the continental
slope stations (stations 3, 7 and 8, bottom depth: 90–400 m), fluxes were dominated by high diffusive
fluxes, which were around 3 times larger than the advective flux term. Further offshore (stations 2 and
9, bottom depth: >400 m) the contribution of advective and diffusive fluxes were similarly low except
for station 5 with particularly strong vertical mixing. Similar vertical dFe (16 µmol m$^{-2}$ d$^{-1}$) to the
upper water column were reported on the shelf at 12°N (Milne et al., 2017). Although, in the study
region atmospheric fluxes of dFe were enhanced relative to global averages (Mahowald et al., 2009)
with mean fluxes of 0.63–1.43 µmol m$^{-2}$ d$^{-1}$, our vertical dissolved Fe fluxes from the shallow O$_2$
depleted waters of 0.95–13.5 µmol m$^{-2}$ d$^{-1}$ exceeded atmospheric fluxes at all stations apart from
station 2 (0.16 µmol m$^{-2}$ d$^{-1}$) furthest offshore and potentially station 9 (0.08 µmol m$^{-2}$ d$^{-1}$), where no
atmospheric fluxes were determined. The weaker influence of atmospheric deposition in this region
close to the coast is in accordance with previous studies that demonstrated sediments to be the major
contributor to the Fe inventory in the coastal region of the eastern tropical Atlantic, whereas the
importance of atmospheric inputs increases further offshore (Milne et al., 2017). Our vertical
advective fluxes are likely lower than the annual average and also lower than usually during the
relaxation period as upwelling favourable winds were particularly low in June 2014.



Dissolved Co fluxes ranged between 2 and 113 nmol m$^{-2}$ d$^{-1}$. These values are lower than reported
upwelling fluxes of dCo of 250 nmol m$^{-2}$ d$^{-1}$ for this region (Noble et al., 2017), but are larger than
atmospheric deposition fluxes of 1.7 nmol m$^{-2}$ d$^{-1}$ (Shelley et al., 2015). Fluxes of dMn are downwards
from surface waters to O$_2$ depleted waters due to higher concentrations in surface waters.
**3.5 Removal mechanisms and particle interactions**
In the top 50 m of the water column a large part of the LpTMs may be part of living biological cells
(e.g. phytoplankton) or organic detritus. Additionally, LpTMs may be part of lithogenic phases from
Saharan dust and sediment particles, or authigenic phases. Authigenic phases are formed in-situ by
TM adsorption onto particle surfaces or by the formation of amorphous TM oxides and hydroxides
(e.g. FeO(OH) in the mineral structure of goethite) (Sherrell and Boyle, 1992).
Iron was mainly present in the size fraction >0.2 µm with TDFe concentrations being 0.44–44.5 times
higher than dFe (<0.2 µm) (Fig. 6a). To investigate the influence of particle load on the distribution
between dissolved and particulate phases, the fraction of Lp (Lp/TD) TMs and Lp concentrations are
plotted against turbidity for Fe, Co and Mn (Fig. 6b, c). A low fraction of LpFe of around 60% was
observed at lowest turbidity. As turbidity increases from 0.1 to 0.2 NTU, the LpFe fraction increased
to >90%. This suggests that the fraction of LpFe is tightly coupled to the particle load. Iron adsorption
onto particles has been demonstrated to be reversible with a constant exchange between dissolved and
particulate fractions (Abadie et al., 2017; Fitzsimmons et al., 2017; John and Adkins, 2012; Labatut et
al., 2014). Furthermore, offshore transport of acid-labile Fe particles originating from reductive
dissolution processes from continental shelf sources was observed in the North Pacific (Lam and
Bishop, 2008) and may contribute to the bioavailable Fe pool. Therefore an important fraction of Fe
may be transported offshore adsorbed to particles and can enter the dissolved pool by cycling between
dissolved and particulate phases.
Manganese and Co mainly occurred in the dissolved form. The LpCo fraction ranged between 0 and
75%, and the fraction and concentration of LpCo, showed linear increases with turbidity, indicating an
influence of particle load on Co size fractionation, similar to Fe. In contrast to Fe and Co, the fraction
of LpMn varied between 3 and 40%, and did not show a correlation with turbidity, whereas LpMn
concentrations showed an increase with turbidity. This indicates that an increased presence of particles
coincided with enhanced LpMn levels, but that the particle load did not substantially influence the
distribution between dMn and LpMn phases and that particles did not contribute to the dMn fraction.
This suggests that particles did not play a major role in transport of dMn, which agrees with a study on
hydrothermal vent plumes, where the distribution of the dMn plume was decoupled from the
distribution of the particulate Mn plume (Fitzsimmons et al., 2017).
The increase in LpFe concentrations with increasing turbidity was weaker in the surface waters
compared to water depths below 50 m (Fig. 6c). This suggests a large additional LpFe source at depth



with either a higher Fe content of particles or the presence of different sizes of particles causing
different responses in turbidity measurements. The large additional LpFe source at depths is likely
associated with benthic dFe inputs, with a subsequent transfer to the particulate phase by adsorption or
oxidation with subsequent formation of Fe(oxihydr)oxides. Enhanced turbidity at depth may also
indicate sediment resuspension, which would result in the release of TM-containing particles from
sediments and enhanced release of dTMs from sediment pore water. The effect of sediment
resuspension is discussed in more detail below (section 3.6.2).
In contrast to Fe, the increase in LpCo and LpMn concentrations with turbidity was similar in surface
waters and below and suggests less variability in the composition of the particulate Co and Mn phase
throughout the water column with a potentially weaker influence of sediment release on the
distribution of particulate Mn and Co. A weaker influence of sediment release might be influenced by
a weaker release of Co and Mn from sediments in the dissolved form and slower oxidation rates
compared to Fe, in particular for Co (Noble et al., 2012), resulting in a slower conversion into the
particulate phase. Such an interpretation based on turbidity data alone, however, is very hypothetical
and would require further investigation of particulate TM species composition in this area.
**3.6 Temporal variability in redox-sensitive trace metals**
Large temporal changes in $O_2$, turbidity and redox-sensitive TMs were observed within a short time
scale of a few days at two repeat stations, station 3A/3B and station 8A/8B (Fig. 7).
Station 3 and 8 were sampled twice with a period of nine days between both deployments for station 3
(Fig. 7a) and two days for station 8 (Fig. 7b). At station 3, $O_2$ concentrations in the upper 50 m were
very similar between both deployments, whereas below 50 m $O_2$ increased from 30 µmol kg$^{-1}$ during
the first deployment to 50 µmol kg$^{-1}$ nine days later. At the same time, turbidity below 50 m had
decreased from 0.35 to below 0.2, and dFe concentrations from a maximum of 10 nmol L$^{-1}$ to 5 nmol
L$^{-1}$ nine days later. In addition, dMn and dCo concentrations decreased from 5 to 3 nmol L$^{-1}$ and 0.14
to 0.12 nmol L$^{-1}$, respectively. Particularly large changes were also observed for LpTM concentrations
with a decrease from 147–322 nmol L$^{-1}$ to 31–51 nmol L$^{-1}$ for LpFe, from 0.066–0.114 nmol L$^{-1}$ to
0.015–0.031 nmol L$^{-1}$ for LpCo and from 1.24–2.64 to 0.16–0.54 for LpMn. In contrast, no changes in
water mass properties (T/S) occurred below 50 m (Fig. 7a).
Similar changes in $O_2$ and turbidity were observed at station 8. During the first deployment a local
minimum in $O_2$ below 30 µmol kg$^{-1}$ was present between 105 m and 120 m water depths which
coincided with a maximum in turbidity of 0.4 (Fig. 7b). In contrast $O_2$ concentrations and turbidity
during the second deployment were relatively constant (50–60 µmol kg$^{-1}$ $O_2$ and turbidity 0.2) below
50 m. At depth of the local $O_2$ minimum and turbidity maximum, concentrations of dFe, dMn and dCo
were elevated during the first deployment with concentrations of 9.4 ± 2.1 nmol dFe L$^{-1}$, 3.7 ± 0.6



nmol dMn L$^{-1}$ and $0.145 \pm 0.033$ nmol dCo L$^{-1}$ in comparison to $4.6 \pm 1.0$ nmol dFe L$^{-1}$, $2.6 \pm 0.5$
nmol dMn L$^{-1}$, and $0.122 \pm 0.028$ nmol dCo L$^{-1}$ at similar depth during the second deployment.

3       *3.6.1   Remineralization*

We compared the results of the redox-sensitive TMs to other nutrient-like TMs and PO$_4$. For both
repeat stations only small changes in dCd (Stn 3A: 0.107–0.231 nmol L$^{-1}$; Stn 3B: 0.135–0.150 nmol
L$^{-1}$) and PO$_4$ (Stn 3A: 1.59–1.85 µmol L$^{-1}$; Stn 3B: 1.55–1.71 µmol L$^{-1}$) concentrations were observed
below 50 m (Fig. 8), suggesting that only a small fraction of dFe under lower O$_2$ conditions was
supplied by more intense remineralization of biogenic particles in the water column.
A weak influence of remineralization processes on the variability in dFe concentrations was confirmed
by substantially higher dFe/C ratios at lower O$_2$ concentrations (40–72 µmol mol$^{-1}$ at Stn 3A compared
to 33–41 µmol mol$^{-1}$ at Stn 3B, both below 50 m water depth). Assuming a dFe/C ratio of around 12
(see section 3.4.1) from remineralization, only about 0.25 nmol L$^{-1}$ of the difference in dFe
concentrations between repeated deployments can be explained by the difference in remineralization,
suggesting that most of the difference in dFe between deployments was caused by changes in source
inputs, such as enhanced sediment release during lower bottom water O$_2$ concentrations, or slower
removal by oxidation under lower O$_2$ conditions.
In contrast, dCo/C ratios were similar between repeat deployments within the OMZ (0.90–1.04 at Stn
3A and 0.92–1.06 µmol mol$^{-1}$ at Stn 3B). Thus, changes in remineralization could be a reason for the
changes in observed dCo concentrations during repeated deployments, indicating that the sensitivity of
dCo sediment input or change in oxidation rates is low at an O$_2$ shift from 30 to 50 µmol kg$^{-1}$.
Similar to Fe, higher dMn/C ratios were observed at lower O$_2$ concentrations (3.4–5.5 µmol mol$^{-1}$ at
Stn 3A compared to 2.1–2.9 µmol mol$^{-1}$ at Stn 3B). These results indicate that other processes than
remineralization are also important for the change in dMn concentrations. An additional factor
compared to Fe, might involve changes in intensity of photoreduction which may be influenced by
differences in surface turbidity observed at station 3 (lower dMn/C and higher surface turbidity during
second deployment). This, however, cannot explain the changes in dMn/C at station 8, where a higher
surface turbidity coincided with a higher dMn/C ratio at the local minimum in O$_2$.
*3.6.2   Atmospheric dust deposition and sediment resuspension*
Within the OMZ at station 3 and 8, dAl concentrations ranged between 10 and 15 nmol L$^{-1}$, and LpAl
concentration between 1.2 and 11.1 nmol L$^{-1}$ and no substantial changes were observed between
deployments (Fig. 8). As lithogenic material has a high Al content, no substantial changes in Al
concentrations signify that lithogenic inputs did not differ much between the deployments.
Consequently neither increased atmospheric input, nor sediment resuspension are likely to explain the
differences in turbidity and redox-sensitive TM concentrations. Hence, changes in turbidity may





mainly have been caused by biogenic particles, such as resuspended organic matter (Thomsen et al.,
2018). This finding can be confirmed by substantial changes in TM/Al ratios observed during the
deployments (Table 2 and Fig. S2). The Fe/Al ratios in the solid phase of underlying sediments during
the cruise were 0.23–0.30 (Schroller-Lomnitz et al., 2018) with Mn/Al ratios of 0.0015–0.0020
(Schroller-Lomnitz, pers. com.). Slight increases in LpAl towards the sediment indicate some
influence of sediment resuspension on the TM distribution. Overall much higher TM/Al ratios
compared to ratios in the sediments and aerosol samples from this region (Fe/Al: $0.37 \pm 0.02$, Co/Al:
$0.00016 \pm 0.00002$, Mn/Al: $0.0061 \pm 0.0002$; Shelley et al., 2015), suggest a large additional source of
Fe, Co and Mn in the OMZ close to the shelf. This again points towards a large influence of benthic
release of Fe, Co and Mn from sediment pore waters and subsequent partial adsorption to particle
surfaces.
*3.6.3    Other possible causes for TM variability*
From the comparison above, we can conclude that the variations in Fe concentrations during repeated
deployments were not caused by increased remineralization or changes in lithogenic inputs from
atmospheric deposition or sediment resuspension. The large changes in the Lp fractions must therefore
be of biogenic or authigenic origin. If all LpCo would be present in biogenic particles of suspended
phytoplankton cells, at our observed maximum of 0.114 nmol $L^{-1}$ LpCo at station 3A we would expect
around 4.6 nmol $L^{-1}$ LpFe in sinking phytoplankton, using an average Fe/Co ratio in phytoplankton of
40 (Moore et al., 2013) (observed ratios close to our study area were 20–40; Twining et al., 2015).
However, LpFe concentrations were 322 nmol $L^{-1}$ and thereby 70 times larger than our estimate in
biogenic particles (4.6 nmol $L^{-1}$), revealing that the majority of LpFe must be authigenically formed.
Altogether our results suggest that changes in particle load as indicated by changes in turbidity do not
comprise a major source of dFe, moreover a sink of previously dissolved Fe. Therefore, higher
dissolved and Lp concentrations during the first deployment with lower $O_2$ concentrations must be
caused by a stronger benthic source of dissolved Fe.
It is not possible to extract from our data whether the stronger benthic source under low $O_2$ conditions
is directly driven by lower $O_2$ concentrations in surface sediments and in the water column resulting in
higher benthic Fe fluxes and slower oxidation rates in the water column, or by a longer residence time
of the water mass on the shelf. However, increased benthic fluxes are in accordance with previous
findings from ex-situ sediment incubation experiments, where Fe fluxes increased with decreasing $O_2$
concentrations (Homoky et al., 2012). Therefore, we hypothesize that with a reduction of bottom water
$O_2$ concentrations from 50 to 30 µmol $kg^{-1}$, drastically more Fe is effectively released from the
sediments by diminished oxidation rates at the sediment-water interface, and that a large fraction gets
directly adsorbed onto particles. Therefore, particles do not compose a major source of Fe here, but
may play an important role in Fe offshore transport.



Due to much lower changes in concentrations of dissolved and LpCo, and the additional effect of
photoreduction and strong scavenging for Mn, we were unable to resolve the main mechanisms for
changes in Co and Mn concentrations with changes in $O_2$ and turbidity. Nevertheless, due to their
similar redox-sensitive behavior and distribution in OMZs, it is likely that they are also affected by
reduced $O_2$ conditions. The magnitude of response however, is much lower.
**4. CONCLUSION**
Sediments are an important source of Fe, Co and Mn to OMZ waters in the Mauritanian shelf region.
Remineralization and atmospheric deposition appear less important than benthic sources for dFe, with
vertical fluxes exceeding atmospheric fluxes but gaining importance with distance from shelf. We
showed that changes in $O_2$ concentrations from 30 to 50 µmol kg$^{-1}$ had a substantial influence on
dissolved and LpFe concentrations and to a lesser extent on Co and Mn concentrations by decreasing
the sediment source strength. The presence of a large part of sediment-derived Fe in the leachable
particulate phase highlights the importance of offshore particle transport on the Fe inventory,
including the dissolved form by reversible scavenging. To our knowledge, this is the first field study
that demonstrated strong short-term variability in redox-sensitive TMs over a few days to be directly
linked to changes in $O_2$. These findings demonstrate that projected long-term changes in oceanic $O_2$
concentrations will impact biogeochemical cycles and have important implications for global TM
distributions and their process parameterisations in biogeochemical models. Current models do not
account for small changes in $O_2$ on TM distributions and benthic TM fluxes. Determining the
processes involved and quantifying the effect of $O_2$ will be crucial for the implementation into current
modeling approaches. Not all processes could be resolved in this study, including the influence of the
residence time of the water masses on the shelf compared to the direct influence of $O_2$, and it is
unclear whether the changes observed on a small scale are readily transferable to a global scale.
Therefore, we suggest further investigations on short-term variability of $O_2$ and particle load in the
Mauritanian and other dynamic OMZs including water column TM measurement in combination with
benthic TM fluxes and more detailed analysis of amount and types/composition of present particles.
*Data availability*. The CTD sensor and nutrient bottle data are freely available at
https://doi.pangaea.de/10.1594/PANGAEA.860480 and
https://doi.pangaea.de/10.1594/PANGAEA.885109 respectively. According to the SFB754 data policy
(https://www.sfb754.de/de/data, all remaining data (trace metal data set) associated with this
manuscript will be published at PANGEA (www.pangea.de, search projects:sfb754) upon publication
of this manuscript.
*Author contributions*. IR analyzed the trace metal concentrations and drafted the manuscript. EPA and
MG designed the project and CS carried out the trace metal sampling at sea. J-LMB oversaw, and BW





carried out, the aluminium sample analysis. MD carried out the microstructure measurements at sea,
oversaw the calculation of the vertical flux estimates and contributed to the writing of the manuscript.
JL carried out the processing of microstructure data and calculation of the eddy diffusivity. JS, BG and
PR carried out the radium isotope analysis and their interpretation. IR and MG oversaw, and FW
carried out, the iodide analysis. All co-authors commented on the manuscript.
*Competing interests*. The authors declare that they have no conflict of interest.
*Acknowledgements*. The authors would like to thank the captain and the crew from RV Meteor and
chief scientist Dr. Stefan Sommer from the M107 cruise. This work was funded by the Deutsche
Forschungsgemeinschaft as part of Sonderforschungsbereich (SFB) 754: 'Climate-Biogeochemistry
Interactions in the Tropical Ocean'. Fabian Wolf is thanked for carrying out the analysis of iodide and
Peter Streu for help with the general lab work.

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





| 1 | and Abadie, C. and Abouchami, W. and Achterberg, E. P. and Agather, A. and Aguliar-Islas, |
|---|---|
| 2 | A. and van Aken, H. M. and Andersen, M. and Archer, C. and Auro, M. and de Baar, H. J. and |
| 3 | Baars, O. and Baker, A. R. and Bakker, K. and Basak, C. and Baskaran, M. and Bates, N. R. |
| 4 | and Bauch, D. and van Beek, P. and Behrens, M. K. and Black, E. and Bluhm, K. and Bopp, |
| 5 | L. and Bouman, H. and Bowman, K. and Bown, J. and Boyd, P. and Boye, M. and Boyle, E. |
| 6 | A. and Branellec, P. and Bridgestock, L. and Brissebrat, G. and Browning, T. and Bruland, K. |
| 7 | W. and Brumsack, H.-J. and Brzezinski, M. and Buck, C. S. and Buck, K. N. and Buesseler, |
| 8 | K. and Bull, A. and Butler, E. and Cai, P. and Mor, P. C. and Cardinal, D. and Carlson, C. and |
| 9 | Carrasco, G. and Casacuberta, N. and Casciotti, K. L. and Castrillejo, M. and Chamizo, E. and |
| 10 | Chance, R. and Charette, M. A. and Chaves, J. E. and Cheng, H. and Chever, F. and Christl, |
| 11 | M. and Church, T. M. and Closset, I. and Colman, A. and Conway, T. M. and Cossa, D. and |
| 12 | Croot, P. and Cullen, J. T. and Cutter, G. A. and Daniels, C. and Dehairs, F. and Deng, F. and |
| 13 | Dieu, H. T. and Duggan, B. and Dulaquais, G. and Dumousseaud, C. and Echegoyen-Sanz, Y. |
| 14 | and Edwards, R. L. and Ellwood, M. and Fahrbach, E. and Fitzsimmons, J. N. and Russell |
| 15 | Flegal, A. and Fleisher, M. Q. and van de Flierdt, T. and Frank, M. and Friedrich, J. and |
| 16 | Fripiat, F. and Fröllje, H. and Galer, S. J. G. and Gamo, T. and Ganeshram, R. S. and Garcia- |
| 17 | Orellana, J. and Garcia-Solsona, E. and Gault-Ringold, M. and George, E. and Gerringa, L. J. |
| 18 | A. and Gilbert, M. and Godoy, J. M. and Goldstein, S. L. and Gonzalez, S. R. and Grissom, K. |
| 19 | and Hammerschmidt, C. and Hartman, A. and Hassler, C. S. and Hathorne, E. C. and Hatta, |
| 20 | M. and Hawco, N. and Hayes, C. T. and Heimbürger, L.-E. and Helgoe, J. and Heller, M. and |
| 21 | Henderson, G. M. and Henderson, P. B. and van Heuven, S. and Ho, P. and Horner, T. J. and |
| 22 | Hsieh, Y.-T. and Huang, K.-F. and Humphreys, M. P. and Isshiki, K. and Jacquot, J. E. and |
| 23 | Janssen, D. J. and Jenkins, W. J. and John, S. and Jones, E. M. and Jones, J. L. and Kadko, D. |
| 24 | C. and Kayser, R. and Kenna, T. C. and Khondoker, R. and Kim, T. and Kipp, L. and Klar, J. |
| 25 | K. and Klunder, M. and Kretschmer, S. and Kumamoto, Y. and Laan, P. and Labatut, M. and |
| 26 | Lacan, F. and Lam, P. J. and Lambelet, M. and Lamborg, C. H. and Le Moigne, F. A. C. and |
| 27 | Le Roy, E. and Lechtenfeld, O. J. and Lee, J.-M. and Lherminier, P. and Little, S. and López- |
| 28 | Lora, M. and Lu, Y. and Masque, P. and Mawji, E. and McClain, C. R. and Measures, C. and |
| 29 | Mehic, S. and Barraqueta, J.-L. M. and van der Merwe, P. and Middag, R. and Mieruch, S. |
| 30 | and Milne, A. and Minami, T. and Moffett, J. W. and Moncoiffe, G. and Moore, W. S. and |
| 31 | Morris, P. J. and Morton, P. L. and Nakaguchi, Y. and Nakayama, N. and Niedermiller, J. and |
| 32 | Nishioka, J. and Nishiuchi, A. and Noble, A. and Obata, H. and Ober, S. and Ohnemus, D. C. |
| 33 | and van Ooijen, J. and O'Sullivan, J. and Owens, S. and Pahnke, K. and Paul, M. and Pavia, F. |
| 34 | and Pena, L. D. and Peters, B. and Planchon, F. and Planquette, H. and Pradoux, C. and |
| 35 | Puigcorbé, V. and Quay, P. and Queroue, F. and Radic, A. and Rauschenberg, S. and |
| 36 | Rehkämper, M. and Rember, R. and Remenyi, T. and Resing, J. A. and Rickli, J. and Rigaud, |
| 37 | S. and Rijkenberg, M. J. A. and Rintoul, S. and Robinson, L. F. and Roca-Martí, M. and |



Rodellas, V. and Roeske, T. and Rolison, J. M. and Rosenberg, M. and Roshan, S. and Rutgers van der Loeff, M. M. and Ryabenko, E. and Saito, M. A. and Salt, L. A. and Sanial, V. and Sarthou, G. and Schallenberg, C. and Schauer, U. and Scher, H. and Schlosser, C. and Schnetger, B. and Scott, P. and Sedwick, P. N. and Semiletov, I. and Shelley, R. and Sherrell, R. M. and Shiller, A. M. and Sigman, D. M. and Singh, S. K. and Slagter, H. A. and Slater, E. and Smethie, W. M. and Snaith, H. and Sohrin, Y. and Sohst, B. and Sonke, J. E. and Speich, S. and Steinfeldt, R. and Stewart, G. and Stichel, T. and Stirling, C. H. and Stutsman, J. and Swarr, G. J. and Swift, J. H. and Thomas, A. and Thorne, K. and Till, C. P. and Till, R. and Townsend, A. T. and Townsend, E. and Tuerena, R. and Twining, B. S. and Vance, D. and Velazquez, S. and Venchiarutti, C. and Villa-Alfageme, M. and Vivancos, S. M. and Voelker, A. H. L. and Wake, B. and Warner, M. J. and Watson, R. and van Weerlee, E. and Alexandra Weigand, M. and Weinstein, Y. and Weiss, D. and Wisotzki, A. and Woodward, E. M. S. and Wu, J. and Wu, Y. and Wuttig, K. and Wyatt, N. and Xiang, Y. and Xie, R. C. and Xue, Z. and Yoshikawa, H. and Zhang, J. and Zhang, P. and Zhao, Y. and Zheng, L. and Zheng, X.-Y. and Zieringer, M. and Zimmer, L. A. and Ziveri, P. and Zunino, P. and Zurbrick, C.: The GEOTRACES Intermediate Data Product 2017, Chem Geol, 493, 210-223, https://doi.org/10.1016/j.chemgeo.2018.05.040, 2018.

Schlosser, C., Streu, P., Frank, M., Lavik, G., Croot, P. L., Dengler, M., and Achterberg, E. P.: $H_2S$ events in the Peruvian oxygen minimum zone facilitate enhanced dissolved Fe concentrations, Sci Rep, 8, https://doi.org/10.1038/s41598-018-30580-w, 2018.

Schmidtko, S., Stramma, L., and Visbeck, M.: Decline in global oceanic oxygen content during the past five decades, Nature, 542, 335-339, https://doi.org/10.1038/nature21399, 2017.

Scholten, J. C., Pham, M. K., Blinova, O., Charette, M. A., Dulaiova, H., and Eriksson, M.: Preparation of Mn-fiber standards for the efficiency calibration of the delayed coincidence counting system (RaDeCC), Mar Chem, 121, 206-214, https://doi.org/10.1016/j.marchem.2010.04.009, 2010.

Scholz, F., Loscher, C. R., Fiskal, A., Sommer, S., Hensen, C., Lomnitz, U., Wuttig, K., Gottlicher, J., Kossel, E., Steininger, R., and Canfield, D. E.: Nitrate-dependent iron oxidation limits iron transport in anoxic ocean regions, Earth Planet Sc Lett, 454, 272-281, https://doi.org/10.1016/j.epsl.2016.09.025, 2016.

Schroller-Lomnitz, U., Hensen, C., Dale, A. W., Scholz, F., Clemens, D., Sommer, S., Noffke, A., and Wallmann, K.: Dissolved benthic phosphate, iron and carbon fluxes in the Mauritanian upwelling system and implications for ongoing deoxygenation, Deep-Sea Res Pt I, in review, 2018.

Severmann, S., McManus, J., Berelson, W. M., and Hammond, D. E.: The continental shelf benthic iron flux and its isotope composition, Geochim Cosmochim Ac, 74, 3984-4004, https://doi.org/10.1016/j.gca.2010.04.022, 2010.



Shelley, R. U., Morton, P. L., and Landing, W. M.: Elemental ratios and enrichment factors in aerosols
2        from the US-GEOTRACES North Atlantic transects, Deep-Sea Res Pt II, 116, 262-272,
3        https://doi.org/10.1016/j.dsr2.2014.12.005, 2015.

Shelley, R. U., Landing, W. M., Ussher, S. J., Planquette, H., and Sarthou, G.: Regional trends in the
fractional solubility of Fe and other metals from North Atlantic aerosols (GEOTRACES
cruises GA01 and GA03) following a two-stage leach, Biogeosciences, 15, 2271-2288,
https://doi.org/10.5194/bg-15-2271-2018, 2018.

Sherrell, R. M. and Boyle, E. A.: The trace metal composition of suspended particles in the oceanic
water column near Bermuda, Earth Planet Sc Lett, 111, 155-174, https://doi.org/10.1016/0012-
821x(92)90176-V, 1992.

Soataert, K., Petzoldt, T., and Meysman, F.: marelac: Tools for Aquatic Sciences, Version 2.1.6,
https://CRAN.R-project.org/package=marelac, 2016. 2016.

Steinfeldt, R., Sultenfuss, J., Dengler, M., Fischer, T., and Rhein, M.: Coastal upwelling off Peru and
Mauritania inferred from helium isotope disequilibrium, Biogeosciences, 12, 7519-7533,
https://doi.org/10.5194/bg-12-7519-2015, 2015.

Stramma, L., Brandt, P., Schafstall, J., Schott, F., Fischer, J., and Kortzinger, A.: Oxygen minimum
zone in the North Atlantic south and east of the Cape Verde Islands, J Geophys Res-Oceans,
113, C04014, https://doi.org/10.1029/2007jc004369, 2008a.

Stramma, L., Johnson, G. C., Sprintall, J., and Mohrholz, V.: Expanding oxygen-minimum zones in
the tropical oceans, Science, 320, 655-658, https://doi.org/10.1126/science.1153847, 2008b.

Stumm, W. and Morgan, J. J.: Aquatic Chemistry: Chemical Equilibria and Rates in Natural Waters,
John Wiley & Sons, New York, 1995.

Sunda, W. G. and Huntsman, S. A.: Effect of sunlight on redox cycles of manganese in the
Southwestern Sargasso Sea, Deep-Sea Res, 35, 1297-1317, https://doi.org/10.1016/0198-
0149(88)90084-2, 1988.

Sunda, W. G. and Huntsman, S. A.: Photoreduction of manganese oxides in seawater, Mar Chem, 46,
133-152, https://doi.org/10.1016/0304-4203(94)90051-5, 1994.

Tebo, B. M. and Emerson, S.: Microbial manganese(II) oxidation in the marine environment: a
quantitative study, Biogeochemistry, 2, 149-161, https://doi.org/10.1007/Bf02180192, 1986.

Tebo, B. M., Bargar, J. R., Clement, B. G., Dick, G. J., Murray, K. J., Parker, D., Verity, R., and
Webb, S. M.: Biogenic manganese oxides: Properties and mechanisms of formation, Annu
Rev Earth Pl Sc, 32, 287-328, https://doi.org/10.1146/annurev.earth.32.101802.120213, 2004.

Thomsen, S., Karstensen, J., Kiko, R., Krahmann, G., Dengler, M., and Engel, A.: Remote and local
drivers of oxygen and nitrate variability in the shallow oxygen minimum zone off Mauritania
in June 2014, Biogeosciences, 1-29, https://doi.org/10.5194/bg-2018-252, 2018.





Tomczak, M.: An analysis of mixing in the frontal zone of South and North Atlantic Central Water off
North-West Africa, Prog Oceanogr, 10, 173-192, https://doi.org/10.1016/0079-
3      6611(81)90011-2, 1981.

Twining, B. S., Rauschenberg, S., Morton, P. L., and Vogt, S.: Metal contents of phytoplankton and
labile particulate material in the North Atlantic Ocean, Prog Oceanogr, 137, 261-283,
https://doi.org/10.1016/j.pocean.2015.07.001, 2015.
Ussher, S. J., Achterberg, E. P., Powell, C., Baker, A. R., Jickells, T. D., Torres, R., and Worsfold, P.
J.: Impact of atmospheric deposition on the contrasting iron biogeochemistry of the North and
South Atlantic Ocean, Global Biogeochem Cy, 27, 1096-1107,
https://doi.org/10.1002/gbc.20056, 2013.
Véron, A., Patterson, C., and Flegal, A.: Use of stable lead isotopes to characterize the sources of
anthropogenic lead in North Atlantic surface waters, Geochim Cosmochim Ac, 58, 3199-3206,
https://doi.org/10.1016/0016-7037(94)90047-7, 1994.
von Langen, P. J., Johnson, K. S., Coale, K. H., and Elrod, V. A.: Oxidation kinetics of manganese(II)
in seawater at nanomolar concentrations, Geochim Cosmochim Ac, 61, 4945-4954,
https://doi.org/10.1016/S0016-7037(97)00355-4, 1997.
Wedepohl, K. H.: The composition of the continental crust, Geochim Cosmochim Ac, 59, 1217-1232,
https://doi.org/10.1016/0016-7037(95)00038-2, 1995.
Weiss, R. F.: The solubility of nitrogen, oxygen and argon in water and seawater, Deep Sea Res and
Oceanographic Abstracts, 17, 721-735, https://doi.org/10.1016/0011-7471(70)90037-9, 1970.
Winkler, L. W.: Bestimmung des im Wasser gelösten Sauerstoffs, Ber Dtsch Chem Ges, 21, 2843-
2855, https://doi.org/10.1002/cber.188802102122, 1988.
Wu, J. F. and Luther, G. W.: Size-fractioned iron concentrations in the water column of the western
North Atlantic Ocean, Limnol Oceanogr, 39, 1119-1129,
https://doi.org/10.4319/lo.1994.39.5.1119, 1994.
Wyrtki, K.: The oxygen minima in relation to ocean circulation, Deep-Sea Res, 9, 11-23,
https://doi.org/10.1016/0011-7471(62)90243-7, 1962.
Zenk, W., Klein, B., and Schroder, M.: Cape-Verde Frontal Zone, Deep-Sea Res, 38, S505-S530,
https://doi.org/10.1016/S0198-0149(12)80022-7, 1991.



1  **Table 1.** Analyzed reference seawater, procedural blanks and detection limits (three times the standard

2  deviation of the blank). Mean values and standard deviation for Cd, Pb, Fe, Ni, Cu, Mn and Co and

3  available consensus values ($\pm$ 1 standard deviation), n = number of measurements.

| | SAFe S (nmol L$^{-1}$) n=11 | SAFe S consensus value (nmol L$^{-1}$) | SAFe D2 (nmol L$^{-1}$) n=7 | SAFe D2 consensus value (nmol L$^{-1}$) | Blank (pmol L$^{-1}$) | Detection limit (pmol L$^{-1}$) |
|---|---|---|---|---|---|---|
| Cd | 0.003 ± 0.002 | 0.001 | 1.089 ± 0.043 | 1.011 ± 0.024 | 2.2 ± 0.3 | 0.8 |
| Pb | 0.050 ± 0.003 | 0.049 ± 0.002 | 0.028 ± 0.001 | 0.029 ± 0.002 | 0.4 ± 0.2 | 0.6 |
| Fe | 0.091 ± 0.009 | 0.095 ± 0.008 | 1.029 ± 0.038 | 0.956 ± 0.024 | 68 ± 10 | 29 |
| Ni | 2.415 ± 0.086 | 2.34 ± 0.09 | 9.625 ± 0.175 | 8.85 ± 0.26 | 112 ± 20 | 59 |
| Cu | 0.514 ± 0.037 | 0.53 ± 0.05 | 2.176 ± 0.152 | 2.34 ± 0.15 | 14 ± 3 | 9.3 |
| Co | 0.005 ± 0.001 | 0.005 ± 0.001 | 0.048 ± 0.003 | 0.047 ± 0.003 | 2.7 ± 0.8 | 2.5 |
| Mn | 0.814 ± 0.033 | 0.810 ± 0.062 | 0.437 ± 0.029 | 0.36 ± 0.05 | 14 ± 6 | 17 |





1   **Table 2.** TM/Al ratios of different fractions for

2   the repeated deployments at station 3 within the

3   OMZ below 50 m water depth.

| Parameter | Stn 3A | Stn 3B |
|---|---|---|
| dFe/dAl | 0.38–0.79 | 0.35–0.37 |
| TDFe/TDAl | 4.00–13.42 | 1.83–2.81 |
| LpFe/LpAl | 10.00–29.50 | 3.64–8.59 |
| dCo/dAl | 0.009–0.011 | 0.009–0.011 |
| TDCo/TDAl | 0.009–0.010 | 0.006–0.008 |
| LpCo/LpAl | 0.007–0.011 | 0.001–0.005 |
| dMn/dAl | 0.26–0.45 | 0.19–0.21 |
| TDMn/TDAl | 0.26–0.32 | 0.12–0.17 |
| LpMn/LpAl | 0.14–0.28 | 0.02–0.09 |





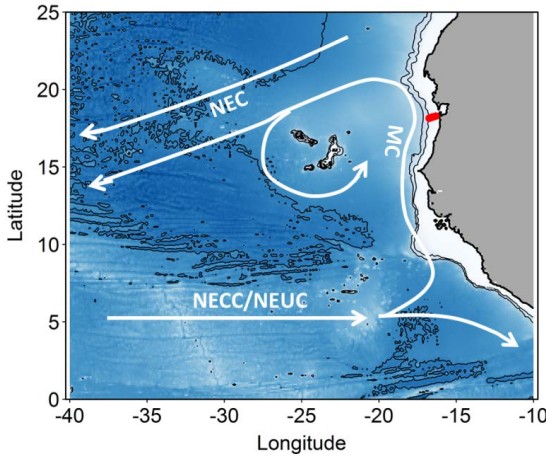

**Figure 1.** Map of the study area. Stations along the transect during M107 (June 2014) are displayed in
red circles and major currents in white lines (adapted from Brandt et al. 2015). MC = Mauritania
Current; NEC = North Equatorial Current; NECC = North Equatorial Countercurrent; NEUC = North
Equatorial Undercurrent.



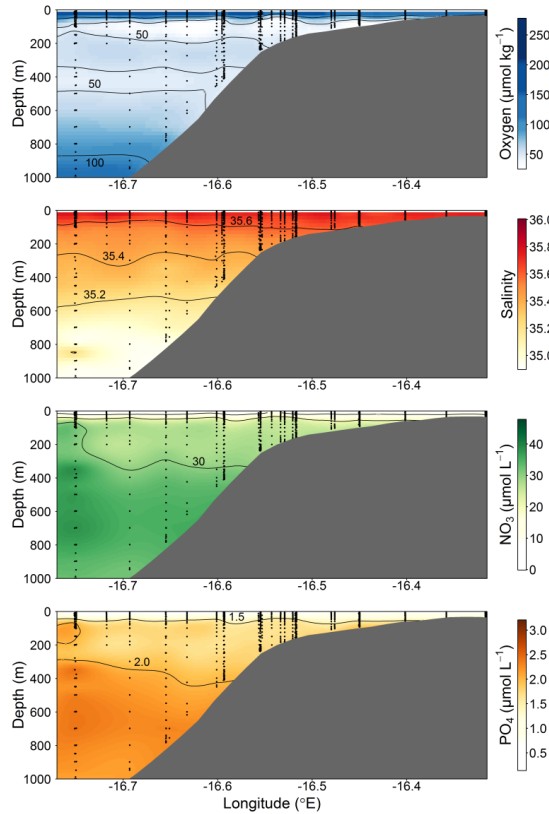

2    **Figure 2.** Section plots of oxygen (μmol kg$^{-1}$), salinity (PSU), NO$_3$ (μmol L$^{-1}$) and PO$_4$ (μmol L$^{-1}$)

3    along the transect off the Mauritanian coast in June 2014.







**Figure 3.** Spatial distributions of dissolved (d) and leachable particulate (Lp) trace metals and $^{224}$Ra/$^{223}$Ra across the Mauritanian shelf at 18°20'N in June 2014. Each sample location is indicated as black dot and oxygen contours at 50 µmol kg$^{-1}$ enclosing the upper and lower OMZ are displayed as black contour lines.



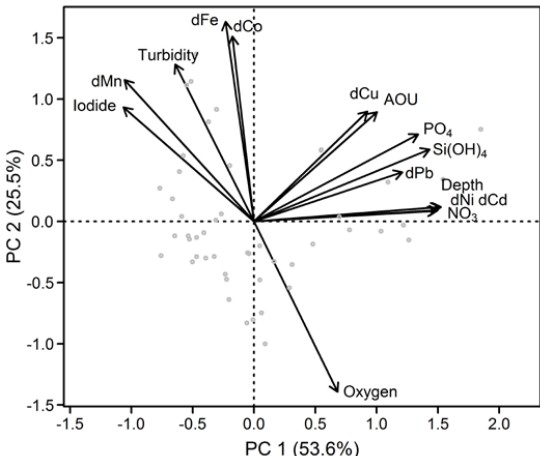

**Figure 4.** Principal component analysis of the Mauritanian shelf data set. Principal component
loadings for each variable are indicated by black vectors. Component scores of each sample are
indicated as grey circles. Loadings/scores have been scaled symmetrically by square root of the
eigenvalue.





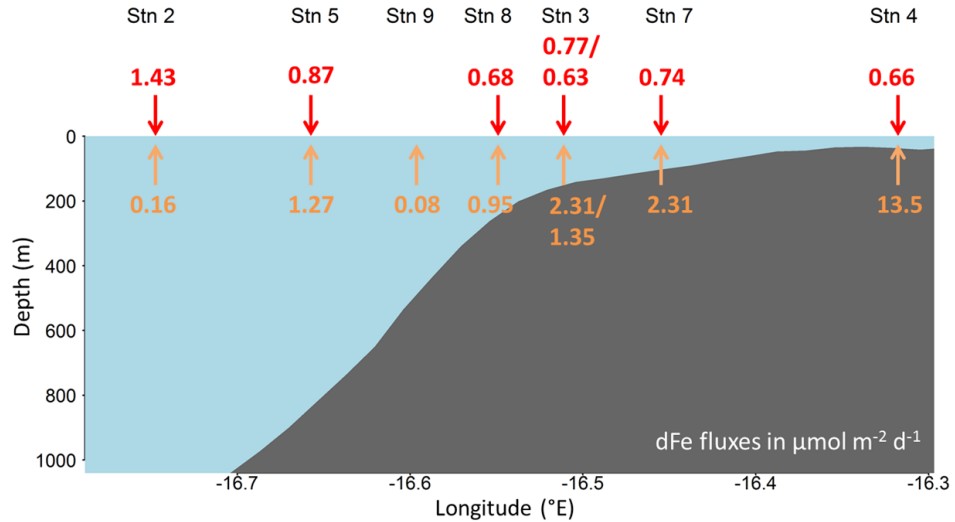

2 **Figure 5.** Atmospheric dFe fluxes (red) and vertical dFe fluxes (orange) in $\mu$mol m$^{-2}$ d$^{-1}$ along the

3 transect at 18°20'N in June 2014.



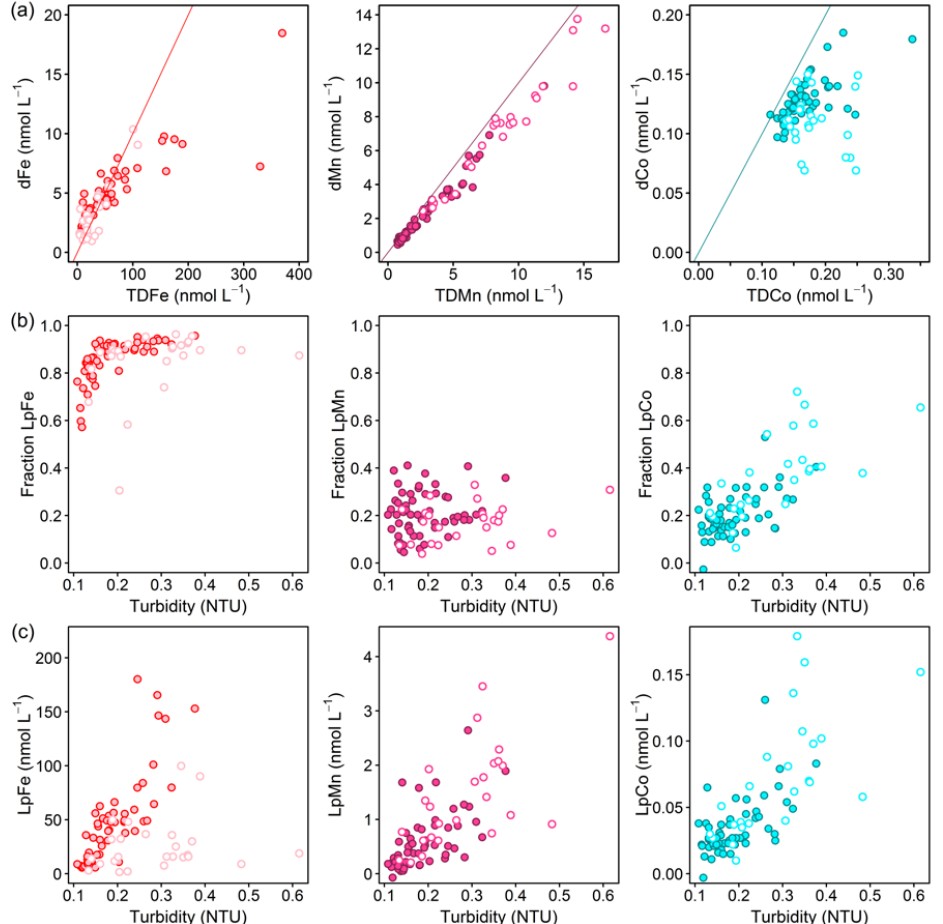

**Figure 6.** (a) Dissolved against total dissolvable trace metal concentrations for Fe (left; red line: TDFe
= 10*dFe), Mn (middle; purple line: TDMn = dMn) and Co (right; turquoise line: TDCo = dCo). (b)
Fraction of leachable particulate trace metals (Lp/TD) against turbidity and (c) Leachable particulate
concentrations against turbidity for Fe (left), Mn (middle) and Co (right). Filled circles display all data
points below 50 m depth, open circles at depths shallower than 50 m.



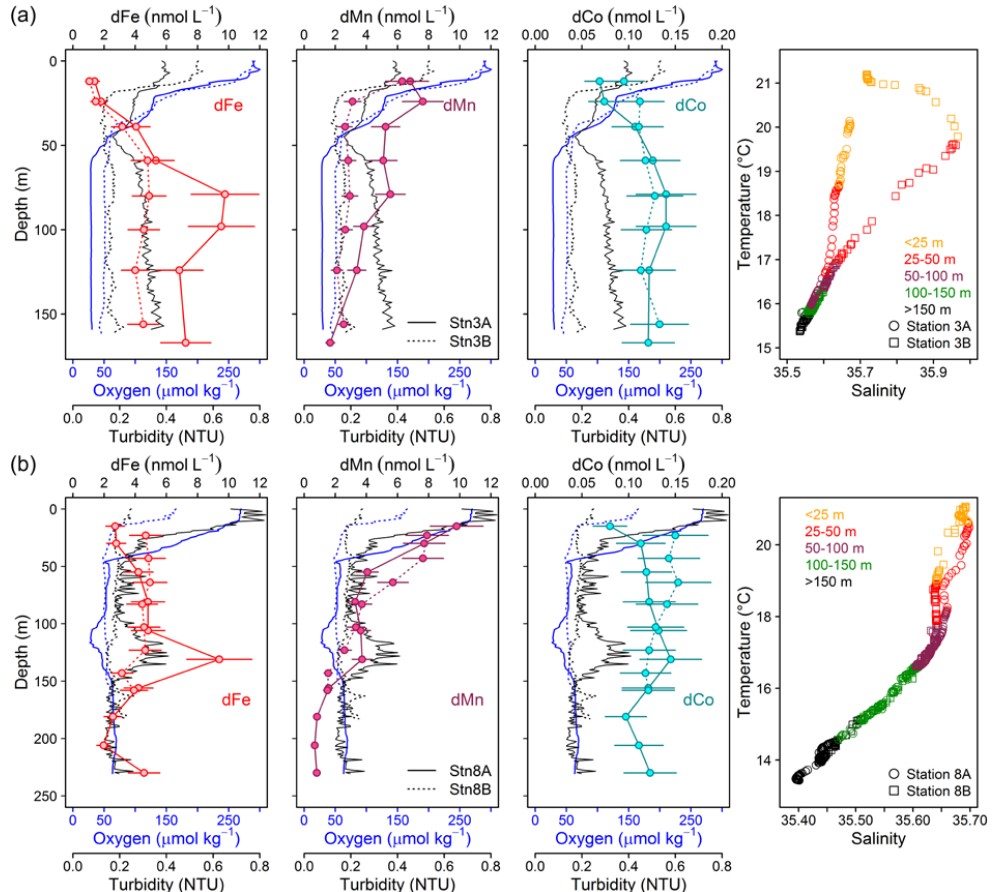

**Figure 7.** Repeat stations: oxygen concentration, turbidity and dissolved trace metals (Fe, Mn and Co) and temperature vs salinity plots. First deployment displayed as solid line and second deployment displayed as dashed line. (a) Station 3 (18.23°N, 16.52°W, 170 m water depth, 9 days between deployments). (b) Station 8 (18.22°N, 16.55°N, 189–238 m water depth, 2 days between deployments).





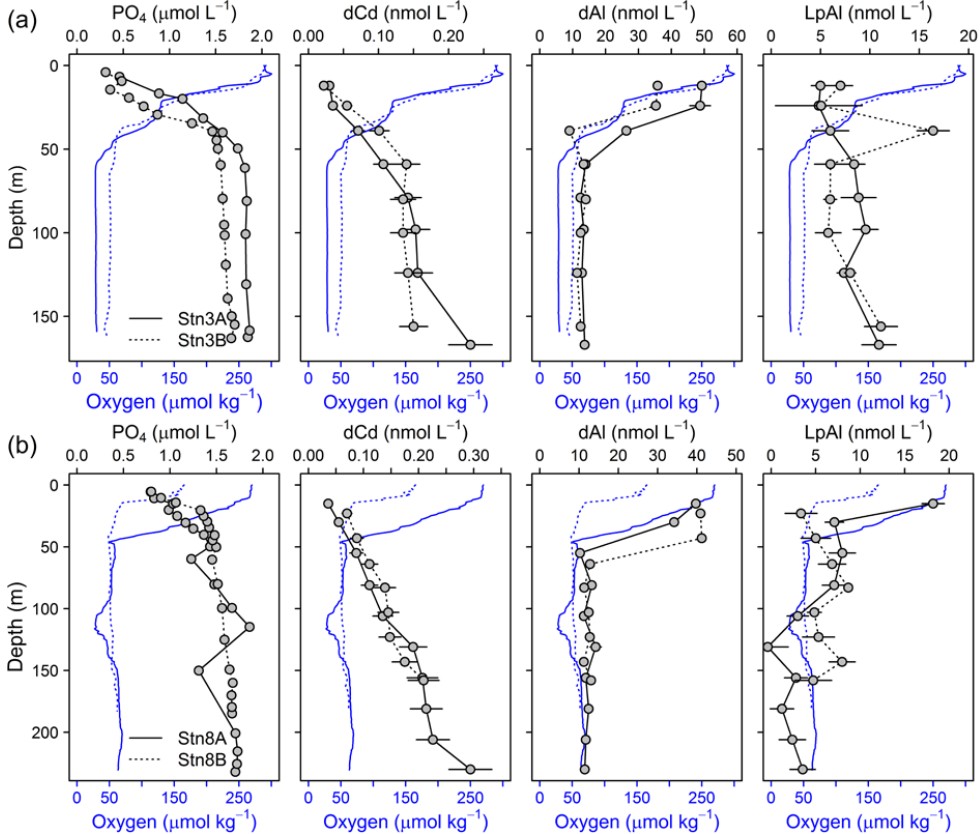

**Figure 8.** Depth profiles of dCd, $PO_4$, dAl and LpAl of repeat stations. First deployment displayed as solid black line and second deployment displayed as dashed black line. Oxygen concentrations are indicated as blue solid line for the first deployment and dashed blue line for the second deployment. (a) Station 3 (18.23°N, 16.52°W, 170 m water depth, 9 days between deployments and (b) Station 8 (18.22°N, 16.55°N, 189–238 m water depth, 2 days between deployments).