# Peer review of "Controls on redox-sensitive trace metals in the Mauritanian oxygen minimum zone"

_Biogeosciences, 2018_

## Referee Comment (RC1) · Anonymous Referee #1 · 21 Jan 2019

See attached review

Please also note the supplement to this comment:
https://www.biogeosciences-discuss.net/bg-2018-472/bg-2018-472-RC1-supplement.pdf

---

## Referee Comment (RC2) · Anonymous Referee #2 · 28 Jan 2019

This is an important paper that illustrates the potential importance of benthic shelf sediments as a source of Fe to the oceans interior, and linking spatial and temporal variability to oxygen concentrations. While I think the authors make a good case for the role of oxygen, the paper is flawed by three serious omissions that must be corrected before publication.

1. Iodide is reported as a critical parameter in the principal component analysis and there are detailed protocols for iodide analysis, yet no data are reported in the paper or in the supplement. These data are of great interest in their own right. While iodide has been reported in truly anaerobic, denitrifying water columns, it has not been well studied in these low oxygen regimes. One presumes that these data will appear in Pangaea eventually, but why not here?

2. Similarly, Ra-228 data are not reported, nor is there any quantitative assessment of Ra-228 correlations with Fe to support their conclusions about lateral transport.

3. The authors imply that the approach used to determine vertical eddy diffusivity will appear in the Supplementary materials, but it does not.

---

## Author Comment (AC1) · 30 Apr 2019

**Response to Referee 1**

Overview:

This paper presents trace metal data from the Mauritanian shelf and places the observed distributions of redox elements in the context of $O_2$ control. The trace metal data look to be excellent, though the flux calculations need some further explaining so the reader can follow them through every step that was made. Overall the arguments presented for $O_2$ as a control in this dynamic environment are very weak and instead the data points more to the role of scavenging, aerosol deposition, resuspension and cross shelf transport. Just because it is in an oxygen minimum zone does not mean that oxygen controls the distribution of redox elements, as the data looks very similar to that from most coastal shelf regions. The paper at present is overly long and should have been edited down before submission to a more concise set of sections concentrating on the main processes as there are several sections (e.g. 3.4.2) that could summarize most of the relevant information to a simple paragraph with inclusion of a summary table. A substantially revised manuscript focusing on the key findings for which there is evidence would likely have significant impact in this field.

**We thank the reviewer for the detailed constructive comments on our manuscript and address each comment in the following. The responses to the reviewer are given in bold font. Changes we made to the manuscript are highlighted in blue. Page and line numbers refer to the position in the original manuscript.**

General Comments:

Contradictory vertical flux information: The main problem with the paper at present is with the vertical flux calculations as the repeat station data is analysed in an Eulerian framework with the concept that it is the same water mass that is being sampled at the same site several days later and horizontal advection is ignored. Yet the paper clearly states that the changes seen between repeat samplings was from an advective inflow (Thomsen et al., 2018) and that this regions is known for its high current velocities in summer (Klenz et al., 2018). While I understand what the authors are trying to link their data with changes in $O_2$, in the absence of a Lagrangian framework this makes no sense as we are left with comparing two snap shots of completely different scenes that just happened to be at the same spot.

**The vertical flux calculation used here is independent of lateral (advective and diffusive) processes. The misunderstanding may be due to an ill-worded sentence in which we tried to explain the deviation of the vertical advective and diffusive fluxes.**
**Certainly, there are no advective terms in a Lagrangian framework, only diffusive terms remain. However, it is unclear to us why one would like to interpret our data in a Lagrangian framework. Thomsen et al. (2019) have shown that the $O_2$ variability is caused by alongshore advection and to a somewhat smaller extent by enhanced local consumption. Here, we look at average vertical processes and have the advantage of sampling during two rather different situations. Close to the shelf at the location of our repeated stations however, the variability in oxygen was explained by changes in local oxygen respiration rates and not by a change in water mass composition (Thomsen et al. 2019). We concur that the description of the oxygen variability on Page 11, Line 17 may have been unclear and have clarified the following part accordingly:**
Variability in oxygen concentrations observed further offshore was attributed to physical transport of SACW into the region (Thomsen et al., 2019). In contrast, closer to the coast, enhanced pelagic

oxygen consumption rates were determined that significantly contribute to the variability in observed oxygen concentrations (Thomsen et al., 2019).

The description of the flux calculations is also not easy to follow and it is unclear exactly what depth ranges are being considered, is it just into the mixed layer or is it over the entire depth range. The strong point is the combination of microstructure profiles and trace metal concentrations to estimate robust values for the diffusive flux. The advective fluxes however are not well constrained at all as the methodology clearly has some problems; firstly close to the coast, scatterometer winds are not reliable due to the masking that takes place close to the coast in the data analysis, so some explanation of how this was taken into account needs to be provided, secondly the

**In the revised version, we have substantially improved the description of the flux calculation based on the reviewer's specific remarks (detailed in the reply to the specific remarks below). We also included a discussion of the uncertainties inherent to determining vertical advective fluxes. The quality of the scatterometer winds close to the coast have been found to be almost as good as for the open sea winds (e.g. Verhoef et al., 2012). Mauritania–Senegalese upwelling region comparisons between scatterometer winds and near coastal land stations have shown differences of up to 20% in meridional and zonal wind component (Ndoye et al., 2014). However, Steinfeldt et al. (2015) validated the parameterized vertical velocities (Gill, 1982) used here against scatterometer winds (see Figure 4a in their paper). The agreement between cruise average vertical velocities (obtained from the helium disequilibrium and turbulence data) and the upwelling velocities derived via Gill's parameterization using satellite winds was very encouraging and motivated us to apply this formalism here. The revised description of the flux calculations in section 2.7 reads as follows:**

**2.7 Turbulence measurements and vertical flux calculations**

In order to advance understanding of the role of benthic Fe supply to the productive surface waters of the upwelling region, vertical diffusive fluxes (eq 1: left term, right hand side) and wind induced vertical advective fluxes (eq 1: right term, right hand side) were estimated. On the continental margin below the surface mixed layer, solutes are transferred vertically toward the near-surface layers by turbulent mixing processes and by vertical advection forced by Ekman divergence (e.g. Kock et al., 2012; Milne et al., 2017; Rhein et al., 2010; Steinfeldt et al., 2015, Tanhua and Liu, 2015):

$$J_Z = K_z \frac{\partial [TM]}{\partial z} + w \cdot \Delta [TM] \qquad (1)$$

Here, $K_z$ is the turbulent eddy diffusivity in $m^2\ s^{-1}$, $\partial [TM]/ \partial z$ the vertical gradient with depth (z) of the TM concentration [TM] in $\mu mol\ m^{-4}$, $\Delta [TM]$ a TM concentration difference in $\mu mol\ m^{-3}$ and $w$ represents vertical velocity in $m\ s^{-1}$. Average advective and diffusive TM fluxes were calculated for a depth interval from the shallow $O_2$-depleted waters to surface waters. The exact depth interval varied for each station (see Table S2) due to differences in the depths where TM samples were collected. The upper depth (8–29 m) was always in layers with enhanced chlorophyll $a$ fluorescence, although for some stations the upper depth was below the surface mixed layer.

Diffusive Fe fluxes were determined by combining TM concentration measurements from the TM-CTD stations with nearby measured microstructure profiles. The microstructure measurements were performed with an MSS90-D profiler (S/N 32, Sea & Sun Technology). The loosely-tethered profiler was optimized to sink at a rate of $0.55\ m\ s^{-1}$ and equipped with three shear sensors, a fast-response temperature sensor, and an acceleration sensor, two tilt sensors and conductivity, temperature, depth sensors sampling with a lower response time. At TM-CTD stations with bottom depths less than 400 m, 18 to 65 microstructure profiles were available at each station. At deeper stations, 5 to 12 profiles were used. Standard processing procedures were used to determine the rate of kinetic

energy dissipation (ε) of turbulence in the water column (see Schafstall et al. (2010) for detailed description). Subsequently, $K_z$ values were determined from $K_\rho = \Gamma\varepsilon N^{-2}$ (Osborn, 1980), where $N$ is stratification and $\Gamma$ is the mixing efficiency for which a value of 0.2 was used. The use of this value has recently been shown to yield good agreement between turbulent eddy diffusivities determined from microstructure measurements and from tracer release experiments performed in our study region (Köllner et al., 2016). The 95% confidence intervals for station-averaged $K_\rho$ values were determined from Gaussian error propagation following Schafstall et al. (2010). Finally, diffusive fluxes were estimated by multiplying station-averaged $K_\square$ with the vertical gradient of the respective TM solute, implicitly assuming $K_z = K_\rho$.

The vertical advective flux by Ekman divergence requires determination of vertical velocity in the water column that varies with depth and distance from the coast line. Convincing agreement between vertical velocities derived from Ekman divergence following Gill (1982) determined from scatterometer winds and from helium isotope disequilibrium within the Mauritanian and Peruvian coastal upwelling regions was found by Steinfeldt et al. (2015) (see their Fig. 4). In their study, vertical velocities were parameterized as (Gill, 1982):

$$w = \frac{\tau_y}{\rho f L_r} e^{-x/L_r} \qquad (2)$$

where $\tau_y$ represents the alongshore wind stress, $\rho$ the density of sea water, $x$ the distance from maximum Ekman divergence taken here as the position at 50 m bottom depth on the shelf and $L_r$ the first baroclinic Rossby radius. The parameterization results from considering the baroclinic response of winds parallel to a coastline in a two-layer ocean (Gill, 1982). The baroclinic Rossby radius $L_r = f^{-1} \sqrt{g \frac{\rho_2 - \rho_1}{\rho} \frac{H_1 H_2}{H_1 + H_2}}$ ($\rho_{1/2}$ and $H_{1/2}$ are density and thickness of the surface and lower layer, respectively) was found to be 15 km from hydrographical data collected during the cruise. Similar values were determined by Steinfeld et al. (2015) in the same region. Using average alongshore wind stress from satellite data (0.057 Nm[-2], determined from daily winds from Remote Sensing Systems ASCAT C-2015, version v02.1 (Ricciardulli and Wentz, 2016) at 18°22.5'N, 016°7.5'W using $\tau_y = \rho_{air} C_d v^2$, where $v$ represents alongshore wind, $C_d$ is drag coefficient for which 1.15×10[-3] was used (e.g. Fairall et al., 2003) and $\rho_{air}$ is density of air) for June 2014, maximum vertical velocities of 3.7×10[-5] m s[-1] were determined for the shelf region (50 m water depth), which decayed offshore to 1.7×10[-6] m s[-1] at the position of the 1000 m isobath at 18°N. As these vertical velocities describe the magnitude of upwelling at the base of the mixed-layer, additional corrections need to be considered for deeper depths. Here, we approximated the vertical decay of $w$ as a linear function which diminishes at the ocean floor.

The calculation of the vertical advective flux supplying solutes from the shallow $O_2$-depleted waters to surface waters requires knowledge of a concentration difference Δ[TM] associated with the upwelling flux. Ideally, the vertical length scale over which the concentration difference is determined can be diagnosed as the TM concentration variance divided by its mean vertical gradient (e.g. Hayes et al., 1991). However, in our study TM concentration time series data are not available. Previous studies have used a vertical length scale of 20 m to calculate the concentration differences between the target depth and the water below (e.g. Hayes et al., 1991; Steinfeldt et al., 2015; Tanhua and Liu, 2015). For our calculations, we chose to use a smaller length scale of 10 m following Hayes et al. (1991) which results in vertical advective TM flux presumably on the lower side of possible values.

**The value of the alongshore wind stress utilized here changed from the original version. This is due to a change in location where the wind stress was calculated for, which is 12 km further inshore. This however has little impact on our calculated dFe fluxes, as the vertical velocity is calculated**

**separately for each station location using a term decreasing vertical velocities with distance from the location where the wind stress was calculated (equation 3).**

**To adequately describe uncertainties inherent to the vertical advective flux calculation we added the following text to section 3.4.3:**

It should be noted that there are considerable uncertainties in the flux estimates presented above. While uncertainties in the diffusive flux originate predominately from the elevated variability of turbulence (see Schafstall et al., 2010 for details), uncertainties in the vertical advective flux originate from unaccounted for  contributions from e.g. the spatial structure of the wind, particularly in the offshore direction, its temporal variability (e.g. Capet et al., 2004; Desbiolles et al. 2014, 2016; Ndoye et al., 2014), and uncertainties in the satellite wind product  near the coast (e.g. Verhoef et al, 2012). Furthermore, the distribution of vertical velocities with depth is assumed to be linear here.

At the time this work was performed along the Mauritanian coast the area that was sampled has typically little or no upwelling present (Cropper et al., 2014; Tanhua and Liu, 2015; Varela et al., 2015). In the present work the authors chose to use the same wind based approach as that used earlier by Steinfeldt et al. (2015) in as study based predominantly on He isotopes. In that study they found upwelling during the summer months (M68-3) though the error bars are quite large 2.4 ± 1.5 x 10-5 m s$^{-1}$. Steinfeldt commented on the differences between their work and Tanhua and Liu, noting that the latter's data set only contains a few stations along 18 N, while their coastal stations along that line also have low vertical velocities (between 0 and 2x10-5 ms$^{-1}$), with much higher upwelling velocities to the north. It is not to say that there wasn't upwelling at this time, just that it needs to be better explained and put into context, indeed support for upwelling at this time comes from comparison to a companion paper also in BGD at present (Thomsen et al., 2018) but the authors don't make mention of that work except for the low oxygen resuspension events. The present paper would clearly benefit from linking more to the Thomsen et al. (2018) work and also to Yücel et al. (2015) work carried out along the same transect as both of those papers show the inherent variability in dissolved parameters in the bottom waters of this region.

**Maps of SST and alongshore winds measured by the research vessel during the measurement program are shown Figure 1 by Thomsen et al. (2019). As stated above, cold surface waters were still present near the Mauritanian coast and moderate equatorward alongshore wind were encountered during the cruise, ensuring offshore Ekman transport. The work by Thomsen et al. (2019) is now referenced in several places of the manuscript. We also compare our results to Yücel et al. (2015).**

The physical oceanography presented in the manuscript would also benefit from consideration of recent modelling studies on upwelling from other EBUS (Jacox and Edwards, 2011, 2012; Lentz, 1992; Lentz and Chapman, 2004; Lentz and Fewings, 2012; Messié and Chavez, 2015). The studies listed here show how the Burger number (Lentz and Chapman, 2004) with its dependence on the shelf slope and buoyancy frequency were important in understanding upwelling in such regions. Inclusion of this information would then also give the current paper more impact as it would be applicable to other EBUS as well. There are also a number of recent papers on the role of filaments and particle fluxes in this region (Bory et al., 2001; Fischer et al., 2009; Iversen et al., 2010; Rees et al., 2011) which also could be useful for interpreting the results found in the current work.

**We thank the reviewer for this comment and the additional references provided. However, we feel that for the current study, such additional details of the physical oceanography are not essential for interpreting our measured trace element concentrations, which is the overall objective of this paper. Particle transport along filaments may indeed be an important transport mechanisms of trace metals in this region, which we mention in the manuscript. However, we cannot further discuss this using our data, as we ourselves didn't analyse particles and their distribution  and composition likely shows a large temporal and spatial variability (as also pointed out in the**

mentioned papers) and we therefor can only refer to the general possible role, but don't see the possibility in further interpreting our results using their data.

$O_2$ as a sole control on metal abundance: This work tries to suggest that changes in metal concentrations in the water column are due to changes in $O_2$ concentrations but the $O_2$ values observed are not low enough to cause significant changes in Fe, Mn or Al redox speciation so it is more likely that the observed changes are related to resuspension events in this very dynamic mixing environment (Schafstall et al., 2010). Resuspension has been known to be a major control on dissolved iron and other metals in shelf regions for some time now (Croot and Hunter, 1998; de Jong et al., 2012; Elrod et al., 2004; Johnson et al., 1999) and this would also impact other hydroxide dominated elements (e.g. Al, Th and to a lesser extent Mn). Colloidal species are also likely important here (Moran and Moore, 1988, 1989; Schlosser and Croot, 2008; Schlosser et al., 2013).

**We are not trying to argue that oxygen is the only control on trace metal distribution and sediment release in this area. In the sections on remineralization and atmospheric deposition, we also highlight the additional influence of these processes. However, the temporal variation in trace metal concentrations observed here seems to be majorly driven by the change in oxygen concentrations despite the large concomitant change in turbidity. In section '3.6.2', we discuss the respective role of sediment resuspension but our dAl and LpAl data do not support a large change in sediment resuspension between the two deployments. Dissolved Al concentrations are very similar between the two deployments, while LpAl is lower at station 8 when dFe and LpFe are higher and slightly higher at station 3 when dFe and LpFe are larger. As pointed out by the reviewer, sediment resuspension (and also dust deposition) would, in addition to Fe, impact Al concentrations.**

Why is $O_2$ unlikely to be a control on trace metals in the Mauritanian shelf? Simple - the shelf region here is typically described as being well oxygenated in the sediments (Gier et al., 2016) with the exception of the extremely shallow parts of the Banc d'Arguin (Duineveld et al., 1993; Kock et al., 2008; Schafstall et al., 2010). The only real sediment work I am aware of is that from Nolting et al. (1999) and Gier et al. (2016) and these took place mostly in oxygenated sediments it appears. That $O_2$ plays a role in the release of such metals is well known (Homoky et al., 2012; Severmann et al., 2010), but iron is only released reductively when $O_2$ is completely depleted (Sundby et al., 1986). The Severmann et a. (2010) work shows a nice relationship between Fe fluxes and $O_2$, with the large caveat that the landers used there likely consume all the $O_2$ in the sediment (though not all of the oxygen in the water in the chamber) before the Fe efflux begins, but then even at the $O_2$ concentrations found on the Mauritanian shelf the flux they predict is not lot large compared to what could be generated by resuspension – additionally intense mixing tends to keep oxygen levels higher lowering the flux of reduced iron but increasing that from resuspension. Interestingly similar iron fluxes to the Mauritanian are seen in the well-oxygenated waters of the Ross Sea (Marsay et al., 2014) indicating the role of resuspension dominates and not oxygen.

**In a study of sediment composition and benthic fluxes during the same cruise (Schroller-Lomnitz et al., 2018), enhanced Fe(II) concentrations of up to 30 µM are observed within a few cm sediment depth, indicating reductive dissolution does occur in these sediments. These Fe(II) concentrations are not as high as observed in some other regions (Severmann et al. 2010), but are for example also similar to Fe(II) concentrations in sediment underlying the entirely anoxic Peruvian OMZ (Scholz et al. 2014). We acknowledge that oxygen concentrations are not the only factor influencing benthic Fe fluxes. Other factors are for example organic matter content, sediment type, sediment grain size and shelf topography influencing sediment resuspension (Homoky et al., 2016). Such factors can result in different benthic Fe fluxes from different regions despite similar bottom water oxygen concentrations. In order to provide this information to the reader, we modified the following sentence at Page 4 Line 5:**

Differences in benthic TM supply in field studies however suggest other factors such as sediment type, shelf topography and organic matter supply also influence the benthic release of TMs (Homoky et al., 2016)

**However, these are all factors that do not vary for the same location within a few days. Sediment resuspension is indeed another factor that might have changed fluxes in our study regions, however as mentioned above, our Al data from the repeated stations do not support this. While oxygen concentrations in the water column are not low enough to reduce Fe in situ (although reduction of Fe might be argued to occur in anoxic micro-environments within particles (Bianchi et al. 2018)), a change in water column oxygen might reduce the oxic-anoxic transition zone within the sediments facilitating Fe(II) efflux and slowing down oxidation kinetics (Homoky et al., 2012, Millero et al., 1987). Dissolved Fe released from reducing sediments might be prevented from immediate precipitation once in contact with oxygen by stabilizing processes such as ligand-complexation and a large fraction of the dFe might be present as colloids. Benthic Fe fluxes in the Ross Sea appear to be around an order of magnitude smaller than the fluxes observed during our cruise in the Mauritanian OMZ (Schroller-Lomnitz et al. 2018). Also dFe concentrations near the seafloor in the Ross Sea are much lower than our observed near bottom dFe concentrations (Marsay et al. 2014).**

A previous study in the Mauritanian upwelling along almost the same transect (Schlosser and Croot, 2009) indicated that iron solubility was apparently controlled by remineralization as it varied with $O_2$, pH and phosphate concentrations. While no measurements of Fe solubility were made in the present work it does raise the question of whether pH may also be a control on the distribution of metals seen in this study but which has not been considered here. Although the PCA analysis made in this work would suggest that the resuspended elements (Fe, Al, Mn) were not related to the remineralization indicators.

**Indeed, pH might have an influence on the solubility of trace metals. Unfortunately, no pH data are available from our cruise. As pointed out by the reviewer, the PCA analysis does not indicate a nutrient-like remineralization-driven distribution of Fe, Co and Mn. Therefore, remineralisation processes do not seem to be a major driver for these elements here.**

The importance of scavenging on dissolved metals is mentioned in the introduction but after page 3 it does not come back into the text. Given that there is high particle numbers across this transect , either from the high productivity in surface waters, dust deposition, or from benthic resuspension events it is amazing that this process is not invoked as a control on dissolved metal concentrations in the discussion. The turbidity data indicates that particles are likely important, but this is not discussed in any context in the manuscript. If $O_2$ is the control that the authors say it is then they should at least follow through from their introduction and examine the role of scavenging on the scavenged elements.

**While we indirectly mention this process at a few places throughout the manuscript, we didn't explicitly use the term scavenging very often. We changed a few paragraphs to use the term scavenging for a better linkage between the processes described in the introduction and our data discussion, and have further extended the discussion in respect to scavenging processes.**

**List of changes:**

**Paragraph Page 21, Line 6:** Particles in the water column can either comprise a source or a sink of dissolved TMs. In the top 50 m of the water column a large part of the LpTMs may be part of living biological cells (e.g. phytoplankton) or organic detritus, and can enter the dissolved TM pool by remineralization (Bruland and Lohan, 2006) Additionally, LpTMs may be part of lithogenic phases from Saharan dust and sediment particles, or authigenic phases. Authigenic phases are formed in-situ by TM adsorption onto particle surfaces or by the formation of amorphous TM oxides and hydroxides

(e.g. FeO(OH) in the mineral structure of goethite) (Sherrell and Boyle, 1992), processes referred to as scavenging.The extent of scavenging processes is largely influenced by the amount and type of particles present (Balistrieri et al., 1981; Honeyman et al., 1988).

**Sentence Page 21, Line 23:** Furthermore, offshore transport of acid-labile Fe particles formed by scavenging (oxidation/adsorption) of dissolved Fe originating from a benthic source was observed in the North Pacific (Lam and Bishop, 2008) and may contribute to the bioavailable Fe pool.

**Sentence Page 22 , Line 2:** The large additional LpFe source at depths is likely associated with benthic dFe inputs, with a subsequent transfer to the particulate phase by scavenging.

Atmospheric flux estimates: Using dissolved Al concentrations to estimate the aerosol flux of the other elements is complicated as while there is good data on the elemental composition of the Saharan aerosols (Fomba et al., 2013; Kandler et al., 2011; Müller et al., 2010) what is missing is reliable fractional solubility data (Baker and Croot, 2010) for all of the elements that represent dissolution in seawater and residence times for all of these elements in the surface ocean. The residence time section in the manuscript is very interesting and could be highlighted and compared more to other works from the same region (Croot et al., 2004; Dammshäuser et al., 2013; Jickells, 1995; Jickells, 1999) as presently this aspect of the work is under developed. In the present work the residence time for Al is most likely on the order of 3-6 months given the large inputs of Al from the Saharan dust or indeed it may be much shorter given work on iron in the same region (Croot et al., 2004).

**We agree with the reviewer. The estimation of atmospheric fluxes using dAl concentrations in the mixed layer is difficult to constrain. Especially in this region where seasonality plays a major role in controlling dAl concentrations in surface waters as observed from numerous north to south transects which display a large range in dAl concentrations (Menzel Barraqueta et al., 2019). Indeed, the residence time appears to be lower in this region as suggested for iron by Croot et al., 2004. In Menzel Barraqueta et al., 2019 they used an Al residence time of 1.25 years (extracted from Han et al., 2008) for the tropical Atlantic and calculated an atmospheric deposition flux well below modeling studies. However, assuming a residence time of 3 months did yield deposition fluxes that were equal, within uncertainty, to modeling studies.  First, we shortened this section as suggested in the reviewer's overview and then added some more detailed discussion about the residence time and solubility. The modified section reads as follows:**

**1.4.1    Atmospheric deposition**

Aluminum is present as a relatively constant fraction of ~8.15 wt% in the continental crust (Rudnick and Gao, 2006), is supplied to open ocean surface waters mainly by atmospheric deposition (Orians and Bruland, 1986) and is not considered to be taken up by phytoplankton (apart from a small amount being incorporated into siliceous diatom frustules; Gehlen et al., 2002). Therefore, dAl in the surface mixed layer is used as a tracer for atmospheric deposition to the surface ocean (Measures and Brown, 1996; Measures and Vink, 2000). The atmospheric input in the study region is mainly influenced by North African/Saharan mineral dust with only a small contribution of anthropogenic sources which differ greatly in TM composition and solubilities from mineral dust (Baker et al., 2013; Patey et al., 2015; Shelley et al., 2015). Close to continental shelves, Al can also be supplied by sediment resuspension in addition to atmospheric input (Menzel Barraqueta et al., 2018; Middag et al., 2012; Moran and Moore, 1991). Enhanced aerosol optical depth above our study region (Supplementary Fig. S3&4) indicates high dust loading at the time of our cruise.

Our dAl concentrations in surface water ranged between 30 and 49 nmol $L^{-1}$ and LpAl between 3.4 and 18.2 nmol $L^{-1}$. Dissolved Al concentrations decreased with depth (Fig. 8), indicating that Al was released by aeolian dust deposition to surface waters and removed through scavenging at depth (Orians and Bruland, 1985).

Dissolved atmospheric deposition fluxes can vary largely depending on the aerosol solubility, which is dependent on aerosol source, atmospheric aerosol processing during transport and dissolution in surface waters (Jickells, 1999). Here, atmospheric dFe fluxes were calculated using the dAl inventory in the surface mixed layer, a residence time of dAl of 0.65 ± 0.45 years as reported for the Canary Current System (Dammshäuser et al., 2011), and a ratio of 0.31 for dust derived dissolved Fe/Al (Buck et al., 2010). This approach is independent of the fractional solubility of Al, as we do not account for total atmospheric deposition fluxes, and only use the already dissolved fraction of Al. However, this approach is dependent on the ratio of Fe/Al from dissolution of aerosols. This ratio, however, is not clearly defined and can vary between different dust sources and deposition pathways, such as wet or dry deposition (e.g. Shelley et al., 2018). In our study region, dry deposition is the dominant deposition pathway, as it is located north of the Intertropical Convergence Zone and precipitation is minimal < 0.001 $g/cm^{-3}$ (NASA). Here, we utilized a ratio observed for total aerosol samples in the remote North Atlantic from a Saharan dust source (Buck et al. 2010). Soluble ratios under the Saharan dust plume were however lower for all leach media (Fe/Al: 0.051–0.25; Shelley et al. 2018), indicating that the ratio of 0.31 utilized here, might result in an overestimation of the dFe flux estimates. This approach also assumes that dAl is only supplied to the surface ocean via atmospheric deposition. Vertical fluxes of Al from sediment resuspension are unlikely to contribute significantly to concentrations of dAl in surface waters here as dAl concentrations decreased with depth, indicating removal of dAl via scavenging.

Mean atmospheric dFe fluxes of the individual stations were 0.63–1.4 $\mu mol\ m^{-2}\ d^{-1}$ (Fig. 5, Supplementary Table S2), values similar to reported fluxes close to our study region of 2.12 $\mu mol\ m^{-2}\ d^{-1}$ further north between 22.5–25°N and 26.5–27.5°W (Rijkenberg et al., 2012) and 0.120 $\mu mol\ m^{-2}\ d^{-1}$ around 20°N close to the African coast (Ussher et al., 2013). The uncertainty in the residence time of dAl, however, creates a large uncertainty in calculated fluxes resulting in a lowest flux of 0.37 $\mu mol\ m^{-2}\ d^{-1}$ when using the largest estimated residence time of 1.1 years and a highest flux of 4.65 $\mu mol\ m^{-2}\ d^{-1}$ when using the shortest estimated residence time of 0.2 years. In fact, a residence time of 3 months has been shown to give similar results for total Al atmospheric deposition fluxes as modeling studies (Menzel Barraquetta et al., 2019). Low residence times of a few months have also been suggested for Al and Fe in areas with a large dust deposition including our study region (e.g. Croot et al. 2004, Dammshäuser et al., 2011). Therefore, we suggest that the atmospheric dFe flux is more likely to be closer to the upper range of our flux estimates. However, the atmospheric deposition fluxes using a short residence time may be larger than the annual average since the dust load is highest between June and August in our study area (Supplementary Fig. S4).

In the present work the atmospheric deposition is not well presented as it does not discuss the role of aerosol solubility in the flux calculations and does not distinguish categorically between the total aerosol flux and what it terms is the dissolved flux – which as the deposition is predominantly dry means that it is metals that are solubilized post deposition. It would help to include at the very least satellite data on the atmospheric aerosol loading to pinpoint if there was dust deposition immediately before or during the occupation of these stations.

**The definition of dFe flux is not very clear in the manuscript at the moment and we added a few changes to clarify this (see changes above). Indeed, fractional solubilities of Al and Fe after deposition are key to the calculations. The calculation using seawater dAl concentrations doesn't require the absolute fractional solubility value, but the ratio of Fe/Al from dust dissolution, which can vary dependant on e.g. dust source, atmospheric processing, and between wet and dry deposition. We added some more detail on this uncertainty to the flux description in order to clarify this. We also added a plot showing satellite data related to dust loading, averaged for the month of our cruise (June 2014) and a plot showing monthly averaged dust loading for the cruise region over 24 months, both in Supplementary Information.**

**Added plots:**

[Figure]

**Figure S3.** Time averaged map of aerosol optical depth 550 nm (dark target), monthly 1 deg (MODIS-Terra MOD08_M3 v6.1) for the month of June 2014.

[Figure]

**Figure S4.** Area-averaged time series of aerosol optical depth 550 nm (dark target), monthly 1 deg (MODIS-Terra MOD08_M3 v6.1). Time series for January 2013 to December 2014 for the region between 19.5W, 17.5N, 16.5W and 18.5N.

Sections 3.5 and 3.6: These sections are long and don't add that much to the paper at present, as they read like an earlier draft of a PhD thesis and they should be shortened and tightly focused on the main points that are supported by the data. The central problem with these sections is that two repeat stations are interpreted mostly in the context of O$_2$ and not simply resuspension.

**We acknowledge the reviewers opinion on the length of these sections, but we think that these two sections are very important for the manuscripts findings as they evaluate the role of scavenging and the influence of oxygen and sediment resuspension or dust deposition on the temporal variability of the trace metal concentrations. Additionally, the section "3.6.2 Atmospheric dust deposition and sediment resuspension" discusses the possibility of sediment resuspension by comparing Al data from the repeated stations.**

Geochemical data from the NW African sediments: I was a little surprised that there was little discussion of the geochemistry of the sediments and then the relationship between the dust and water column compositions. A further paper that could be included is that of Itambi et al. (2010) who examined the magnetic mineral inventory of the sediments off Senegal.

**We added a brief description of the geochemistry of the sediments in our study region at Page 11, after Line 23:**

The sediments in the study area contain a large amount of carbonate, biogenic silica and quartz (Hartman et al. 1976). The fraction of sand and mud varies largely depending on bottom depth, with sand comprising between 7 and 70% of the dry weight (Dale et al., 2014). The particulate organic carbon (POC) content varies between 0.55 wt% at shallow depth (66 and 90 m) and increases to 3.3 wt% at 1108 m depth (Schroller-Lomnitz et al., 2019). A more detailed description of the sediments underlying our study region and sediment parameters collected on the same cruise, including Fe(II) concentrations and Fe/Al ratios, are given in Schroller-Lomnitz et al. (2019).

**We also thank the reviewer for providing the additional reference. However, as this reference relates to much longer time scales, which might not be relevant for our study, and as it is in a different location at a much greater water depth (Depths 3500 m), we feel this is probably not a necessary addition.**

Other particle data for the North Atlantic: I was surprised not to see any comparison to the particle data of Kuss and coworkers (Kuss and Kremling, 1999a, b; Kuss et al., 2010; Scholten et al., 2001; Waniek et al., 2005) as while it is only near surface data it does cover a similar transect and include time series data from sediment traps. There is also other recent trap data from close to the Canaries which gives more seasonal data (Brust et al., 2011; Brust and Waniek, 2010).

**The particle data in the listed references were all quite far from our study region and were likely to receive atmospheric inputs in different magnitudes and from different sources. Our location was much closer to the coast, which has a likely influence on the particle composition. We only have the leachable particulate fraction in our study here and no total particulate data and therefore we do not see a benefit from including such a comparison.**

Specific Comments:

P1 Line 20: What evidence is provided here that dust deposition did not play a role?

**The assumption that dust deposition 'only played a minor role' on the differences between the two deployments was based on the determination of dissolved and leachable particulate Al concentrations at the repeated stations, which do not show much difference between the two deployments (see section 3.6.2.). This does not mean that trace metals are not also supplied by atmospheric deposition, but didn't seem to be the driver of the variability at repeated stations.**

**The sentence in the Abstract was modified as follows:**

Variations in organic matter remineralization and lithogenic inputs (atmospheric deposition or sediment resuspension; assessed using Al as indicator for lithogenic inputs) only played a minor role in redox-sensitive TM variability.

P1 Line 23: How were the DFe atmospheric fluxes estimated? As it is most likely dry deposition so how was the dissolution assessed?

**See response to general comments above. We used dAl inventories, which comprises the already dissolved fraction and applied a conversion by using a ratio of Fe/Al from atmospheric deposition which was previously obtained for aerosol samples in a nearby region (Buck et al. 2010).**

**We added the following information to the sentence:**

Vertical dFe fluxes from $O_2$-depleted subsurface to surface waters (0.08–13.5 µmol $m^{-2}$ $d^{-1}$) driven by turbulent mixing and vertical advection were an order of magnitude larger than atmospheric

deposition fluxes (0.63–1.43 µmol m$^{-2}$ d$^{-1}$; estimated using dAl inventories in the surface mixed layer) in the continental slope and shelf region.

**Also see changes to made to the discussion section for atmospheric deposition fluxes above.**

P5 Line 19: What was the purpose of the 2nd Mn cartridge if it was not analysed? (P6, Line 24)

**Both Mn cartidges were used for studies on Th-234. The $^{224}$Ra/$^{223}$Ra ratios shown here, are only determined from one cartridge. We added the following information:**

In this work, $^{224}$Ra/$^{223}$Ra ratios are shown, which were analyzed from the first Mn cartridge.

P5 Line 20: Please include information on the flow rate through the pumps.

**We added the following information to the sentence:**

The pumped water volumes varied between 1000 L and 1700 L and flow rates were 10–15 L min$^{-1}$.

P6 Line 19: The reagent is Lumogallion not Lumogallium.

**Thank you. We corrected this in the text.**

P6 Line 20: Lumogallion

**Corrected in text.**

P6 Line 24: Please include information about the methodology that was used for these measurements and the basic operating conditions.

**We extended the methods description accordingly, which now reads as follows:**

2.3 Aluminum measurements

Aluminum concentrations were determined in surface water samples for all stations along the transect and at two stations (3 and 8) for the entire water column. Samples were analyzed for Al using the batch lumogallion method (Hydes and Liss, 1976). Acidified samples were buffered manually with a 2 M ammonium acetate buffer (Romil, UpA) to a pH between 5.1 and 5.2. The buffer was prepared using ammonium hydroxide (Romil, UpA) and acetic acid (Romil, UpA) in de-ionized water (MilliQ, Millipore). Buffered samples were spiked with a 2 mg L$^{-1}$ lumogallion (TCI) solution allowing the complexing agent to be in excess. The lumogallion solution was prepared in 2 M ammonium acetate buffer (Romil, UpA). After spiking, samples were heated up for 1.5 h at 80°C in an oven (Heratherm, Thermo Scientific) and left to cool down overnight at room temperature to allow the formation of a fluorescent Al complex. Samples were measured using a fluorescence spectrophotometer (Cary Eclipse, Agilent). The samples were measured with an excitation and emission wavelength of 465 and 555 nm, respectively. The excitation and emission slits were set to 10. The plastic cuvettes used for the measurements were pre-cleaned in a 2 M HCl (Trace metal grade, Fisher) for at least 24 h. In between samples, the cuvette was thoroughly rinsed with de-ionized water followed by actual sample. The same cuvette was used during an analytical session. All samples were analyzed in duplicate and the concentrations calculated from the peak heights via standard addition. Samples and reagent natural fluorescence was monitored by analyzing their content in the absence of the complexing agent. The standards were prepared in low trace metal seawater from a 500 nmol L$^{-1}$ stock standard solution prepared from a 1000 ppm Al standard solution (Merck Millipore). A typical calibration had the following standard concentrations: 0, 10, 20, 40, and 60 nmol L$^{-1}$. GEOTRACES reference seaweater (GS) was run with a mean average Al value of 27.76 ± 0.17 nmol L$^{-1}$ (n=4; consensus value 28.2 ± 0.2 nmol L$^{-1}$).

P6 Line 30: The iodide data is not presented so the reader can not assess what the relevance of this parameter is. For instance, was the iodide concentration ever more than what might be expected for total iodine at the salinity of the seawater as this would be a good indicator of excess iodine from a benthic source.

**Iodide concentrations were enhanced within the oxygen minimum zone but not to high values as observed in the Peruvian OMZ and they were not higher than would be expected for total iodine. Therefore we do not see evidence for a benthic source. However, we acknowledge that it would be of interest to show the spatial distribution and added the following figure of the Iodide distribution in Supplementary Material:**

[Figure]

**Figure S1.** Spatial distributions of Iodide across the Mauritanian shelf at 18°20'N in June 2014. Each sample location is indicated as black dot and oxygen contours at 50 μmol kg$^{-1}$ enclosing the upper and lower OMZ are displayed as black contour lines.

P7 Line 4: Were the Turbidity and Chlorophyll sensors calibrated during this expedition? Was the Turbidity sensor zeroed between CTD casts?

**For the turbidity data used in the manuscript, we used the Wetlab sensor output corrected by the calibration coefficients provided by the manufacturer. No further calibration was applied. Further, the turbidity sensor was not zeroed between individual CTD casts. However, we did not observe a drift of the turbidity readings. Instead, a constant offset of 0.12 NTU (nephelometric turbidity unit) was distinct in the data throughout the cruise. To make the calibration procedure more clear, we added:**
Turbidity and chlorophyll *a* were measured with a combined Wetlabs turbidity and fluorescence sensor that was attached to the CTD. The output of both sensors was corrected using the calibration provided by the manufacturer.
**See also comment below.**

P7 line 9: So the decay was assumed linear for all stations, no matter the depth?

**We assume the reviewer is referring to the oxygen sensor calibration here. The sentence in the manuscript (page 7 line 7-8) stated: "The O$_2$ calibration was undertaken using a linear fit with respect to O$_2$ concentration, temperature, and pressure."**
**For calibration of the Seabird oxygen sensor, we first use standard Seabird processing during which the sensor calibration coefficients provided by the manufacturer are applied. The output of the Seabird processing is oxygen concentration in μmol/kg. In a second processing step, we fit the oxygen sensor data to the oxygen concentration measured from water samples on board the cruise via Winkler titration. This is done by minimize the differences between the Winkler titration samples from the bottle data and the respective oxygen sensor data using linear functions for pressure (i.e. depth), temperature and oxygen concentrations itself. For some oxygen sensors, a quadratic fit in pressure (i.e. depth) works better than a linear fit. This, however, was not the case**

**for the oxygen sensors (dual Seabird CTD system) used during the M107 cruise. The uncertainty of this calibration is provided by the standard deviation between the samples and the sensor readings after applying the calibration. We mention this uncertainty in the text. The oxygen sensors are typically calibrated at Seabird laboratories annually or every second year, depending on the duration the sensor was used during cruises.**

**These details above are given in the cruise report of the M107 cruise. To keep this manuscript concise, we replaced the sentence (page 7 line7-8) by the following text:**

Oxygen sensor data was initially processed using calibration coefficients provided by the manufacturer. Subsequently, $O_2$ sensor data were fitted to the $O_2$ concentrations determined by the Winkler titration method using linear functions for temperature, $O_2$ and pressure (i.e. depth).

P7 Line 17: How was the tap-water assessed to be Ra free? Was it provided from the ship's system as they can be notorious for Ra contamination. More details need to be provided here.

**The following changes were made to the manuscript (Page 7, Line 17) to clarify this:**

On-board the ship the Mn-cartridges and Mn-fibers were washed with Ra-free tap water to remove any residual sea salt and particles. Ra was removed from the tap water by passing it through a Mn-fiber filled cartridge. Afterwards, both cartridges and fibers were partially dried with filtered compressed air to remove excess water.

P7 Line 24: How was the $^{228}$Th assessed? Was it only from the background counts after 3 weeks when the initial $^{224}$Ra had decayed?

**Yes. Th-228 was measured 3 weeks after the first measurement of $^{224}$Ra. The following sentence was added at Page 7 Line 24 to make this clear to the reader:**

The $^{228}$Th activity was measured three weeks after the first measurement of $^{224}$Ra, when the initial $^{224}$Ra had decayed.

P8 Line 2: Concentration difference between which levels? At times it seems it is between 8-29 m (P 8, Line 8) and at other times it seems it is over multiple levels (P9, Line 15)? This section of the methodology needs to be better explained.

**Indeed, the text was unclear, thank you. We changed the lines formerly on page 8, line 8 to:**
Average advective and diffusive TM fluxes were calculated for a depth interval from the shallow $O_2$-depleted waters to surface waters. The exact depth interval varied for each station (see Table S2) due to differences in the depths where TM samples were collected. The upper depth (8–29 m) was always in layers with enhanced chlorophyll *a* fluorescence, although for some stations the upper depth was below the surface mixed layer.

**The discussion on former page 9, line 15 was changed to:**
Ideally, the vertical length scale over which the concentration difference is determined can be diagnosed as the TM concentration variance divided by its mean vertical gradient (e.g. Hayes et al., 1991). However, in our study TM concentration time series data are not available. Previous studies have used a vertical length scale of 20 m to calculate the concentration differences between the target depth and the water below (e.g. Hayes et al., 1991; Steinfeldt et al., 2015; Tanhua and Liu, 2015). For our calculations, we chose to use a smaller length scale of 10 m following Hayes et al. (1991) which results in vertical advective TM flux presumably on the lower side of possible values.

P8 Line 4: How valid is it to ignore lateral fluxes in a region where there are strong filaments and cross shelf transport (Fischer et al., 2009; Gabric et al., 1993; Klenz et al., 2018; Rees et al., 2011; Schafstall et al., 2010)? Indeed the recent paper by Klenz et al. (2018) shows that at this time there are very high current velocities along the 18 ° N line.

**Thank you for this comment. Apparently, our description here is unclear. In fact, the vertical fluxes determined in the manuscript do not depend on lateral fluxes. With the sentence in question "The equation is solved by vertically integrating the tracer transport budget equation between two vertical layers while ignoring lateral fluxes, changes of w with depth and assuming steady state." (page 8 lines 3-5) we wanted to express how the two vertical fluxes, vertical advection and vertical diffusion, can be derived from the transport budget equation which is fundamental to the analysis (e.g. Brandt et al., 2010). Transport budget equations balance advective and diffusive flux divergences. As many studies determine transport budgets for the mixed-layer, fluxes instead of flux divergences are used in the vertical component (e.g. Schafstall et al., 2010; Rhein et al., 2010; Kock et al., 2012; Steinfeldt et al., 2015; Tanhua and Liu, 2015). However, fundamentally all these studies solve**

$$\frac{\partial C}{\partial t} = -u\frac{\partial C}{\partial x} - v\frac{\partial C}{\partial y} - w\frac{\partial C}{\partial z} + \frac{\partial}{\partial x}K_x\frac{\partial C}{\partial x} + \frac{\partial}{\partial y}K_y\frac{\partial C}{\partial y} + \frac{\partial}{\partial z}K_z\frac{\partial C}{\partial z} - S^- + S^+$$

**Where $S^-$ are sinks and $S^+$ are sources of solute concentration C. Expressions for the vertical advective flux and the vertical diffusive flux as used in the manuscript (and the other studies mentioned above) can be derived by: "The equation is solved by vertically integrating the tracer transport budget equation between two vertical layers while ignoring lateral fluxes, changes of w with depth and assuming steady state." as we had stated in the text in the earlier version. However, judging from the reviewer's remark, the sentence does not advance understanding of the nature of the fluxes determined in our manuscript and elsewhere, but is unclear instead. We thus decided to delete the sentence.**

P8 Line 14: How close to the bottom did the microstructure profiles get to? Typically they do not go into the benthic boundary layer for fear of doing damage to the probes if they hit the sediment. Information on how close to the bottom the sensors got would be extremely helpful in assessing whether or not the fluxes from the sediment were well constrained. Similar information for the trace metal sampling should also be supplied.

**We run the microstructure probe all the way to bottom and thus obtain full depth profiles of the dissipation rate of turbulent kinetic energy. The profiler is not damaged when impinging on the bottom. However, we are calculating fluxes in the upper water column between 20 m and 50 m depth. Neither near-bottom turbulence data nor TM data are needed for determining the diffusive and advective fluxes presented here.**

P9 Line 5: I am missing a step here as this website seems to provide only wind speed data, so how was the alongshore wind stress calculated? The number seems reasonable given global compilations (Varela et al., 2015) but what are the uncertainties? There is a NOAA site that calculates everything https://www.pfeg.noaa.gov/products/PFEL/modeled/indices/upwelling/NA/how_computed.html

**The wind stress was calculated from daily ASCAT wind measurements from 18°22.5'N, 016°22.5°W using $\tau_y = \rho_{air}C_d v^2$ where $v$ represents alongshore wind, $C_d$ is drag coefficient for which 1.15x10$^{-3}$ was used (e.g. Fairall et al., 2003) and $\rho_{air}$ is density of air. The value of 0.057Nm$^{-2}$ represents an average of all wind stress values that were determined from individual ASCAT wind velocities available for June 2014. To provide this information to the reader, we added:** (0.057 Nm$^{-2}$, determined from daily winds from Remote Sensing Systems ASCAT C-2015, version v02.1 (Ricciardulli and Wentz, 2016) at 18°22.5'N, 016°7.5°W using $\tau_y = \rho_{air}C_d v^2$, where $v$ represents alongshore wind, $C_d$ is drag coefficient for which 1.15x10$^{-3}$ was used (e.g. Fairall et al., 2003) and $\rho_{air}$ is density of air)

**As stated above, differences between land stations and ASCAT winds in the order of 20% were found by Ndoye et al. (2014). However, as further detailed above, other uncertainties dominate the errors in our flux calculations.**

P9 line 5: How was the wind close to the coast assessed? For ASCAT wind velocities closer than ~70 km (25-km products) or ~35 km (12.5-km products) from the coast are flagged because of land contamination. This is due to the fact that - in the case of the 12.5-km product - backscatter measurements (σ0) of up to 35 km away from each WVC centre are used in the spatial averaging. It would appear that station 4 would be too close to the coast to use an ASCAT wind product, so some explanation needs to be provided here.

**The wind product used in this study is described above. The used product "Remote Sensing Systems ASCAT C-2015, version v02.1" includes the 12.5 Km ASCAT L1B backscatter data from EUMETSAT. We use the wind values from the position 18°22.5'N, 016°22.5'W which are not flagged. The position is located about 35km away from shore. Our methodology to determine vertical velocities follows Steinfeldt et al. (2015), who used the first available ASCAT satellite wind value along with the Gill parameterization. We could have also used wind data measured by the research vessel (that compared quite well to the satellite derived winds) but refrained from doing so because the Gill paramerization was validated using satellite winds. As stated before, we now include a discussion of the uncertainties of our flux determinations.**

P9 Line 6: These estimates of the upwelling velocity contradict the findings of Tanhua and Liu (2015) and others (Cropper et al., 2014), who reported that there was no upwelling over the summer, the same season when this expedition took place. The upwelling flux estimates listed here are also close to the maximum rates estimated using tracers by those authors for periods when the upwelling is active so some explanation is required here – see the general comment above regarding this.

**Cropper et al. (2014) discuss upwelling indices and SST gradients averaged between 12° and 19°N. Our analysis is located at the northern boundary of their box. Differences between their and our analysis most likely originates from regional differences is thus not mentioned here. Further, Tanhua and Liu (2015) used data collected in between July 12 and August 6, 2006, one month later than the data reported here. Again, we refrain from a discussion because we believe it would leave the reader more confused. We added some reference to the Tanhua and Liu study and the seasonality of upwelling to section 3.1.**

P9 Line 12-17: The explanation for the flux estimate being made here needs to be better described, as for most readers usually the vertical advective flux would be the velocity times the concentration and would not involve the gradient in the concentration as shown on line 17 here. It needs to be explained then why you use the gradient approach and which way around the gradient is (e.g. based on the equation as it stands you could have upwelling but still have a net negative flux from the surface waters if the metal concentration gradient is negative, this would happen when there was a higher metal concentration in the surface waters than below).

**We eliminated the gradient expression equation in the revised version of the manuscript and instead simply refer to concentration differences. We also simplified the description of the approach and now state:**

Ideally, the vertical length scale over which the concentration difference is determined can be diagnosed as the TM concentration variance divided by its mean vertical gradient (e.g. Hayes et al., 1991). However, in our study TM concentration time series data are not available. Previous studies have used a vertical length scale of 20 m to calculate the concentration differences between the target depth and the water below (e.g. Hayes et al., 1991; Steinfeldt et al., 2015; Tanhua and Liu,

2015). For our calculations, we chose to use a smaller length scale of 10 m following Hayes et al. (1991) which results in vertical advective TM flux presumably on the lower side of possible values.

P10 Line 8: So horizontal advection is at its greatest in summer, but horizontal fluxes are not considered in interpreting the trace metal and $O_2$ data?

**In the manuscript, we discuss the potential origin of elevated trace metal concentrations from the shelf at the transect and from the shelf further south. We also mention a potential influence of changes in the residence time of the water masses on the shelf as a factor which influences trace metal concentrations. In order to highlight this further we added some discussion about a change in source waters and local oxygen respiration for the repeated stations based on observations by Thomsen et al. (2018). See answer to comment below.**

P11 Line 2: Though it should be noted they are an order of magnitude or more lower than the Celtic Sea and the St Lawrence seaway.

**For clarification, we added this information to the sentence:**

Vertical fluxes of nutrients driven by mixing processes are amongst the largest reported in literature, however lower than in the Celtic Sea (Tweddle et al., 2013) and the lower St. Lawrence Estuary (Cyr et al., 2015).

P11 Line 21: Other studies in the same region have pointed to the role of CDOM and reactive oxygen species in potentially controlling metal distributions (Heller et al., 2016; Wuttig et al., 2013). Indeed Heller et al. (2016) observed changes in FDOM consistent that could be consistent with microbial respiration of resuspended sediment material, a process also invoked by Thomsen et al. (2018) to explain low oxygen patches in these waters.

**Reactive oxygen species and CDOM might in fact have an influence on trace metal concentrations in surface waters and maybe also in the deep ocean. But this field is not well understood yet and we don't feel like we can add anything that helps to better understand this interaction or helps interpreting our data. The influence of CDOM is particularly unclear as it is involved in the production of superoxide in the surface ocean during photoreduction (Micinski et al. 1993), but CDOM may also be a sink of superoxide (Heller et al., 2016). Furthermore, in the ETNA Fe appeared to be relative inert to the reaction with superoxide ($O_2^-$) even though this element showed the most pronounced changes coinciding with $O_2$ changes, (Wuttig et al., 2013).**

P11 Line 17: If Thomsen et al. (2018) indicate this was from horizontal advection then this would invalidate the approach for estimating the vertical fluxes – see the general comment on this above.

**See answer to general comment.**

P12 Line 4: The Hawco et al. (2016) paper is for the Pacific and a much more $O_2$ depleted water column with sulfidic sediments so a very different environment.

**We removed the Hawco et al. (2016) reference in this sentence.**

P12 Line 8: The values are also similar to values found by Wuttig et al. (2013) for the Tropical North Atlantic and near to coastal West Africa.

**We appreciate pointing out this important reference for trace metal data in this region. We added the Wuttig et al. (2013) reference to this sentence. The sampled locations have a greater distance from the coast and bottom depth than our coastal stations and are more comparable with our stations further offshore where we observe a similar concentration range.**

P12 Line 14: Of direct relevance to this work is the result that oxygen has also been shown to have a strong impact on Co speciation in the Tropical North Atlantic (Baars and Croot, 2015). This work should therefore be included in the discussion of the present data set.

**Thanks for pointing out this important reference here. We have added the reference to the sentence on P12 Line 4 and also used it in section 3.6.1 in order to discuss the dCo variability between repeated stations. The following was added to the paragraph starting at Page 23 Line 17:**

This is in accordance with previously observed correlation of dCo with $PO_4^{3-}$ in addition to $O_2$ (Baars and Croot, 2015; Saito et al., 2017). However, we observed a very low Pearson correlation of dCo with $PO_4$ of only 0.15 compared to oxygen (-0.58) (Supplementary Table S1) below 50 m water depth, suggesting a stronger influence of oxygen than remineralization on the overall distribution of dCo for our study area.

P12 Line 26: Mn data is also found in the work of Wuttig et al. (2013) and this work is again of direct relevance to the current paper with regard to the relevance of reactions with oxygen.

**We also added this reference to this paragraph.**

P12 Line 35: Please define LOQ here and what its value is.

**We added the definition of LOQ the first time used in this section, which is on Page 11 Line 30, as follows:**

In contrast, LpCo concentrations varied between below limit of quantification (LOQ) and 0.179 nmol $L^{-1}$ and were generally highest in surface waters and close to the coast (Fig. 3d).

**The Limit of Quantification is also defined in the methods section including how it was calculated. It is not possible to state a single value here, as we quantified the LOQ individually for each station and depth using the percentage uncertainty of the dissolved and total dissolvable concentration, as both are used to quantify the leachable particulate concentration.**

P13 Line 24: Is there any evidence for sulfide release from the sediments in this region (see the general comment on this above also)?

**Sulfide was present at around 10 cm sediment depth for water depth of 47 and 90 m, whereas sulfate reduction was observed close to the sediment surface down to 236 m water depth (Gier et al. 2017). Also see answer to general comment above.**

P13 Line 36: Did you encounter MOW in this work?

**No, we did not encounter MOW here.**

P14 Line 27: It is worth noting here that dilution does not impact the ratio of the two radium isotopes. Though the assumption is that there is only a single uniform benthic source.

**We added this explanation to the description of the use of $^{224}Ra/^{223}Ra$ ratios in the paragraph above at Page 14 Line 23:**

The ratio $^{224}Ra/^{223}Ra$ is not affected by dilution assuming there is no mixing with waters having significantly different $^{224}Ra/^{223}Ra$ ratios.

P14 Line 29: Radium is conservative the other elements are not – resuspension and scavenging are the most likely controls as the $O_2$ concentration is not low enough to impact iron or manganese.

**See answer to general comment on the influence of oxygen concentrations. Scavenging processes may be influenced by oxygen concentrations and sediment release may be enhanced under lower $O_2$.**

P15 Line 7: This location is not on the 18° N transect so is it a source region to the south which is then advected north as you suggest so again the horizontal fluxes seem to dominant here.

**Yes, there are horizontal fluxes, but unfortunately we are unable to quantify them in the present study.**

P15 Line 15: How was the PCA performed? Were the data all normalized to have a mean of 0 and a standard deviation of 1 before analysis? P15 Line 18: $O_2$ and AOU are linearly related so they would be expected to be orthogonal to each other in the PCA – this is ok though as only the PCA components need to be independent.

**The PCA data are scaled to unit variance. In the original figure legend we state: Loadings/scores have been scaled symmetrically by square root of the eigenvalue. For full details we refer the reviewer and readers to the R documentation for the PCA calculation function used (as described in the original manuscript).**

P16 Line 7: What evidence is there that the sediments are anoxic? Sediment work performed along the same transect line shows no evidence for this (Gier et al., 2016). See the general comment on this above.

**See response to general comment: High Fe(II) values (~20–30 µM) were observed within a few cm of sediment indicating reductive dissolution is occurring (Gier et al., 2017; Schroller-Lomnitz et al., 2018).**

P16 Line 24: This assumes that all of the dissolved iron is from remineralization and not from dust dissolution (Baker et al., 2013; Baker and Croot, 2010; Baker et al., 2006a; Baker and Jickells, 2006; Baker and Jickells, 2017; Baker et al., 2006b), given the proximity to the Sahara and the impact it is on Aluminum concentrations this seems a contradiction. There is also the potential for direct remineralization of Saharan dust by zooplankton (Barbeau et al., 2001; Barbeau and Moffett, 2000; Barbeau et al., 1996; Laglera et al., 2017; Schmidt et al., 2016).

**Yes, this assumes that the utilized offshore reference values from previous publication are mainly influenced by remineralization. There is a large uncertainty in the calculation and therefore we are using the full range of lowest and highest dFe/C ratios. This is just a crude estimate. We already emphasize this in the text though: "These offshore ratios may still be influenced by an atmospheric source of dFe, which would result in an overestimation of dFe/C ratios from remineralization and thereby an overestimation of the fraction of remineralized dFe"**

P17 Section 3.4.2 – See the general comment about the atmospheric fluxes.

**Answered above**

P19 Line 6: See the general comment above regarding residence times.

**Answered above**

P19 Line 32: This would indicate it is being scavenged. There is no discussion of the role of scavenging in this manuscript after the introduction. Surely this deserves some attention given it is likely a major control on dissolved metal concentrations.

**We modified the sentence accordingly to highlight the influence of scavenging as follows:**

Vertical fluxes of Al from sediment resuspension are unlikely to contribute significantly to concentrations of dAl in surface waters here as dAl concentrations decreased with depth, indicating removal of dAl via scavenging.

**Also see response to general comments above.**

P20 Line 11: Closest to the shelf is also likely to be the most impacted by the ASCAT limitation near the coast so it will have to be explained if this value is far enough away from the coast. The station 4 advective flux is also an order of magnitude more than any of the other stations so some indication of why this is needs to be supplied as the reader does not see the distribution at station 4.

**The parameterization from Gill uses a single wind stress value close to the coast. No offshore wind data is used. The vertical velocities away from the coast are determined by the x-dependence (exponential decay). We refer the reviewer to equation (2).**

P20 Line 16: Has anyone checked the sea surface height data? Were there any mesoscale eddies around at this time (Karstensen et al., 2015)?

**SSH, SST and shipboard velocity data did not show mesoscale eddies in the study area. In addition the analyses of Schütte et al. (2016) showed that they are mainly generated at the eastern boundary during boreal summer, whereas our cruise was in June.**

P20 Line 32: Though most of the sediment is derived from Saharan dust and biogenic material, so you can't escape the dust entirely as a source.

**Yes, Saharan dust can indirectly be a source of trace metal after being deposited to the seafloor and then undergoing reductive and non-reductive dissolution. In this section we only compare the direct atmospheric fluxes to the vertical fluxes from below, not discussing where the trace metals supplied by vertical fluxes originated from, which could be from a benthic source potentially influenced by dust deposition to the seafloor. We agree that it is important to mention the influence of dust derived trace metals on sediment release somewhere in the manuscript.**

**Therefore, we added the following statement to the Conclusion part of the manuscript (P25 Line9):**

However, deposition of atmospheric dust is a source of Fe to sediments in our study region and consequently indirectly contributes to benthic released TMs.

P21 Line 15: What does an NTU represent here? It has not been defined in the paper anywhere. This could simply be background noise of an uncalibrated instrument as figure 7 seems to show 0.2 NTU most of the time.

**Thank you for pointing out that we did not properly introduce NTU. The values in NTU (nephelometric turbidity unit) refers to the unit of data output of the Wetlab turbidity sensor. It represents the proportion of white light scattered back to a transceiver by the particulate load in a body of water, represented on an arbitrary scale and referenced against measurements made in the laboratory by the manufacturer on aqueous suspensions of formazin beads.  For the turbidity data used in the manuscript, we used the Wetlab sensor output corrected by the calibration coefficients provided by the manufacturer. No further calibration was applied. The most frequent small turbidity value measured in the deep ocean and in large distance to the seafloor was 0.12 NTU. Standard deviation in this oceanic region was 0.01 NTU, suggesting that a value larger than 0.14 NTU reliably measure elevated turbidity in the water column.**
**To make this more clear to the reader, we added following sentences to the manuscript at Page 7, line 4 : "Turbidity ..." to the following:**

Turbidity and chlorophyll *a* were measured with a combined Wetlabs turbidity and fluorescence sensor that was attached to the CTD. The output of both sensors was corrected using the calibration provided by the manufacturer. Throughout this manuscript, turbidity data are presented in nephelometric turbidity units (NTU). The noise level of the sensor in our data set was found to be lower than 0.14 NTU.

P21 Line 19: Reductive dissolution is occurring in the oxygenated waters here?

**Thanks for pointing out the potential for misunderstanding this sentence. We meant from reductive dissolution occurring in the sediments representing the TM source. We changed the sentence to the following:**

Furthermore, offshore transport of acid-labile Fe particles formed by scavenging (oxidation/adsorption) of dissolved Fe originating from a benthic source was observed in the North Pacific (Lam and Bishop, 2008) and may contribute to the bioavailable Fe pool.

P22 Line 17: This is assuming that it is the same water mass being sampled, but early we are told that this is water that came from the south, so it clearly isn't and thus trying to argue that this is a temporal trend in the same water is flawed. At the very least you could at least show the data on the same density surfaces. See the general comment on this above.

**For the repeated stations below 50 m - where we observe the shift in trace metals - temperature vs salinity plots indicate that the water masses didn't change. The exact source of the water and the velocity of water mass transport may still be different even though the water mass properties seem to be the same. The change in oxygen at these locations discussed in Thomsen et al. (2019), were also described as not being caused by a change in water masses. These author argued that the percentage of South Atlantic Central Waters bringing in oxygen were not a reason for the shift in O$_2$, and it was likely that the change in oxygen concentrations could be explained by in-situ oxygen consumption/respiration processes. The temporal variability at this location is not attributed with a change in water masses although it may come from the south. Further offshore low oxygen anomalies were associated with locally increased SACW fractions (Thomsen et al., 2019). The SACW fraction was observed to stay relatively constant near the seabed. As stated above, we think the description in the original manuscript was misleading and adjusted the paragraph on Page 11, Line 17.**

**We also added these data plotted against density to the Supplementary Information.**

P25 Line 14: You have shown the variability in a shelf system but no evidence that this is controlled by O$_2$ at this location. The recent paper by Schlosser et al. (2018) did however show the impact of a sulfidic event on trace metal concentrations in the water column so that is an example already.

**See response to general comment. The influence of sulfidic event in an anoxic water column on trace metal distribution is a very important finding, but the influencing processes are very different from the influence of different levels of oxygen concentrations and therefore not directly comparable to our findings. In the study by Schlosser et al. (2018), no short-term variability in oxygen concentrations was observed and H$_2$S concentrations were also only determined for one deployment at one station.**

P26 Line 4: FW are not the initials of one of the authors, so we don't know who this is until we read later in the acknowledgements.

**Thanks for pointing this out. We added the full name of that person here.**

Figure 3: There is no iodide data shown, yet it was used in the PCA and Radium was not.

**We added a figure showing the spatial distribution of iodide to the Supplementary Material. See answer to comment on iodide above.**

Figure 3: The radium data shows considerable 'young waters' in the near surface offshore, some more explanation is needed for where this might have contacted the sediments last. Or is it an artefact of the contouring in R as it is hard to see the data points.

**The Radium data at this place shows indeed 'younger waters' and is not an artefact of the contouring. We describe in the original manuscript that this water has a likely origin from the shelf further south. We added a sentence referring to Thomsen at el. (2019), describing the contribution of SACW in this area. We are unable to provide more information about the origin of this water.**

Figure 6: suggest you use a different symbol for the leachable as when they overlap it is hard to tell which is which when they are all the same symbol.

**Thanks for this comment, to avoid confusion, we turned the symbol for all data points above 50 m into diamonds.**

[Figure]

**Figure 1.** (a) Dissolved against total dissolvable trace metal concentrations for Fe (left; red line: TDFe = 10*dFe), Mn (middle; purple line: TDMn = dMn) and Co (right; turquoise line: TDCo = dCo). (b) Fraction of leachable particulate trace metals (Lp/TD) against turbidity and (c) Leachable particulate

concentrations against turbidity for Fe (left), Mn (middle) and Co (right). Filled circles display all data points below 50 m depth, open diamonds at depths shallower than 50 m.

Figure 7: As above it would be easier to read if different symbols were used.

**Thanks again. We converted changed the symbols for the second deployment into triangles.**

[Figure]

**Figure 2.** Repeat stations: oxygen concentration, turbidity and dissolved trace metals (Fe, Mn and Co) and temperature vs salinity plots. First deployment displayed as solid line and circles and second deployment displayed as dashed line and triangles. (a) Station 3 (18.23°N, 16.52°W, 170 m water depth, 9 days between deployments). (b) Station 8 (18.22°N, 16.55°N, 189–238 m water depth, 2 days between deployments).

Figure 7: The oxygen and turbidity anomalies mentioned seem to be related as mentioned in the text but the turbidity is more gaussian shaped while the oxygen has a bimodal distribution. It would have been more useful to have plotted all of this in density space so that we could see if the water masses were similar as the T/S properties look quite different but the symbols are so large that this information is lost.

**We added a plot showing the density distribution of the different deployments to the Supplementary Material:**

[Figure]

**Figure S5.** Density plots for oxygen concentration, turbidity and dissolved trace metals (Fe, Mn and Co) for repeated profiles. First deployment displayed as solid line and second deployment displayed as dashed line. (a) Station 3 (18.23°N, 16.52°W, 170 m water depth, 9 days between deployments). (b) Station 8 (18.22°N, 16.55°N, 189–238 m water depth, 2 days between deployments).

Figure 8: Different symbols between deployments would help. For station 8 and the dissolved Cd it is impossible to see if the data in the deep is the same or there was only 1 time that you sampled the deep waters.

**We have also changed the symbols here accordingly.**

[Figure]

**Figure 3.** Depth profiles of dCd, PO₄, dAl and LpAl of repeat stations. First deployment displayed as solid black line and circles and second deployment displayed as dashed black line and triangles. Oxygen concentrations are indicated as blue solid line for the first deployment and dashed blue line for the second deployment. (a) Station 3 (18.23°N, 16.52°W, 170 m water depth, 9 days between deployments and (b) Station 8 (18.22°N, 16.55°N, 189–238 m water depth, 2 days between deployments).

Figure 8: Why the phosphate depletion at 150 m at station 8? It seems there is also data missing from above and below that point – any explanation why? Is this data point anomalous then as it is does not appear to be oceanographically consistent?

**We agree that it appears to outlie from the other data, but we have no explanation for this. There are no phosphate data missing from this profile, but agree this is unfortunate as this would allow a better evaluation of the reliability of this point.**

**We thank the reviewer again for their detailed treatment of our manuscript.**

**References:**

Baars, O., and Croot, P. L.: Dissolved cobalt speciation and reactivity in the eastern tropical North Atlantic, Mar Chem, 173, 310-319, https://doi.org/10.1016/j.marchem.2014.10.006, 2015.

Baker, A. R., Adams, C., Bell, T. G., Jickells, T. D., and Ganzeveld, L.: Estimation of atmospheric nutrient inputs to the Atlantic Ocean from 50°N to 50°S based on large-scale field sampling: Iron and other dust-associated elements, Global Biogeochem Cy, 27, 755-767, https://doi.org/10.1002/gbc.20062, 2013.

Balistrieri, L., Brewer, P. G., and Murray, J. W.: Scavenging residence times of trace metals and surface chemistry of sinking particles in the deep ocean, Deep Sea Res Part A. Oceanogr Res Pap, 28(2), 101-121, https://doi.org/10.1016/0198-0149(81)90085-6, 1981.

Bianchi, D., Weber, T. S., Kiko, R., and Deutsch, C.: Global niche of marine anaerobic metabolisms expanded by particle microenvironments, Nature Geoscience, 11(4), 263, 2018.

Brandt, P., V. Hormann, A. Körtzinger, M. Visbeck, G. Krahmann, L. Stramma, R. Lumpkin and C. Schmid: Changes in the ventilation of the oxygen minimum zone of the tropical North Atlantic, J Phys Oceanogr, 40, 1784–1801, https://doi.org/10.1175/2010JPO4301.1, 2010.

Buck, C. S., Landing, W. M., Resing, J. A., and Measures, C. I.: The solubility and deposition of aerosol Fe and other trace elements in the North Atlantic Ocean: Observations from the A16N CLIVAR/CO2 repeat hydrography section, Mar Chem, 120, 57-70, https://doi.org/10.1016/j.marchem.2008.08.003, 2010.

Capet, X. J., Marchesiello, P., and McWilliams, J. C.: Upwelling response to coastal wind profiles, Geophys Res Lett, 31, L13311, https://doi.org/10.1029/2004GL020123, 2004.

Croot, P. L., Streu, P., and Baker, A. R.: Short residence time for iron in surface seawater impacted by atmospheric dry deposition from Saharan dust events, Geophys Res Lett, 31(23), https://doi.org/10.1029/2004GL020153, 2004.

Cropper, T. E., Hanna, E., and Bigg, G. R.: Spatial and temporal seasonal trends in coastal upwelling off Northwest Africa, 1981–2012. Deep Sea Research Part I: Oceanographic Research Papers, 86, 94-111, https://doi.org/10.1016/j.dsr.2014.01.007, 2014.

Cyr, F., Bourgault, D., Galbraith, P. S., and Gosselin, M.: Turbulent nitrate fluxes in the Lower St. Lawrence Estuary, Canada, J Geophys Res-Oceans, 120, 2308-2330, https://doi.org/10.1002/2014jc010272, 2015.

Dammshäuser, A., Wagener, T., and Croot, P. L.: Surface water dissolved aluminum and titanium: Tracers for specific time scales of dust deposition to the Atlantic?, Geophys Res Lett, 38, L24601, https://doi.org/10.1029/2011gl049847, 2011.

Desbiolles, F., Blanke, B., and Bentamy, A.: Short-term upwelling events at the western African coast related to synoptic atmospheric structures as derived from satellite observations, J Geophys Res-Oceans, 119, 461-483, https://doi.org/10.1002/2013JC009278, 2014.

Desbiolles, F., B. Blanke, A. Bentamy and C. Roy (2016), Response of the Southern Benguela upwelling system to fine-scale modifications of the coastal wind. Journal of Marine Systems, 156, 46-55.

Fairall, C.W., Bradley, E.F., Hare, J.E., Grachev, A.A., and Edson, J.B.: Bulk Parameterization of Air–Sea Fluxes: Updates and Verification for the COARE Algorithm. J Climate, 16, 571–591, https://doi.org/10.1175/1520-0442(2003)016<0571:BPOASF>2.0.CO;2, 2003.

Gehlen, M., Beck, L., Calas, G., Flank, A. M., Van Bennekom, A. J., and Van Beusekom, J. E. E.: Unraveling the atomic structure of biogenic silica: Evidence of the structural association of Al and Si in diatom frustules, Geochim Cosmochim Ac, 66, 1601-1609, https://doi.org/10.1016/S0016-7037(01)00877-8, 2002.

Gier, J., Löscher, C. R., Dale, A. W., Sommer, S., Lomnitz, U., and Treude, T.: Benthic dinitrogen fixation traversing the oxygen minimum zone off Mauritania (NW Africa), Frontiers in Marine Science, 4, 390, https://doi.org/10.3389/fmars.2017.00390, 2017.

Gill, A.: Atmosphere-Ocean Dynamics, Academic Press, California, 1982.

Han, Q., Moore, J. K., Zender, C., Measures, C., and Hydes, D.: Constraining oceanic dust 760 deposition using surface ocean dissolved Al, Global Biogeochemical Cycles, 22(2), 2008.

Hayes, S. P., Chang, P., and McPhaden, M. J.: Variability of the sea surface temperature in the eastern equatorial Pacific during 1986–1988, J Geophys Res, 96, 10553-10566, https://doi.org/10.1029/91JC00942, 1991.

Heller, M. I., Wuttig, K., & Croot, P. L.: Identifying the sources and sinks of CDOM/FDOM across the Mauritanian Shelf and their potential role in the decomposition of superoxide (O2-), Frontiers in Marine Science, 3, 132, https://doi.org/10.3389/fmars.2016.00132, 2016.

Homoky, W. B., Severmann, S., McManus, J., Berelson, W. M., Riedel, T. E., Statham, P. J., and Mills, R. A.: Dissolved oxygen and suspended particles regulate the benthic flux of iron from continental margins, Marine Chemistry, 134, 59-70, https://doi.org/10.1016/j.marchem.2012.03.003, 2012.

Homoky, W. B., Weber, T., Berelson, W. M., Conway, T. M., Henderson, G. M., van Hulten, M., Jeandel, C., Severmann, S., and Tagliabue, A.: Quantifying trace element and isotope fluxes at the ocean-sediment boundary: a review, Philos T R Soc A, 374, 20160246, https://doi.org/10.1098/rsta.2016.0246, 2016.

Honeyman, B. D., Balistrieri, L. S., and Murray, J. W.: Oceanic trace metal scavenging: the importance of particle concentration, Deep Sea Res Part A. Oceanogr Res Pap, 35(2), 227-246, https://doi.org/10.1016/0198-0149(88)90038-6, 1988.

Hydes, D. J. and Liss, P. S.: Fluorimetric method for determination of low concentrations of dissolved aluminum in natural waters, Analyst, 101, 922-931, https://doi.org/10.1039/an9760100922, 1976.

Jickells, T. D.: The inputs of dust derived elements to the Sargasso Sea; a synthesis, Mar Chem, 68(1-2), 5-14, https://doi.org/10.1016/S0304-4203(99)00061-4, 1999.

Kock, A., Schafstall, J., Dengler, M., Brandt, P., and Bange, H. W.: Sea-to-air and diapycnal nitrous oxide fluxes in the eastern tropical North Atlantic Ocean, Biogeosciences, 9, 957-964, https://doi.org/10.5194/bg-9-957-2012, 2012.

Lam, P. J. and Bishop, J. K. B.: The continental margin is a key source of iron to the HNLC North Pacific Ocean, Geophys Res Lett, 35, L07608, https://doi.org/10.1029/2008gl033294, 2008.

Marsay, C. M., Sedwick, P. N., Dinniman, M. S., Barrett, P. M., Mack, S. L., & McGillicuddy, D. J.: Estimating the benthic efflux of dissolved iron on the Ross Sea continental shelf, Geophysical Research Letters, 41(21), 7576-7583, https://doi.org/10.1002/2014GL061684, 2014.

Measures, C. I. and Brown, E. T.: Estimating dust input to the Atlantic Ocean using surface water aluminium concentrations. In: The impact of desert dust across the Mediterranean, Guerzoni, S. and Chester, R. (Eds.), Environmental Science and Technology Library, Springer, Dordrecht, 1996.

Measures, C. I. and Vink, S.: On the use of dissolved aluminum in surface waters to estimate dust deposition to the ocean, Global Biogeochem Cy, 14, 317-327, https://doi.org/10.1029/1999gb001188, 2000.

Menzel Barraqueta, J.-L., Schlosser, C., Planquette, H., Gourain, A., Cheize, M., Boutorh, J., Shelley, R., Pereira, L. C., Gledhill, M., Hopwood, M. J., Lacan, F., Lherminier, P., Sarthou, G., and Achterberg, E. P.: Aluminium in the North Atlantic Ocean and the Labrador Sea (GEOTRACES GA01 section): roles of continental inputs and biogenic particle removal, Biogeosciences, 15, 5271-5286, https://doi.org/10.5194/bg-15-5271-2018, 2018.

Menzel Barraqueta, J.-L., Klar, J. K., Gledhill, M., Schlosser, C., Shelley, R., Planquette, H. F., Wenzel, B., Sarthou, G., and Achterberg, E. P.: Atmospheric deposition fluxes over the Atlantic Ocean: a GEOTRACES case study, Biogeosciences, 16, 1525-1542, https://doi.org/10.5194/bg-16-1525-2019, 2019.

Micinski, E., Ball, L. A., and Zafiriou, O. C.: Photochemical oxygen activation - superoxide radical detection and production-rates in the Eastern Caribbean, *J Geophys Res Oceans* 98, 2299–2306. https://doi.org/10.1029/92JC02766, 1993.

Middag, R., de Baar, H. J. W., Laan, P., and Huhn, O.: The effects of continental margins and water mass circulation on the distribution of dissolved aluminum and manganese in Drake Passage, J Geophys Res-Oceans, 117, C01019, https://doi.org/10.1029/2011jc007434, 2012.

Millero, F. J., Sotolongo, S., & Izaguirre, M.: The oxidation kinetics of Fe (II) in seawater, Geochimica et Cosmochimica Acta, 51(4), 793-801, https://doi.org/10.1016/0016-7037(87)90093-7, 1987.

Milne, A., Schlosser, C., Wake, B. D., Achterberg, E. P., Chance, R., Baker, A. R., Forryan, A., and Lohan, M. C.: Particulate phases are key in controlling dissolved iron concentrations in the (sub)tropical North Atlantic, Geophys Res Lett, 44, 2377-2387, https://doi.org/10.1002/2016gl072314, 2017.

Moran, S. B. and Moore, R. M.: The potential source of dissolved aluminum from resuspended sediments to the North Atlantic Deep Water, Geochim Cosmochim Ac, 55, 2745-2751, https://doi.org/10.1016/0016-7037(91)90441-7, 1991.

Ndoye, S., Capet, X., Estrade, P., Sow, B., Dagorne, D., Lazar, A., Gaye, A. and Brehmer, P.: SST patterns and dynamics of the southern Senegal-Gambia upwelling center, J Geophys Res Oceans, 119, 8315–8335, https://doi.org/10.1002/2014JC010242, 2014.

Orians, K. J. and Bruland, K. W.: Dissolved aluminum in the Central North Pacific, Nature, 316, 427-429, https://doi.org/10.1038/316427a0, 1985.

Orians, K. J. and Bruland, K. W.: The biogeochemistry of aluminum in the Pacific Ocean, Earth Planet Sc Lett, 78, 397-410, https://doi.org/10.1016/0012-821x(86)90006-3, 1986.

Osborn, T. R.: Estimates of the local rate of vertical diffusion from dissipation measurements, J Phys Oceanogr, 10, 83-89, https://doi.org/10.1175/15200485(1980)010<0083:Eotlro>2.0.Co;2, 1980.

Patey, M. D., Achterberg, E. P., Rijkenberg, M. J., and Pearce, R.: Aerosol time-series measurements over the tropical Northeast Atlantic Ocean: Dust sources, elemental composition and mineralogy, Mar Chem, 174, 103-119, https://doi.org/10.1016/j.marchem.2015.06.004, 2015.

Rhein, M., Dengler, M., Sültenfuß, J., Hummels, R., Hüttl-Kabus, S., and Bourles, B.: Upwelling and associated heat flux in the equatorial Atlantic inferred from helium isotope disequilibrium, J Geophys Res, 115, C08021, https://doi.org/10.1029/2009JC005772, 2010.

Ricciardulli, L., and Wentz, F. J.: Remote Sensing Systems ASCAT C-2015 Daily Ocean Vector Winds on 0.25 deg grid, Version 02.1. Santa Rosa, CA: Remote Sensing Systems. Available at www.remss.com/missions/ascat, 2016.

Rijkenberg, M. J. A., Steigenberger, S., Powell, C. F., van Haren, H., Patey, M. D., Baker, A. R., and Achterberg, E. P.: Fluxes and distribution of dissolved iron in the eastern (sub-) tropical North Atlantic Ocean, Global Biogeochem Cy, 26, GB3004, https://doi.org/10.1029/2011gb004264, 2012.

Rudnick, R. L., and Gao, S.: Composition of the continental crust. In: Treatise on geochemistry, Holland, H. D. and Turekian, K. K. (Eds.), Pergamon, Oxford, UK, 2006.

Saito, M. A., Noble, A. E., Hawco, N., Twining, B. S., Ohnemus, D. C., John, S. G., Lam, P., Conway, T. M., Johnson, R., Moran, D., and McIlvin, M.: The acceleration of dissolved cobalt's ecological stoichiometry due to biological uptake, remineralization, and scavenging in the Atlantic Ocean, Biogeosciences, 14, 4637-4662, https://doi.org/10.5194/bg-14-4637-2017, 2017.

Schafstall, J., Dengler, M., Brandt, P., and Bange, H.: Tidal-induced mixing and diapycnal nutrient fluxes in the Mauritanian upwelling region, J Geophys Res-Oceans, 115, C10014, https://doi.org/10.1029/2009jc005940, 2010.

Schlosser, C., Streu, P., Frank, M., Lavik, G., Croot, P. L., Dengler, M., and Achterberg, E. P.: $H_2S$ events in the Peruvian oxygen minimum zone facilitate enhanced dissolved Fe concentrations, Sci Rep, 8, https://doi.org/10.1038/s41598-018-30580-w, 2018.

Scholz, F., Severmann, S., McManus, J., Noffke, A., Lomnitz, U., and Hensen, C.: On the isotope composition of reactive iron in marine sediments: Redox shuttle versus early diagenesis. Chemical Geology, 389, 48-59, https://doi.org/10.1016/j.chemgeo.2014.09.009, 2014.

Schroller-Lomnitz, U., Hensen, C., Dale, A. W., Scholz, F., Clemens, D., Sommer, S., Noffke, A., and Wallmann, K.: Dissolved benthic phosphate, iron and carbon fluxes in the Mauritanian upwelling system and implications for ongoing deoxygenation, Deep-Sea Res Pt I, 143, 70-84, https://doi.org/10.1016/j.dsr.2018.11.008, 2019.

Schütte, F., Brandt, P., and Karstensen, J.: Occurrence and characteristics of mesoscale eddies in the tropical north-east Atlantic Ocean, Ocean Sci., 12, 663-685, doi:10.5194/os-12-663-2016, 2016.

Severmann, S., McManus, J., Berelson, W. M., and Hammond, D. E.: The continental shelf benthic iron flux and its isotope composition, Geochim Cosmochim Ac, 74, 3984-4004, https://doi.org/10.1016/j.gca.2010.04.022, 2010.

Shelley, R. U., Morton, P. L., and Landing, W. M.: Elemental ratios and enrichment factors in aerosols from the US-GEOTRACES North Atlantic transects, Deep-Sea Res Pt II, 116, 262-272, https://doi.org/10.1016/j.dsr2.2014.12.005, 2015.

Shelley, R. U., Landing, W. M., Ussher, S. J., Planquette, H., and Sarthou, G.: Regional trends in the fractional solubility of Fe and other metals from North Atlantic aerosols (GEOTRACES cruises GA01 and GA03) following a two-stage leach, Biogeosciences, 15, 2271-2288, https://doi.org/10.5194/bg-15-2271-2018, 2018.

Sherrell, R. M., and Boyle, E. A.: The trace metal composition of suspended particles in the oceanic water column near Bermuda, Earth Planet Sc Lett, 111, 155-174, https://doi.org/10.1016/0012-821x(92)90176-V, 1992.

Sommer, S., Dengler, M., and Treude, T.: Benthic element cycling, fluxes and transport of solutes across the benthic boundary layer in the Mauritanian oxygen minimum zone, (SFB754) – Cruise No. M107 – May 30 – July 03, 2014 – Fortaleza (Brazil) – Las Palmas (Spain), METEOR-Berichte, M107, 54 pp., DFG-Senatskommission für Ozeanographie, https://doi.org/10.2312/cr_m107, 2015.

Steinfeldt, R., Sultenfuss, J., Dengler, M., Fischer, T., and Rhein, M.: Coastal upwelling off Peru and Mauritania inferred from helium isotope disequilibrium, Biogeosciences, 12, 7519-7533, https://doi.org/10.5194/bg-12-7519-2015, 2015.

Tanhua, T., and Liu, M.: Upwelling velocity and ventilation in the Mauritanian upwelling system estimated by CFC-12 and SF6 observations, J Mar Sys, 151, 57-70, https://doi.org/10.1016/j.jmarsys.2015.07.002, 2015.

Thomsen, S., Karstensen, J., Kiko, R., Krahmann, G., Dengler, M., and Engel, A.: Remote and local drivers of oxygen and nitrate variability in the shallow oxygen minimum zone off Mauritania in June 2014, Biogeosciences, 16, 979-998, https://doi.org/10.5194/bg-16-979-2019, 2019.

Tweddle, J. F., Sharples, J., Palmer, M. R., Davidson K., and McNeill, S.: Enhanced nutrient fluxes at the shelf sea seasonal thermoclinecaused by stratified flow over a bank, Prog Oceanogr, 117, 37–47, https://doi.org/10.1016/j.pocean.2013.06.018, 2013.

Ussher, S. J., Achterberg, E. P., Powell, C., Baker, A. R., Jickells, T. D., Torres, R., and Worsfold, P. J.: Impact of atmospheric deposition on the contrasting iron biogeochemistry of the North and South Atlantic Ocean, Global Biogeochem Cy, 27, 1096-1107, https://doi.org/10.1002/gbc.20056, 2013.

Verhoef, A., Portabella, M., and Stoffelen, A.: High resolution ASCAT scatterometer winds near the coast, IEEE, Trans Geosci Remote Sens, 50, 2481-248, https://doi.org/10.1109/TGRS.2011.2175001, 2012.

Wuttig, K., Heller, M. I., and Croot, P. L.: Pathways of Superoxide ($O_2^-$) Decay in the Eastern Tropical North Atlantic, Environ Sci Technol, 47(18), 10249-10256, https://doi.org/10.1021/es401658t, 2013.

Yücel, M., Beaton, A. D., Dengler, M., Mowlem, M. C., Sohl, F., and Sommer, S.: Nitrate and Nitrite Variability at the Seafloor of an Oxygen Minimum Zone Revealed by a Novel Microfluidic In-Situ Chemical Sensor, PLoS ONE 10(7), e0132785, https://doi.org/10.1371/journal.pone.0132785, 2015.

---

## Author Comment (AC2) · 30 Apr 2019

Response to Referee 2

Referee general comment: This is an important paper that illustrates the potential importance of benthic shelf sediments as a source of Fe to the oceans interior, and linking spatial and temporal variability to oxygen concentrations. While I think the authors make a good case for the role of oxygen, the paper is flawed by three serious omissions that must be corrected before publication.

Response: We thank the reviewer for the constructive comments and address each of them below.

Referee Comment 1. Iodide is reported as a critical parameter in the principal component analysis and there are detailed protocols for iodide analysis, yet no data are reported in the paper or in the supplement. These data are of great interest in their own right. While iodide has been reported in truly anaerobic, denitrifying water columns, it has not been well studied in these low oxygen regimes. One presumes that these data will appear in Pangaea eventually, but why not here?

Response to Comment 1: Thanks for pointing out the importance of our iodide data. We have now added a plot showing the spatial distribution of iodide in the Supplementary Material (plot shown below). The data will also be uploaded in Pangea.

Referee Comment 2. Similarly, Ra-228 data are not reported, nor is there any quantitative assessment of Ra-228 correlations with Fe to support their conclusions about lateral transport.

Response to Comment 2: Unfortunately Ra-228 was not determined along our transect. Therefore, we can't provide such information. Due to the long half-life of 228Ra of 5.7 years, this isotope would show sediment-water interaction on longer time scales. In our study close to the source, shorter timescales are of larger interest. Additionally, there doesn't necessarily need to be a correlation between Fe and 228Ra when Fe is released from sediments, because 228Ra behaves conservative, whereas iron is removed by scavenging.

Referee Comment 3. The authors imply that the approach used to determine vertical eddy diffusivity will appear in the Supplementary materials, but it does not.

Response to Comment 3: The approach used to determine vertical eddy diffusivity is explained in detail in the method section of the main manuscript in the paragraph starting at Page 8, Line 9. For a more detailed description, we refer to Schafstall et al. (2010).

[Figure]

[Figure]

**Fig. 1.** Figure S1. Spatial distributions of Iodide across the Mauritanian shelf at 18°20'N in June 2014. Each sample location is indicated as black dot and oxygen contours at 50 $\mu$mol kg-1 enclosing the upper

---

## Referee Report (RR1)

bg-2018-472
Controls on redox-sensitive trace metals in the Mauritanian oxygen minimum zone

***Comments on revised manuscript:***

The authors have improved the manuscript significantly along the lines suggested by the reviewers. There is however still an issue regards oceanographic nomenclature, the authors suggest that during the two occupations of the same sampling site, the water masses are the same. This description though is based on a T/S relationship, which might have been fine for physical oceanography in the past, but a broader chemical oceanography view, taking in oxygen and nutrients for instance, would indicate that they are distinctly different water masses with regard to their recent history. In this case they could be described as water parcels instead of water masses.

***Specific comments:***

*Calculation of $K_Z$ using microstructure probe*: In the reply to my comment it was mentioned that the microstructure profiler data near the bottom, not trace metal data, were not needed to determine the diffusive and advective fluxes between 20 and 50 m. The data from the benthic boundary layer would be useful to include in the context of recent work from the Peruvian OMZ (Croot et al., 2019) where Fe(II) distributions were used to calculate the Fe(II) fluxes from the sediments, and also the Oregon coast where $O_2$ variations in the sediments were examined (McCann-Grosvenor et al., 2014).

*Vertical velocities used in calculations and their contribution to the calculated fluxes:* One thing that still was not clear in the revised manuscript was what the vertical advection contribution was to the flux estimates. Including a paragraph on the relative contributions would be useful in this respect as it still looks like in many cases this is a key term because of the high concentrations. Also in this context, near the benthic boundary layer tides [e.g. Peru (Mosch et al., 2012), Oregon (McCann-Grosvenor et al., 2014).] may have a significant vertical velocity associated with them (Trowbridge and Lentz, 2018) and in particular for 'updraft events' related to the tides (Sevadjian et al., 2015) how would this then impact the estimation of the fluxes? The reason why I mention this is that in the present work the flux calculations are still not discussed in detail as to the contributions and uncertainties.

*Low $O_2$ waters arriving from the south*: In the context of the authors work, another recent work has shown that there has been very low $O_2$ concentrations in near surface waters to the south of the present study site and in the path of waters to that region (Machu et al., 2019).

**References cited:**

Croot, P.L., Heller, M.I. and Wuttig, K., 2019. Redox Processes Impacting the Flux of Iron(II) from Shelf Sediments to the OMZ along the Peruvian Shelf. ACS Earth and Space Chemistry, 3(4): 537-549.

Machu, E. et al., 2019. First Evidence of Anoxia and Nitrogen Loss in the Southern Canary Upwelling System. Geophysical Research Letters, 46(5): 2619-2627.

McCann-Grosvenor, K., Reimers, C.E. and Sanders, R.D., 2014. Dynamics of the benthic boundary layer and seafloor contributions to oxygen depletion on the Oregon inner shelf. Continental Shelf Research, 84: 93-106.

Mosch, T. et al., 2012. Factors influencing the distribution of epibenthic megafauna across the Peruvian oxygen minimum zone. Deep Sea Research Part I: Oceanographic Research Papers, 68(0): 123-135.

Sevadjian, J.C., McPhee-Shaw, E.E., Raanan, B.Y., Cheriton, O.M. and Storlazzi, C.D., 2015. Vertical convergence of resuspended sediment and subducted phytoplankton to a persistent detached layer over the southern shelf of Monterey Bay, California. Journal of Geophysical Research: Oceans, 120(5): 3462-3483.

Trowbridge, J.H. and Lentz, S.J., 2018. The Bottom Boundary Layer. Annual Review of Marine Science, 10(1): 397-420.

---

## Author Response (AR2)

Response to Referee 1

*Comments on revised manuscript:*
The authors have improved the manuscript significantly along the lines suggested by the reviewers. There is however still an issue regards oceanographic nomenclature, the authors suggest that during the two occupations of the same sampling site, the water masses are the same. This description though is based on a T/S relationship, which might have been fine for physical oceanography in the past, but a broader chemical oceanography view, taking in oxygen and nutrients for instance, would indicate that they are distinctly different water masses with regard to their recent history. In this case they could be described as water parcels instead of water masses.

**We would like to thank the reviewer for the constructive comments on the revised version of our manuscript. We address each specific comment below. The responses to the reviewer are given in bold font. Changes we made to the manuscript are highlighted in blue. Page and line numbers refer to the position in the revised manuscript.**

**Thanks for pointing out the ambiguous nomenclature – we have changed the use of the term "water mass" to "water parcel" at the relevant passage (Page 23, Line 11-12):**

In contrast, no changes in temperature and salinity of the water parcel occurred below 50 m (Fig. 7a).

*Specific comments:*
*Calculation of $K_z$ using microstructure probe*: In the reply to my comment it was mentioned that the microstructure profiler data near the bottom, not trace metal data, were not needed to determine the diffusive and advective fluxes between 20 and 50 m. The data from the benthic boundary layer would be useful to include in the context of recent work from the Peruvian OMZ (Croot et al., 2019) where Fe(II) distributions were used to calculate the Fe(II) fluxes from the sediments, and also the Oregon coast where $O_2$ variations in the sediments were examined (McCann-Grosvenor et al., 2014).

**As mentioned by the reviewer, we could determine dissolved Fe fluxes in the bottom boundary layer, however not Fe(II) fluxes, as only measurements of the total dissolved Fe fraction (Fe(II) + Fe(III)) are available from the water column here. Adding those fluxes and discussion would require the addition of a whole additional chapter. We don't feel that the benefits of adding such data would justify further extending the already long manuscript.**

*Vertical velocities used in calculations and their contribution to the calculated fluxes:* One thing that still was not clear in the revised manuscript was what the vertical advection contribution was to the flux estimates. Including a paragraph on the relative contributions would be useful in this respect as it still looks like in many cases this is a key term because of the high concentrations. Also in this context, near the benthic boundary layer tides [e.g. Peru (Mosch et al., 2012), Oregon (McCann-Grosvenor et al., 2014).] may have a significant vertical velocity associated with them (Trowbridge and Lentz, 2018) and in particular for 'updraft events' related to the tides (Sevadjian et al., 2015) how would this then impact the estimation of the fluxes? The reason why I mention this is that in the present work the flux calculations are still not discussed in detail as to the contributions and uncertainties.

**The relative contributions of advective and diffusive fluxes and large uncertainties are discussed in the manuscript on Page 20 and 21 and the numbers are detailed in Table S2. To make it easier for the reader to follow the respective contributions we added the relevant numbers from Table S2 and details on estimated uncertainty calculations to the main text on Page 20 and 21. The modified section reads as follows:**

**Page 20, Line 18-35:** Vertical dFe fluxes increased by two orders of magnitude from 70 km offshore to the shallow shelf region. On the shelf (bottom depth: 50 m), an elevated mean dFe flux of 13.5 $\mu$mol m$^{-2}$ d$^{-1}$ was estimated. The contribution from vertical advection (upwelling) here (11.99 $\mu$mol m$^{-2}$ d$^{-1}$) was an order of magnitude larger than the diffusive flux (1.56 $\mu$mol m$^{-2}$ d$^{-1}$). Our estimate agrees with a reported vertical dFe flux of 16 $\mu$mol m$^{-2}$ d$^{-1}$ on the shelf at 12°N (Milne et al., 2017). Average estimates from the upper continental slope and the lower shelf region (stations 3, 7 and 8, bottom depth: 90–300 m) were between 1 µmol m$^{-2}$ d$^{-1}$ and 2.5 µmol m$^{-2}$ d$^{-1}$. Here, the vertical diffusive fluxes dominated (0.72–1.75 µmol m$^{-2}$ d$^{-1}$) and were about a factor of three larger than vertical advective fluxes (0.22–0.68 µmol m$^{-2}$ d$^{-1}$). The elevated diffusive fluxes at the upper continental slope and lower shelf region are due to enhanced diapycnal mixing that originates from tide – topography interactions (Schafstall et al., 2010). At 170 m depth of the repeated station (3), vertical dFe flux estimates were 2.3 µmol m$^{-2}$ d$^{-1}$ and 1.4 µmol m$^{-2}$ d$^{-1}$, respectively. The differences in the two values are due to differences in the strength of turbulent mixing during the two station occupations. For the offshore stations 2 and 9 (bottom depth > 500 m), mean dFe fluxes were 0.08–0.16 µmol m$^{-2}$ d$^{-1}$ with similar contributions of diffusive and advective fluxes. However, one offshore station (station 5) exhibited elevated dFe fluxes of 1.3 µmol m$^{-2}$ d$^{-1}$ with a large contribution of the diffusive flux term (1.03 µmol m$^{-2}$ d$^{-1}$). Here, diapycnal mixing was determined from only 5 microstructure profiles that exhibited elevated turbulence levels. It is thus very likely that the observations captured a rare elevated mixing event during station occupation and the associated elevated vertical fluxes do not represent a longer-term average.

**Page 21, Line 9-15:** Uncertainties in the diffusive flux originate predominately from the elevated variability of turbulence (see Schafstall et al., 2010 for details) and were calculated here using the upper and lower 95% confidence interval of diffusivity measurements. Uncertainties in the vertical advective flux originate from unaccounted for contributions from e.g. the spatial structure of the wind, particularly in the offshore direction, its temporal variability (e.g. Capet et al., 2004; Desbiolles et al. 2014, 2016; Ndoye et al., 2014), and uncertainties in the satellite wind product near the coast (e.g. Verhoef et al, 2012), and were accounted for by using an estimated error of 50% for the upwelling velocity. Furthermore, the distribution of vertical velocities with depth is assumed to be linear here.

*Low O$_2$ waters arriving from the south*: In the context of the authors work, another recent work has shown that there has been very low O$_2$ concentrations in near surface waters to the south of the present study site and in the path of waters to that region (Machu et al., 2019).
**We thank the reviewer for pointing out this important study in this region and included the reference with its major findings relevant for our study at Page12, Line 15 as follows:**

[revised manuscript text omitted]